# Paraspeckle condensation is controlled via TDP-43 polymerization and linked to neuroprotection

Rachel E. Hodgson[1,42], Wan-Ping Huang[1,42], Ruaridh Lang[1,42], Vedanth Kumar[1], Haiyan An[2], Emil G. P. Stender[3], Zhaklin P. Chalakova[1,40], Mark D. Driver[4,41], Anna Sanchez Avila[1], Brittany C. S. Ellis[1], Emily Day[1], Jessica A. Rayment[1], Kyungmin Baeg[5], Andrew Strange[1], Tobias Moll[1], Gareth S. A. Wright[6], Joke J. F. A. van Vugt[7], Project MinE ALS Sequencing Consortium*, Scott P. Allen[1], Nicolas Locker[8], Ianthe Pitout[9], Susan Fletcher[9], Patrick R. Onck[4], Olivier Duss[5], Johnathan Cooper-Knock[1] & Tatyana A. Shelkovnikova[1] ✉

The paraspeckle is a disease-relevant biomolecular condensate assembled from long non-coding RNA (lncRNA) NEAT1_2 ribonucleoprotein particles. Paraspeckle biogenesis is suppressed in normal tissues, yet it can be rapidly upregulated under stress. Here we demonstrate that a neurodegeneration-linked RNA-binding protein TDP-43 inhibits NEAT1_2 ribonucleoprotein particle condensation into the paraspeckle, in a concentration-dependent manner, which requires its intact polymerization and RNA binding. This effect is counterbalanced by core paraspeckle proteins such as FUS. Below disruptive concentrations, TDP-43 can be recruited into paraspeckles, forming non-liquid clusters. Under stress, TDP-43 sequestration into de novo nuclear condensates alleviates paraspeckle suppression and increases their dynamism. NEAT1_2 middle-part and 3′-end UG repeats mediate paraspeckle regulation by TDP-43 cotranscriptionally and post assembly, respectively. The deletion of the 3′-end UG repeat increases paraspeckle stability and cytoprotection in stressed human neurons. Consistently, longer 3′-end UG repeats are linked to shorter survival in the neurodegenerative disease amyotrophic lateral sclerosis. Thus, TDP-43 is a critical regulator of paraspeckle condensates linked to cytoprotection.

Paraspeckles are nuclear membraneless organelles (MLOs), or biomolecular condensates, seeded and sustained by an 'architectural' long non-coding RNA (lncRNA) NEAT1_2[1]. Paraspeckle roles described so far include the regulation of gene expression, via the sequestration of transcription factors and RNAs[1,2]. NEAT1_2/paraspeckles are absent or very sparse in the majority of healthy mammalian tissues[3] but are upregulated during development[4,5], by environmental stresses[6,7] and in disease, including cancer and neurodegeneration[8,9]. For example, although paraspeckles are almost undetectable in healthy postmitotic neurons in human tissue, their de novo assembly was observed in the motor neurons of patients with amyotrophic lateral sclerosis (ALS)[10–12].

The NEAT1 locus gives rise to two NEAT1 isoforms overlapping in the 5′-end and produced by differential 3′-end processing; only the long, read-through NEAT1_2 transcript is required for paraspeckle formation[13–15]. NEAT1_2 forms complexes with SFPQ/NONO heterodimers and other proteins cotranscriptionally, assembling NEAT1

A full list of affiliations appears at the end of the paper. *A list of authors and their affiliations appears at the end of the paper.
✉e-mail: t.shelkovnikova@sheffield.ac.uk

ribonucleoprotein particles (NEAT1 RNPs),[15,16] followed by the coalescence of these paraspeckle precursors into mature condensates—paraspeckle 'spheroids'[17,18]. This assembly step is primarily driven by the liquid–liquid phase separation (LLPS) activity of FUS protein, with contribution from other paraspeckle proteins (PSPs)[18]. Within the spheroid, NEAT1_2 termini and the middle region localize to the outer layer ('shell') and centre ('core'), respectively[18].

TDP-43 is an RNA-binding protein with a wide expression pattern and a plethora of regulatory roles[19]. TDP-43 dysfunction is considered a main culprit in neurodegenerative diseases such as ALS[20]. In addition to being mutated in a small subset of familial ALS cases, TDP-43 is ubiquitously dysregulated in sporadic ALS[21]. TDP-43 is a component of several MLOs, including paraspeckles[15,19]. Interestingly, in contrast to most other PSPs, TDP-43 localizes to the paraspeckle spheroid shell[18]. This protein was also found to regulate the NEAT1 isoform ratio[4]. However, the role of TDP-43 in paraspeckle condensate regulation, beyond NEAT1 isoform levels, remains largely unknown.

Here, we demonstrate, using single-molecule dual RNA fluorescence in situ hybridization (RNA-FISH), super-resolution microscopy (SRM), mutagenesis, fluorescence recovery after photobleaching (FRAP), optogenetics, high-content imaging, in vitro reconstitution of condensates, in silico analysis, engineered cell lines, physiological neuronal models and a large human genetic dataset, that TDP-43 is a key regulator of paraspeckle condensate biogenesis downstream of NEAT1_2 expression—a mechanism realized via TDP-43 self-assembly and position-dependent binding on NEAT1_2 and relevant to the cellular stress response and neurodegeneration.

## Results

### TDP-43 is a negative regulator of paraspeckle condensation

While studying TDP-43 localization to paraspeckles, we noticed a striking defect in paraspeckle spheroid assembly upon TDP-43 overexpression (OE). HeLa, SH-SY5Y or MCF7 cells expressing exogenous TDP-43 lost large NEAT1_2-positive foci and instead accumulated smaller NEAT1-positive particles (Fig. 1a and Extended Data Fig. 1a). In HeLa cells that possess more prominent paraspeckles than the other two lines, residual large NEAT1 foci were sometimes detectable (Fig. 1a). Paraspeckle dispersal by TDP-43 was concentration-dependent, with paraspeckles dispersed in 91.8% of high–medium-expressing but only in 12.5% of low-expressing cells (Fig. 1b–d). The ectopic expression of GFP alone, shell-localized PSPs other than TDP-43 (RBM14, BRG1, hnRNPF)[17,18], core-localized PSPs (SFPQ, NONO, FUS)[18] or cytoplasmically localized TDP-43ΔNLS did not cause paraspeckle dispersal (Extended Data Fig. 1a–c). As large tags and GFP dimerization can affect the properties of tagged proteins, we sought to validate these findings

using different systems. Untagged TDP-43 and TDP-43 with SNAP-tag were also found to disrupt paraspeckles (Extended Data Fig. 1d–f), confirming that it is a TDP-43-specific effect. On average, ~3.5-fold TDP-43 upregulation over the endogenous led to detectable paraspeckle dispersal in HeLa cells (Extended Data Fig. 1g,h). TDP-43 GFP, which was most efficient in paraspeckle disruption (~37% transfected cells), was primarily used for mechanistic studies.

Using dual NEAT1 RNA-FISH followed by SRM (structured illumination microscopy, SIM or Airyscan), we first confirmed that the core–shell spheroid structure can be visualized using commercial NEAT1 (Stellaris) probes. These probes span longer regions than those used in the original study[18] (1–3,700 for the 5′-end and 3,800–11,700 for the middle region versus 15–1,066 and 7,231–9,403, respectively) but still provide a sufficiently tight/focused signal for the core–shell analysis (Fig. 1e,f). The SRM revealed that small NEAT1-positive foci in TDP-43-OE cells correspond to paraspeckle precursors—NEAT1 RNPs, detected as paired red (core)–green (shell) signals (Fig. 1g). In addition to the individual NEAT1 RNPs ('singlets'), we detected doublets, triplets and a mixed population of 'multiplets'. The quantification (>400 units analysed) revealed a prevalence of singlets and doublets (68.2% of counted particles) (Fig. 1g). Notably, in TDP-43-OE cells, the middle parts of precursors (red) were joined, whereas the termini (green) were typically found apart (for example, Fig. 1g, circle). Similar results were obtained with a custom 3′-end NEAT1 probe (20,210–22,744, data not shown). The residual NEAT1-positive clusters present in a proportion of cells with high TDP-43 OE (Fig. 1a, arrow) were found to represent disordered collections of NEAT1 RNPs largely devoid of intact spheroids (Fig. 1h). TDP-43 was enriched in these RNP aggregates (Extended Data Fig. 1i) and in individual NEAT1 RNPs (Fig. 1i).

We next sought to validate these findings in a physiological model and with homogeneous TDP-43 expression. Paraspeckle assembly oscillates during differentiation[4] and has been linked to neurodegeneration[10]; therefore, we utilized human induced pluripotent stem cell (iPS cell)-derived neural precursors and motor neurons. Paraspeckles are readily detectable in day 16 neural precursors but are very low in motor neurons[10]. Untagged TDP-43 was ectopically expressed, using lentiviral delivery, in day 36 human motor neurons, achieving a uniform, approximately twofold OE in both cell types (Fig. 1j). Owing to low NEAT1_2/paraspeckle levels in neurons, we reasoned that this level of TDP-43 OE would be sufficient for a detectable effect. To increase detection sensitivity, we used NEAT1_2 RNAscope-ISH, which uses signal amplification, after the validation of this approach in HeLa cells (Extended Data Fig. 1j). With these tools, we demonstrated that TDP-43 upregulation also leads to paraspeckle dispersal in neuronal cells (Fig. 1k and Extended Data Fig. 1k).

---

**Fig. 1 | TDP-43 regulates paraspeckle condensation. a**, Non-transfected (NT) cells with intact paraspeckle clusters (inset 1) and TDP-43-OE cells with dispersed paraspeckles (inset 2); the arrow indicates a TDP-43-OE cell with residual NEAT1-positive clusters. **b**, An example of low-expressing cells; cells were classified as high-, medium- or low-expressing. **c**, The quantification is for HeLa, from a representative experiment; the number of cells analysed (n) is indicated within bars. FL, full-length. **d**, The profiles were drawn from images in **a** (HeLa). In **a**–**d**, the paraspeckle dispersal in TDP-43 overexpressing cells. TDP-43 with a GFP tag on its N-terminus was used. Scale bars, 15 μm. **e**, The NEAT1_2 arrangement in the paraspeckle spheroid and positions of probes used for dual NEAT1 RNA-FISH. **f**, Paraspeckle spheroids and their higher-order assemblies in HeLa cells as detected by dual NEAT1 RNA-FISH and SIM. **g,h**, The spheroid disruption and precursor accumulation in HeLa cells with TDP-43 OE: diffuse (**g**) and clustered (**h**) TDP-43 distribution. s, singlet, d, doublet, t, triplet, m, multiplet. A doublet with disrupted NEAT1_2 end joining is circled. Representative images and quantification from a representative experiment are shown. The number of cells quantified (n) is indicated within bars. **i**, The TDP-43 GFP enrichment in NEAT1 RNPs. **j**, The TDP-43 OE confirmation in human motor neurons (hMNs) using lentiviral (LV) delivery. Representative images are shown. Untagged TDP-43 and day 36 hMNs were used. A representative western blot is shown (mock and

TDP-43 LV samples were run on the same gel). Scale bar, 10 μm. **k**, The increased TDP-43 dosage in hMNs leads to paraspeckle dispersal. RNAscope-ISH with a NEAT1_2 probe was used for amplified paraspeckle detection. Untagged TDP-43 and day 36 hMNs were used. Representative images and quantification from a representative experiment are shown. The number of cells quantified (n) is indicated within bars. Scale bar, 50 μm. **l,m**, The paraspeckle precursor: the spheroid ratio is cell line-specific: smaller NEAT1-positive particles corresponding to NEAT1 RNPs and bundles are more abundant than paraspeckle spheroids (**l**), and NEAT1_2 is more extractable (**m**) in SH-SY5Y cells as compared with HeLa. Arrows indicate spheroids. In **l**, quantification was performed on n = 10 cells from a representative experiment. ****P < 0.0001, one-tailed Mann–Whitney U test. In **m**, the semi-extractability was measured by RT–qPCR. Graphs show mean ± s.d. N = 4; *P < 0.05 (P = 0.0286), one-tailed Mann–Whitney U test. **n**, TDP-43 depletion shifts the equilibrium towards paraspeckle spheroids. Representative images and quantification of particle size in TDP-43-sufficient and -deficient SH-SY5Y cells. Shell (5′-end NEAT1) probe was used; s, spheroid; 288 and 216 particles were quantified for scrambled and TDP-43 siRNAs, respectively, from n ≥ 10 fields of view, from a representative experiment. ****P < 0.0001, one-tailed Student's t-test. Representative images are shown in **a**, **b**, **f** and **h**.

Having characterized the phenotypes elicited by increased TDP-43 dosage, we focused on the reciprocal condition—TDP-43 loss of function. First, we showed that in HeLa cells, TDP-43 depletion does not affect the core–shell arrangement of paraspeckle spheroids (Extended Data Fig. 2a,b). However, this cell line has a high prevalence

of spheroids and only a small precursor population, precluding an efficient analysis of precursor condensation. Thus, for TDP-43 depletion analysis, we used SH-SY5Y cells that have a significantly larger precursor pool, as confirmed by particle quantification and NEAT1_2 semi-extractability analysis (Fig. 1l,m and Extended Data Fig. 2c).

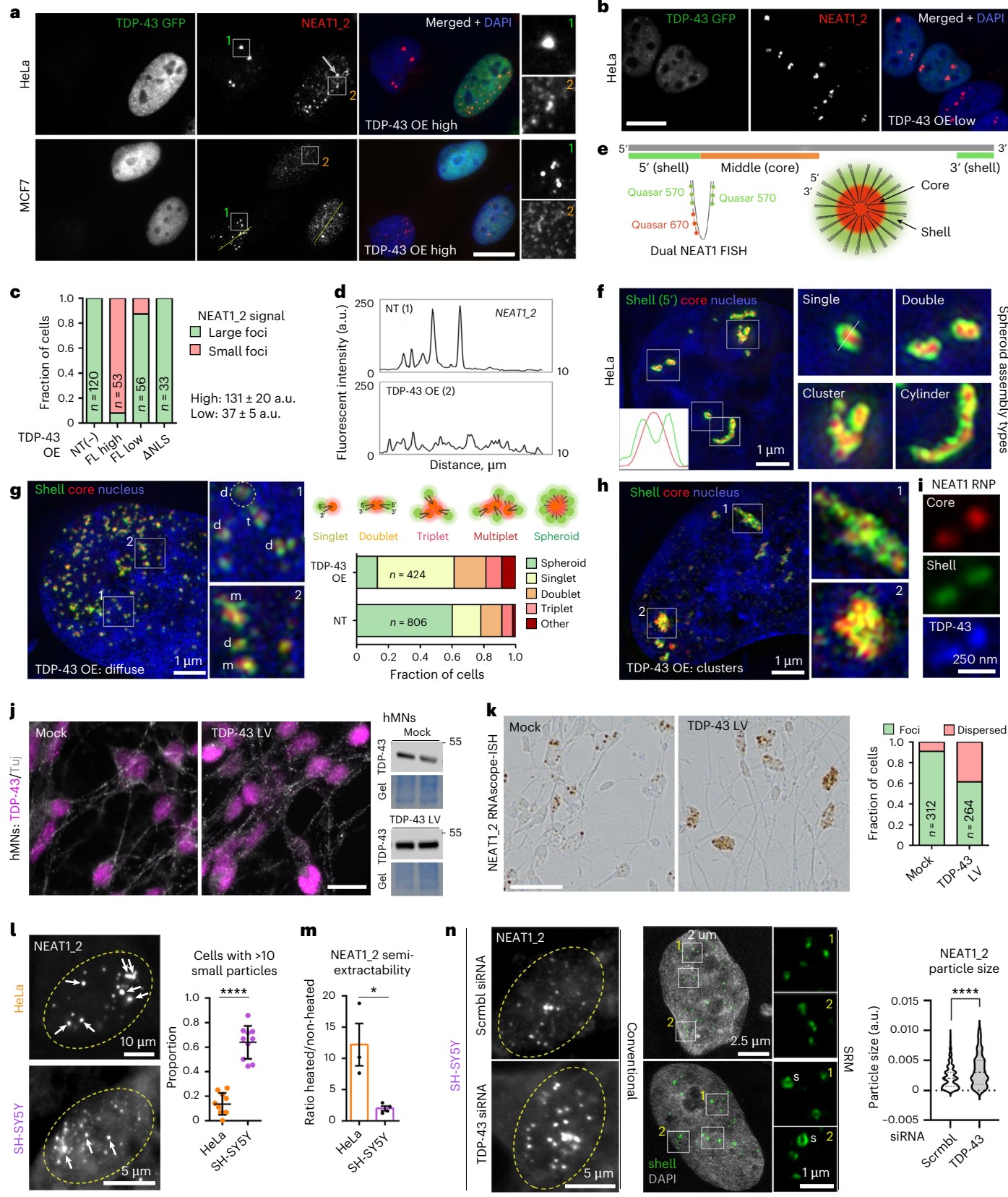

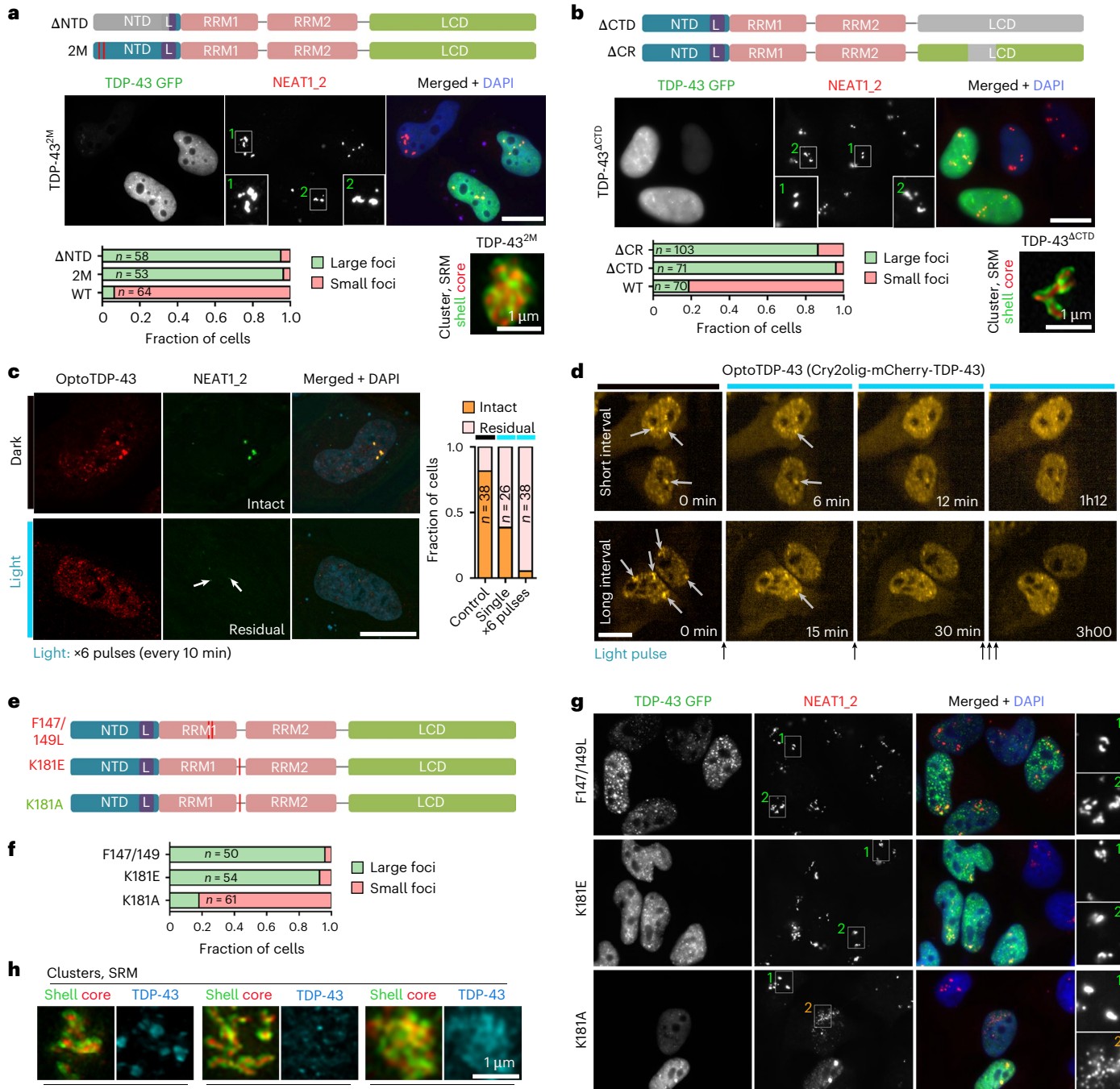

**Fig. 2 | TDP-43 oligomerization and RNA-binding are both required for paraspeckle condensate regulation. a,b,** Intact TDP-43 self-assembly is required for paraspeckle dispersal: TDP-43 N- (**a**) and C-terminal (**b**) mutants were used. Representative images of paraspeckle clusters (conventional and SRM) and their quantification, from a representative experiment, are shown. The number of cells (*n*) analysed is indicated within bars. Non-transfected (NT)/low-expressing (inset 1) and high-expressing cells (inset 2). **c,d,** The light-induced submicroscopic oligomerization of optoTDP-43 expressed at a near-endogenous level leads to paraspeckle disruption: representative images and their quantification (from a representative experiment) for NEAT1 RNA-FISH (**c**) and time-lapse imaging of cells with optoTDP-43-positive paraspeckle clusters (**d**) are shown. The number of cells (*n*) analysed is indicated within bars. Arrows indicate residual paraspeckles in light-stimulated cells in **c** or intact paraspeckle clusters before light stimulation in **d**. See also Extended Data Fig. 4 for more details. **e–h,** The paraspeckle spheroid disruption by TDP-43 is dependent on its RNA-binding capacity: TDP-43 mutants used (**e**), the quantification of paraspeckle phenotypes (**f**) and representative images of mutant TDP-43-OE cells (conventional microscopy (**g**) and SRM of paraspeckle clusters (**h**), from a representative experiment, are shown. The number of cells analysed (*n*) is indicated within bars. NT/low-expressing (inset 1) and high-expressing (inset 2) cells. Experiments were done in HeLa cells; only cells with high–medium TDP-43 GFP expression were included in the quantification. Scale bars, 10 μm.

The latter property is defined by NEAT1_2 incorporation into spheroids and measured by determining the increase in NEAT1_2 yield after sample heating during RNA isolation[12,22]. TDP-43 depletion caused a significant (*P* < 0.0001) shift from smaller to larger NEAT1 particles, as revealed by automated quantification with a threshold set to exclude spheroid clusters and capture only single spheroids and precursors (Fig. 1n). Thus, TDP-43 loss promotes NEAT1 RNP condensation and spheroid formation.

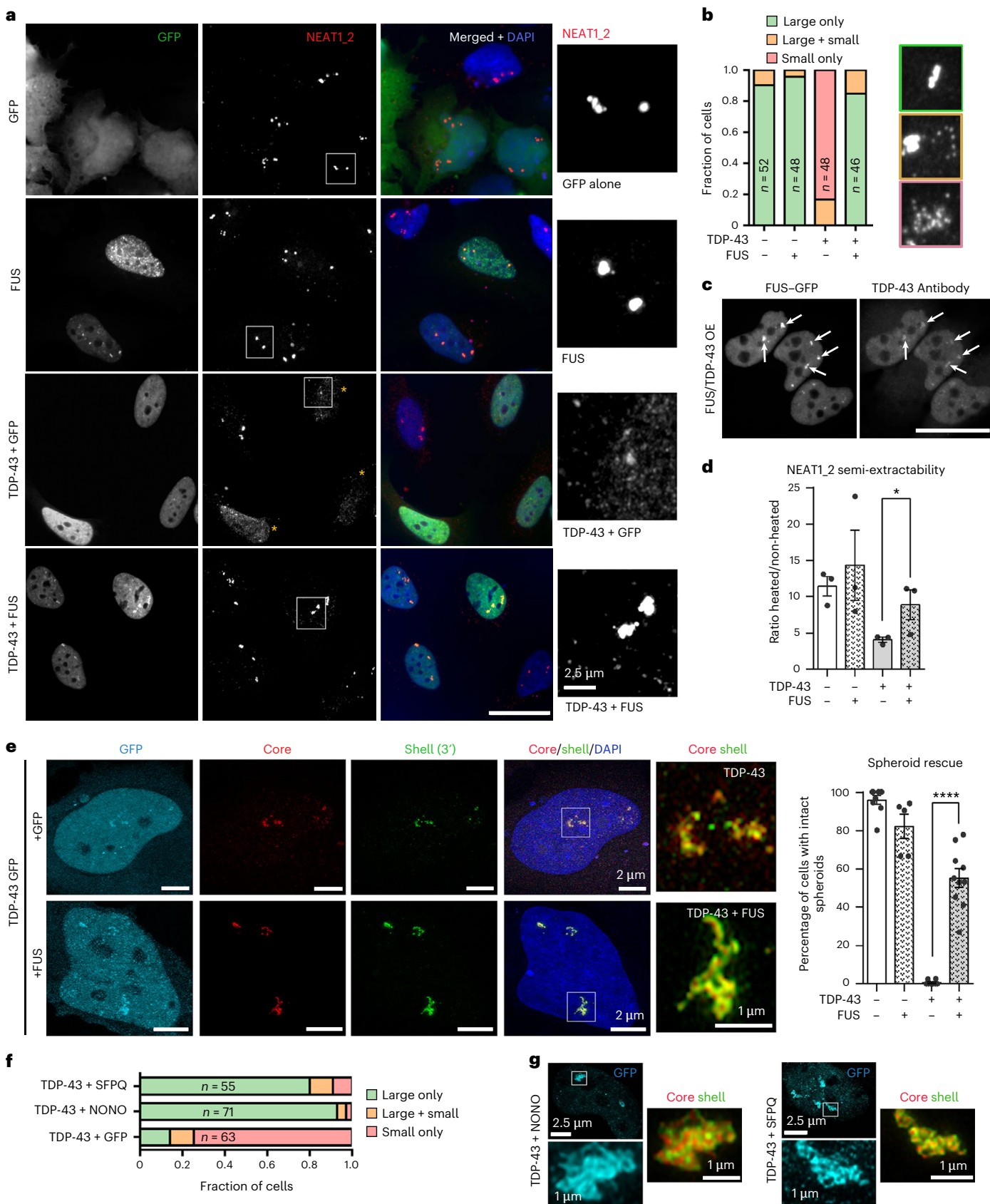

### TDP-43 oligomerization and RNA binding are both required for suppression of paraspeckle condensation

To establish whether TDP-43 self-assembly is required for its effect on paraspeckle condensation, we used a panel of TDP-43 point and deletion mutants (N-terminally tagged with GFP; Extended Data Fig. 3a). TDP-43 has a folded N-terminal domain (NTD) and a low-complexity C-terminal domain (CTD), both of which are important for its higher-order assembly[23]. By analysing TDP-43 high–medium-expressing cells with similar

**Fig. 3 | Increasing FUS dosage restores paraspeckle spheroid assembly in cells with TDP-43 OE. a,b,** The paraspeckle cluster restoration by FUS supplementation in TDP-43-OE cells: representative images (**a**) and the quantification of paraspeckle phenotypes (**b**) in cells coexpressing TDP-43 GFP with GFP or FUS–GFP are shown. Asterisks in **a** indicate cells with dispersed paraspeckles. The number of cells quantified in **b** (*n*) is shown within bars (from a representative experiment). Scale bar, 20 μm. **c,** TDP-43 is enriched within paraspeckle clusters (arrows) in FUS–GFP-OE cells. A representative image is shown. Scale bar, 20 μm. **d,** The TDP-43 OE-driven decrease in NEAT1_2 semi-extractability is rescued by FUS supplementation. NEAT1_2 semi-extractability was analysed by RT–qPCR. The graph shows the mean ± s.e.m. *N* = 4; *P = 0.05,

one-tailed Mann–Whitney *U* test. **e,** The FUS supplementation restores spheroid formation in TDP-43-OE cells. Representative images and the quantification of spheroid assembly in cells coexpressing TDP-43 GFP with GFP or FUS–GFP are shown. Spheroid numbers analysed: 424 (non-transfected, NT), 87 (FUS), 290 (TDP-43 + GFP), 308 (TDP-43 + FUS), from *n* = 5–10 fields of view (FoV), from a representative experiment. The graph shows the mean ± s.e.m. ****P < 0.0001, one-way ANOVA with Tukey post hoc test. **f,g,** The NONO or SFPQ supplementation restores spheroid formation in TDP-43-OE cells: the quantification (**f**) and SRM images (**g**) are shown, from a representative experiment. The number of cells (*n*) analysed in **f** is indicated within bars. Representative images are shown in **g**. HeLa cells were used.

nuclear levels across all variants, we found that as few as two point mutations in the NTD (E14A/E17A, TDP-43²ᴹ)[24] prevent paraspeckle dispersal; expectedly, this was also true for the TDP-43^ΔNTD mutant (amino acids (a.a.) 1–98 deleted) (Fig. 2a). Likewise, TDP-43^ΔCR, lacking the conserved α-helical self-assembly region in CTD[25,26] and TDP-43^ΔCTD mutant were both unable to disperse paraspeckles (Fig. 2b). Notably, although the reduced TDP-43 self-assembly impairs its nuclear retention in HEK293 cells[27,28], TDP-43^WT and the above mutants displayed similar nuclear–cytoplasmic ratios in HeLa cells, albeit with a minor increase in the cytoplasmic signal for the TDP-43²ᴹ variant (Extended Data Fig. 3b), which could be due to cell line-specific differences. Using disuccinimidyl glutarate (DSG) crosslinking, we verified that both TDP-43²ᴹ and TDP-43^ΔCR are deficient in homo-oligomerization (Extended Data Fig. 3c). All TDP-43 self-assembly mutants were nevertheless enriched in paraspeckles (Fig. 2a,b), despite lower nuclear TDP-43 granulation overall–consistent with a previous study[28]. Finally, we found that ALS-linked TDP-43 mutations affecting different protein domains retain their ability to disrupt paraspeckles upon OE (Extended Data Fig. 3d), supporting that it is a robust property of TDP-43 that can buffer mild structural defects.

Previously, TDP-43 loss was shown to increase paraspeckle formation owing to the isoform switch from NEAT1_1 to NEAT1_2, via a polyadenylation mechanism[4]. Yet, TDP-43 OE also results in NEAT1_2 accumulation[29] (Extended Data Fig. 3e). Using quantitative reverse transcription PCR (RT–qPCR), we found that whereas the OE of oligomerization-deficient mutants leads to NEAT1_2 downregulation, the OE of TDP-43^WT and its ALS-linked mutants upregulates NEAT1_2, without significant change to total NEAT1 levels (Extended Data Fig. 3e). These data suggest that NEAT1_1 production is promoted specifically by monomeric TDP-43, whereas the oligomerization of exogenous TDP-43 with its endogenous counterpart may lead to TDP-43 loss-of-function in NEAT1 isoform regulation.

NEAT1_2 was previously reported to have a minor spliced isoform lacking the 20,360–21,153 region, which is more extractable[22]; however, this isoform was undetectable by reverse transcription PCR (RT–PCR) upon either upon TDP-43 OE or its knockdown (Extended Data Fig. 3f).

To test the effect of enhanced TDP-43 self-association at a near-endogenous level, we generated 'optoTDP-43'–full-length TDP-43 N-terminally tagged with an opto-module Cry2olig, (Extended Data Fig. 4a,b). OptoTDP-43 expressing cells had near-physiological TDP-43 levels owing to endogenous TDP-43 downregulation (Extended Data Fig. 4c,d). We reasoned that the augmented TDP-43 polymerization triggered by blue light would be sufficient for paraspeckle dispersal in the absence of TDP-43 OE. Indeed, single or repetitive blue-light exposure led to a significant loss of large NEAT1_2 positive foci–paraspeckle clusters (Fig. 2c,d). By contrast, Cry2olig alone did not affect paraspeckle clusters upon blue light stimulation (Extended Data Fig. 4e).

To study TDP-43 RNA binding contribution, two RNA-binding deficient mutants, F147/149L[30] and K181E[31], were used, alongside the RNA-binding proficient K181A as a control (Fig. 2e). The RNA-binding deficient mutants formed abundant granules in the nucleus, in a concentration-dependent manner, probably corresponding to stable, non-LLPS aggregates[32]. The F147/149L and K181E substitutions, but not K181A, prevented paraspeckle dispersal (Fig. 2f,g), indicating that RNA binding is a prerequisite for paraspeckle condensate regulation by TDP-43. Residual NEAT1-positive clusters were found to be composed of intact spheroids in cells expressing F147/149 L or K181E but were disordered in K181A-expressing cells; only the latter variant was enriched in these RNP aggregates (Fig. 2h), similar to TDP-43^WT (Extended Data Fig. 1i).

## TDP-43 antagonizes core PSPs in paraspeckle condensation
The FUS protein plays a critical role in paraspeckle condensation, where its depletion precludes spheroid assembly, without affecting NEAT1 RNP abundance[18]. We tested whether FUS supplementation could counterbalance TDP-43's disruptive effect on paraspeckles. The coexpression conditions were established in titration experiments in HeLa cells, achieving TDP-43 and FUS coexpression in >95% cells, with ~3–4-fold OE of both proteins (Extended Data Fig. 5a–d). FUS–GFP, but not GFP alone, restored paraspeckle clusters in the

**Fig. 4 | TDP-43 inhibits FUS condensation in vitro. a,** The ImmuCon: assay schematic. **b,** Recombinant proteins used in the study. Representative western blots are shown. **c,** The FUS condensate and TDP-43 cluster/aggregate formation in vitro and their sensitivity to 1,6-HD. Representative images are shown. See also Extended Data Fig. 7 for more details. **d,** A circularity analysis of FUS and TDP-43 in vitro assemblies; 149 and 217 particles were included in the analysis (*n* = 5 fields of view (FoV)), from a representative experiment. The graph shows the mean ± s.d. **e,f,** The effect of TDP-43 on FUS condensates; representative images (**e**) and the quantification (**f**) are shown. TDP-43 supernatant fraction, at the indicated concentration, was added to FUS samples (2.5 μM), and condensates were fixed after a 10-min incubation. R, residual FUS condensate. The graph shows the mean ± s.e.m. *n* = 5–9 FoV for 0.0–2.5 μM and *n* = 2 FoV for 5 μM were analysed from a representative experiment. ****P < 0.0001, one-way ANOVA with Dunnett's post hoc test. **g,** The TDP-43 NTD (a.a. 1–98) does not prevent FUS condensation and remains soluble. Representative images and the quantification are shown. The graph shows the mean ± s.e.m. *n* = 5 FoV per condition were analysed from a representative experiment. n.s., non-significant, one-way ANOVA with Dunnett's post hoc test. **h,** The UG-rich RNA potentiates FUS condensate

disruption by TDP-43. FUS was used at 2.5 μM and TDP-43 at 0.1 μM. Both RNA oligonucleotides were used at 0.25 μM. The graph shows the mean ± s.e.m. *n* = 7 FoV per condition were analysed from a representative experiment. **P < 0.01 (*P* = 0.002), ***P < 0.001 (*P* = 0.0003), one-tailed Mann–Whitney *U* test. **i,** The UG-rich RNA promotes TDP-43 clustering in vitro, as demonstrated by ImmuCon. TDP-43 supernatant fraction (0.5 μM) was mixed with 0.25 μM RNA oligonucleotide. Number of bigger clusters (particles ≥100 pixels), and their area were quantified. The graph shows the mean ± s.d. *n* = 4–5 FoV per condition were analysed from a representative experiment. **P < 0.01 (*P* = 0.0079), one-tailed Mann–Whitney *U* test. **j,** The UG-rich RNA leads to TDP-43 higher-order assembly in vitro, as demonstrated by fractionation and western blot. TDP-43 samples were prepared as in **i**, followed by fractionation into supernatant (S, soluble protein and small clusters) and pellet (P, large clusters pelleted by centrifugation) and western blot for TDP-43, immediately after sample mixing (0 min) and after a 10-min incubation. The graph shows the mean ± s.e.m. *N* = 4, *P < 0.05 (*P* = 0.0143), one-tailed Mann–Whitney *U* test. Image in **a** created in BioRender; Shelkovnikova, T. https://biorender.com/c4vzzyl (2025).

majority of TDP-43-OE cells (Fig. 3a,b). TDP-43 was detectable in the restored clusters (Fig. 3c), indicating that the rescue was not due to TDP-43 displacement from paraspeckles by FUS. The FUS coexpression also partially rescued reduced NEAT1_2 semi-extractability in TDP-43-OE cells (Fig. 3d). The majority of NEAT1-positive clusters restored by FUS were composed of spheroids with intact core–shell structure ($P < 0.0001$, Fig. 3e). These results were corroborated by coexpressing FUS–GFP with TDP-43 SNAP-tag, which led to a

similar level of paraspeckle rescue (Extended Data Fig. 5e). Proximity ligation assay (PLA) and co-immunoprecipitation (co-IP) demonstrated detectable but limited interaction between TDP-43 and FUS in the nucleus (~20 interactions per cell by PLA) (Extended Data Fig. 6a,b). Using co-IP and PLA, we found that oligomerization-deficient TDP-43 mutants, which do not cause paraspeckle dispersal (Fig. 2), retain their ability to interact with FUS (Extended Data Fig. 6b,c). Therefore, the negative effect of increased TDP-43 dosage

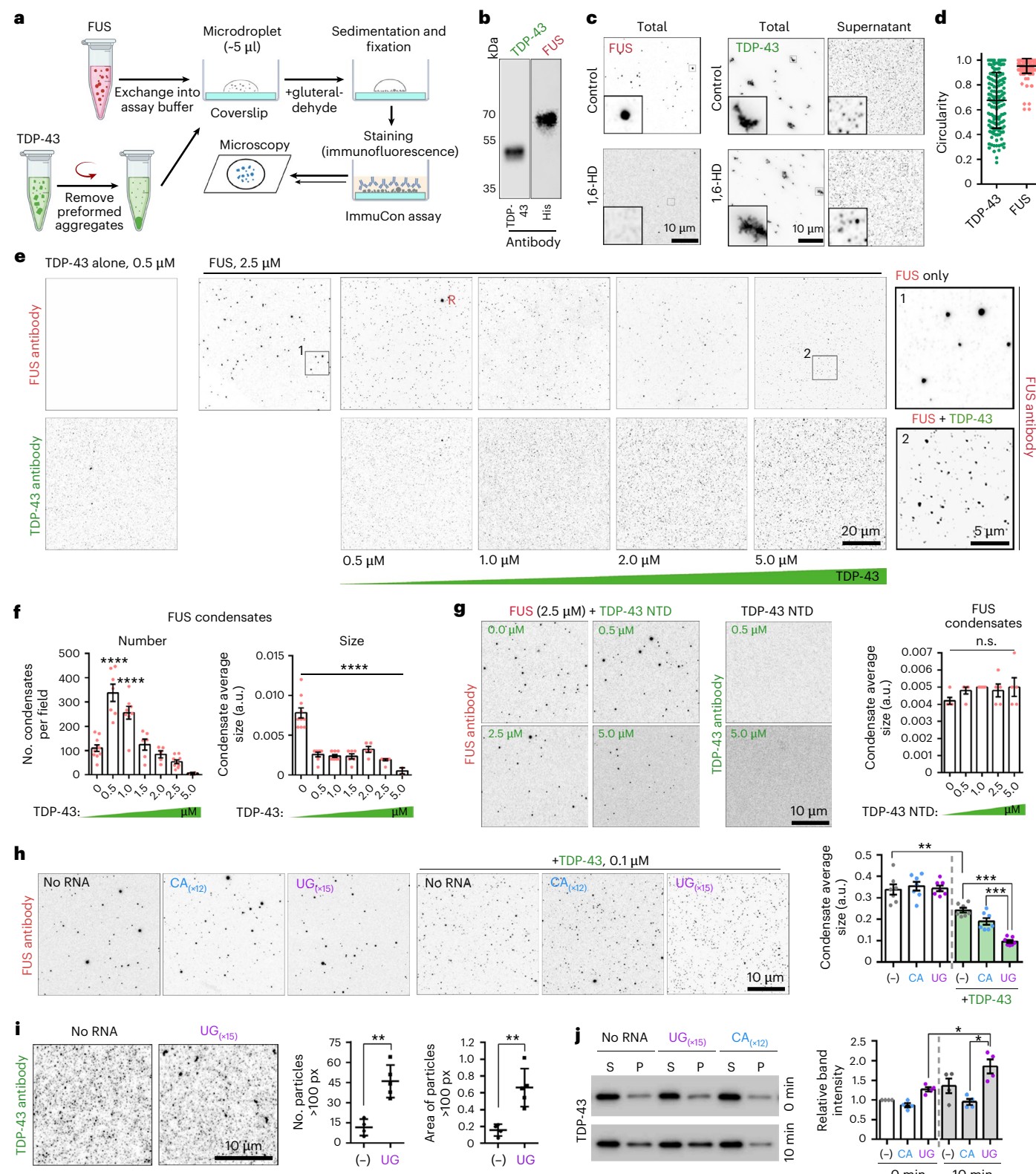

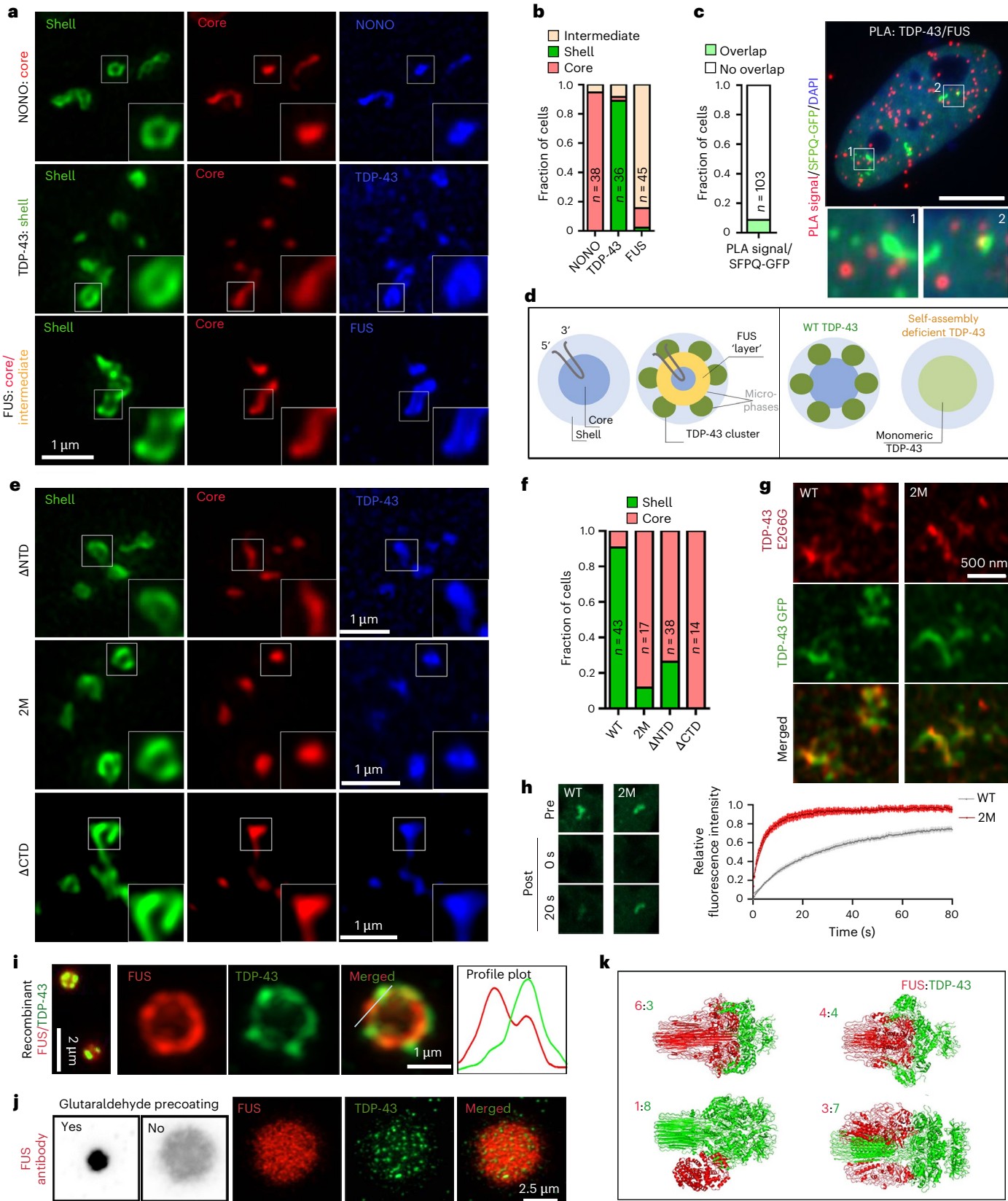

on paraspeckle condensation is not mediated by its excessive interaction with FUS molecules. Finally, the double TDP-43/FUS knockdown or knockdown of TDP-43 in FUS-knockout (KO) cells could not rescue spheroid assembly (data not shown), in line with the pivotal role of FUS in precursor condensation.

We next examined the possible redundancy of core PSPs in facilitating the paraspeckle condensation. Supplementing TDP-43-OE cells with exogenous SFPQ and NONO indeed could also rescue paraspeckle condensation, with both paraspeckle clusters and spheroids restored (Fig. 3f,g and Extended Data Fig. 6d). Consistently, ectopic

**Fig. 5 | Molecular basis for TDP-43 and FUS microphase separation within paraspeckles. a,b,** The GFP-tagged PSPs display the expected localization within spheroids; representative images (**a**) and the quantification (**b**) (from a representative experiment) are shown. For TDP-43, low-expressing cells with intact paraspeckles were analysed. The number of cells analysed (*n*) is indicated within bars. See Extended Data Fig. 8a for more details. **c,** TDP-43 and FUS rarely interact in paraspeckles as shown by PLA analysis of SFPQ-labelled paraspeckles. Representative images and the quantification (from a representative experiment) are shown. The number of cells analysed (*n*) is indicated. **d,** A model for the subspheroid localization of FUS and normal and oligomerization-deficient TDP-43. **e,f,** The oligomerization-deficient TDP-43 mutants localize to the paraspeckle core/intermediate layer: representative images (**e**) and the quantification (**f**) (from a representative experiment) are shown. The number of

cells analysed (*n*) is indicated within bars. **g,** The monomer-specific anti-TDP-43 antibody (E2G6G) recognizes TDP-43[2M] in the core but not TDP-43[WT] clusters in the shell. Low-expressing cells with intact paraspeckles were analysed. Representative images are shown. **h,** TDP-43[2M] displays a significantly higher mobility in paraspeckles than WT protein. FRAP was performed on paraspeckle clusters in TDP-43 GFP-expressing cells. Data from a representative experiment are shown (*n* = 10 cells, two paraspeckle clusters in each). **i,j,** The recombinant TDP-43 and FUS proteins form microphases within residual FUS condensates in FUS/TDP-43 samples in vitro: conventional imaging (left) and confocal super-resolution (single optical section) (right) are shown (**i**), alongside the detection of TDP-43 clusters within partially disassembled FUS condensates formed on coverslips without fixative precoating (**j**). Representative images are shown. **k,** The AlphaFold 3 predictions for FUS-TDP-43 oligomers at different protein ratios.

---

expression of FUS, SFPQ or NONO in a FUS-KO cell line[12] led to a significant rescue of paraspeckle assembly (Extended Data Fig. 6e–i). Strikingly, the expression of a chimeric protein in which the FUS low-complexity domain (LCD) was replaced with a disordered, prion-like domain from a yeast protein Sup35[33,34] also resulted in paraspeckle rescue; this protein localized to the spheroid core (Extended Data Fig. 6e–i). These data indicate that paraspeckle condensation relies on the core PSP LCD dosage rather than a specific sequence.

## TDP-43 limits FUS condensation in vitro

To corroborate cellular findings in vitro, we used recombinant FUS for modelling paraspeckle condensates. We developed an immunostaining assay for condensate analysis in vitro ('ImmuCon') (Fig. 4a), using recombinant TDP-43 and FUS—commercial or purified in-house (Fig. 4b and Extended Data Fig. 7a,b). At concentrations ≥1.25 μM, FUS readily formed round, droplet-like 0.1–1 μm structures, in physiological salt and without crowding agents, after 10–30 min of incubation (Fig. 4c and Extended Data Fig. 7c). These droplets had a high circularity index, were sensitive to 1,6-hexanediol (1,6-HD) and high salt and displayed re-entrant phase behaviour (Fig. 4c,d and Extended Data Fig. 7d,e), indicative of their LLPS nature. By contrast, the recombinant TDP-43 formed large filamentous aggregates that could be removed by low-speed centrifugation, leaving soluble protein and small, <100 nm clusters in the supernatant (Fig. 4c). TDP-43 clusters and aggregates were insensitive to 1,6-HD and molecular chaperones *bis*-ANS[35] and trimethylamine *N*-oxide (TMAO)[36] known to modulate TDP-43 LLPS (Fig. 4c and Extended Data Fig. 7f), indicating that they correspond to non-LLPS structures.

For the co-incubation experiments, TDP-43 supernatant after aggregate removal was added at an increasing concentration to FUS samples (final FUS concentration of 2.5 μM). FUS is ~2.5 times more abundant than TDP-43 in the nucleus (7.6 μM versus 2.9 μM)[37]; therefore, we tested a range of FUS:TDP-43 ratios—between 5:1 and 1:2. Even the lowest TDP-43 concentration used (0.5 μM; 5:1) disrupted

FUS condensates, virtually eliminating large FUS droplets—accompanied by an increase in droplet numbers (Fig. 4e,f). At the highest TDP-43 concentration (5 μM), only small (<100 nm) FUS condensates remained, which were often below the detection threshold, leading to reduced condensate count (Fig. 4e,f). Consistently, in an orthogonal assay for measuring condensate formation and stability, Taylor dispersion-induced phase separation (TDIPS)[38], FUS readily underwent condensation when exchanged into the LLPS assay buffer, whereas TDP-43 co-injection caused a burst of monomerization (Extended Data Fig. 7g). Finally, the FUS condensate size was significantly reduced in the presence of TDP-43 in a different phase separation assay—using proteins labelled with cyanine dyes via a small ybbR peptide tag (Extended Data Fig. 7h). By contrast, the recombinant TDP-43 NTD (a.a. 1–98; Extended Data Fig. 7b), which was completely diffuse in ImmuCon, had no effect on FUS condensates at concentrations of up to 5 μM (Fig. 4g). Thus, TDP-43 prevents FUS condensate fusion and hence growth, which requires its ability to polymerize.

The TDP-43 RNA-binding capacity is required for paraspeckle dispersal in cells (Fig. 2). NEAT1_2 contains several UG repeats—the high-affinity binding sites for TDP-43[39,40]. Using synthetic RNA oligonucleotides and TDP-43 at low concentration (0.1 μM) for more efficient detection of possible additive effect of RNA, we found that the addition of UG$_{(×15)}$, but not control CA$_{(×12)}$ RNA, potentiates FUS condensate disruption by TDP-43 (Fig. 4h). Furthermore, the UG-rich RNA was found to enhance TDP-43 clustering (Fig. 4i,j), consistent with its role in promoting TDP-43 phase separation[41].

## TDP-43 and FUS form demixed microphases within condensates including paraspeckles

Below disruptive concentrations, TDP-43 can be 'accepted' into paraspeckle condensates—forming TDP-43-containing spheroids, where TDP-43 and FUS localize to the paraspeckle shell and core, respectively[18]. However, the molecular basis of this spatial segregation is unknown. We first confirmed that GFP-tagged PSPs can be used as a

---

**Fig. 6 | TDP-43 sequestration into de novo nuclear condensates (TCs) alleviates paraspeckle suppression during stress. a–c,** The restored biogenesis of paraspeckle spheroids upon TC assembly in TDP-43-OE cells: representative conventional microscopy images for stress recovery stages (**a**), the quantification of cells with NEAT1-positive foci (**b**) and SRM images confirming the intact core–shell structure of de novo assembled spheroids (arrows) (**c**) are shown. The TC formation in TDP-43 GFP-expressing cells was induced by NaAsO$_2$, and cells with high TDP-43 OE were analysed. Time stamps in **a** are times after NaAsO$_2$ wash-off. In **b**, graph shows mean ± s.d. 45 and 49 cells (*n* = 10 fields of view (FoV) per condition) were analysed for basal and stage 2/3 conditions, respectively, from a representative experiment. ****P* < 0.0001, one-tailed Mann–Whitney *U* test. **d,** The transcriptional inhibition prevents paraspeckle spheroid restoration during the recovery from NaAsO$_2$ stress. The transcriptional inhibitor DRB was added after the NaAsO$_2$ wash-off, and cells were analysed after 1 h of recovery. In total, 69 and 61 cells (*n* = 15 FoV per condition) were analysed for dimethyl sulfoxide (DMSO) and DRB conditions, respectively, from a representative experiment.

The graph shows the mean ± s.d. ****P* < 0.0001, one-tailed Mann–Whitney *U* test. **e,f,** The depletion of endogenous TDP-43 from paraspeckles during TC assembly: representative images and profile plots for TCs/paraspeckles (**e**) and the quantification of overlap (**f**) are shown. *n* = 20 cells, with at least 2–3 TCs in each, were analysed per condition, from a representative experiment. ****P* < 0.0001, one-way ANOVA with Dunnett's post hoc test. **g,** The NEAT1_2 semi-extractability is decreased in TDP-43-depleted cells. Relative NEAT1_2 levels in heated and non-heated samples are shown alongside the ratio. Graphs show mean ± s.e.m. *N* = 4 or 6, **P* < 0.05 (*P* = 0.0143), ***P* < 0.01 (*P* = 0.0011), one-tailed Mann–Whitney *U* test. Left: the TDP-43 knockdown was confirmed by RT–qPCR in the same set of samples. **h,** The TDP-43 downregulation results in reduced paraspeckle clustering. In total, 148 and 123 cells (from *n* = 5 FoV per condition) were analysed for scrambled and TDP-43 siRNA, respectively, from a representative experiment. The graph shows the mean ± s.e.m. ***P* < 0.01 (*P* = 0.004), one-tailed Mann–Whitney *U* test. MG132-treated condition (4-h exposure) characterized by augmented paraspeckle clustering is shown for comparison. Scale bars, 10 μm.

tool in spheroid substructure studies, by confirming the localization of GFP-tagged SFPQ and NONO to the core and TDP-43 to the shell (Fig. 5a,b and Extended Data Fig. 8a). The TDP-43 signal appeared 'interrupted', forming clusters in the shell, consistent with a previous report[18]. Notably, FUS's signal appeared 'wider' than the NEAT1 core and dimmer in the centre (Fig. 5a,b), forming an 'intermediate layer'. Two other shell-localized proteins, RBM14 and hnRNP F, formed clusters and patches, respectively; however, they did not overlap with TDP-43 clusters in the shell (Extended Data Fig. 8b,c). Nor did these clusters contain other proteins previously detected in TDP-43-enriched nuclear

condensates[19] (Extended Data Fig. 8d). PLA showed that TDP-43–FUS interactions within paraspeckles are extremely rare (Fig. 5c), supporting that, within spheroids, these proteins are confined to separate, tightly packed microphases (Fig. 5d).

Surprisingly, unlike TDP-43[WT], its oligomerization-deficient variants largely localized to the spheroid core (Fig. 5e,f). A 'monomeric' TDP-43 antibody (E2G6G) that predominately recognizes non-NTD oligomerized protein[42] readily detected the core-localized TDP-43[2M] but did not recognize TDP-43[WT] clusters in the shell (Fig. 5g). Therefore, these TDP-43 clusters are composed of TDP-43 oligomers assembled

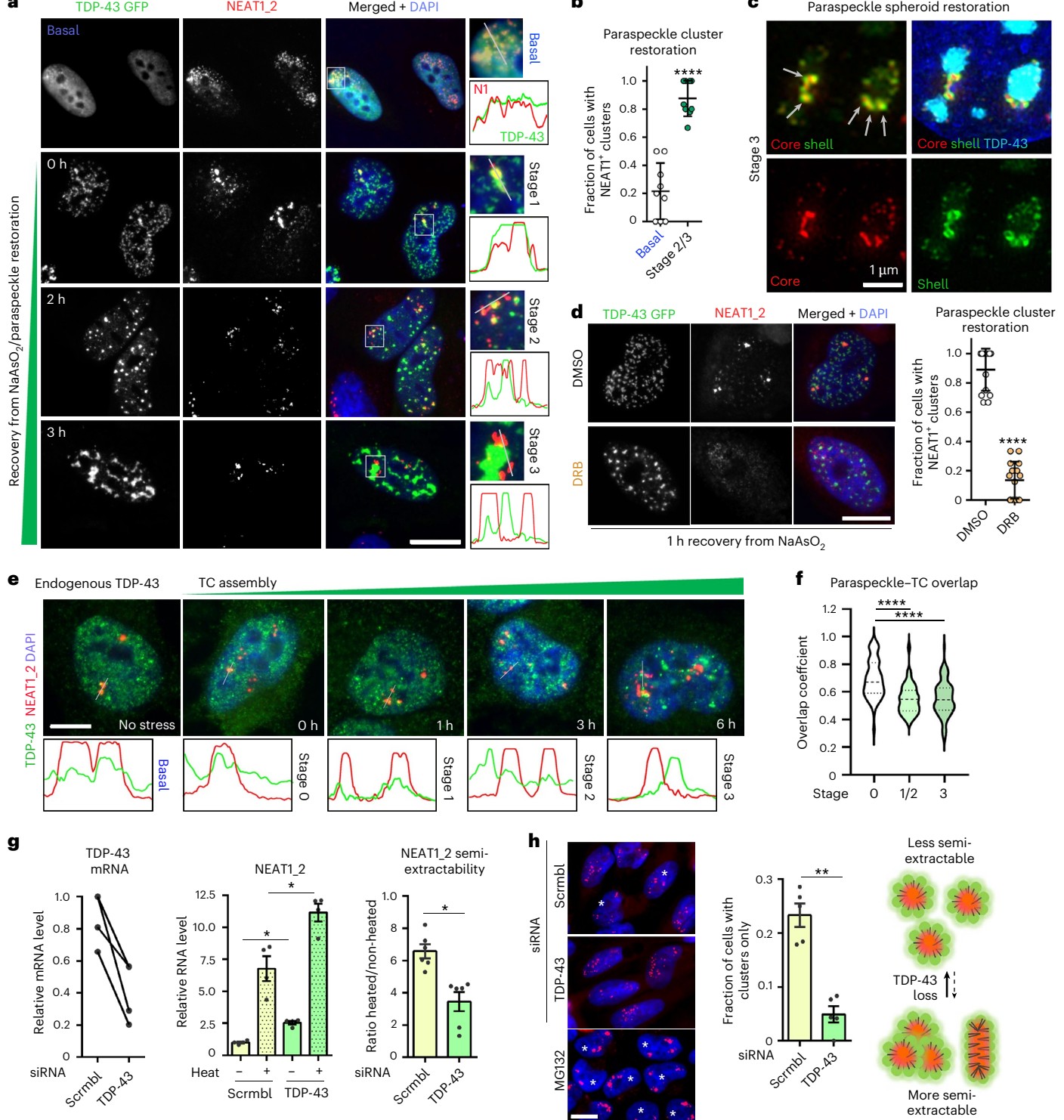

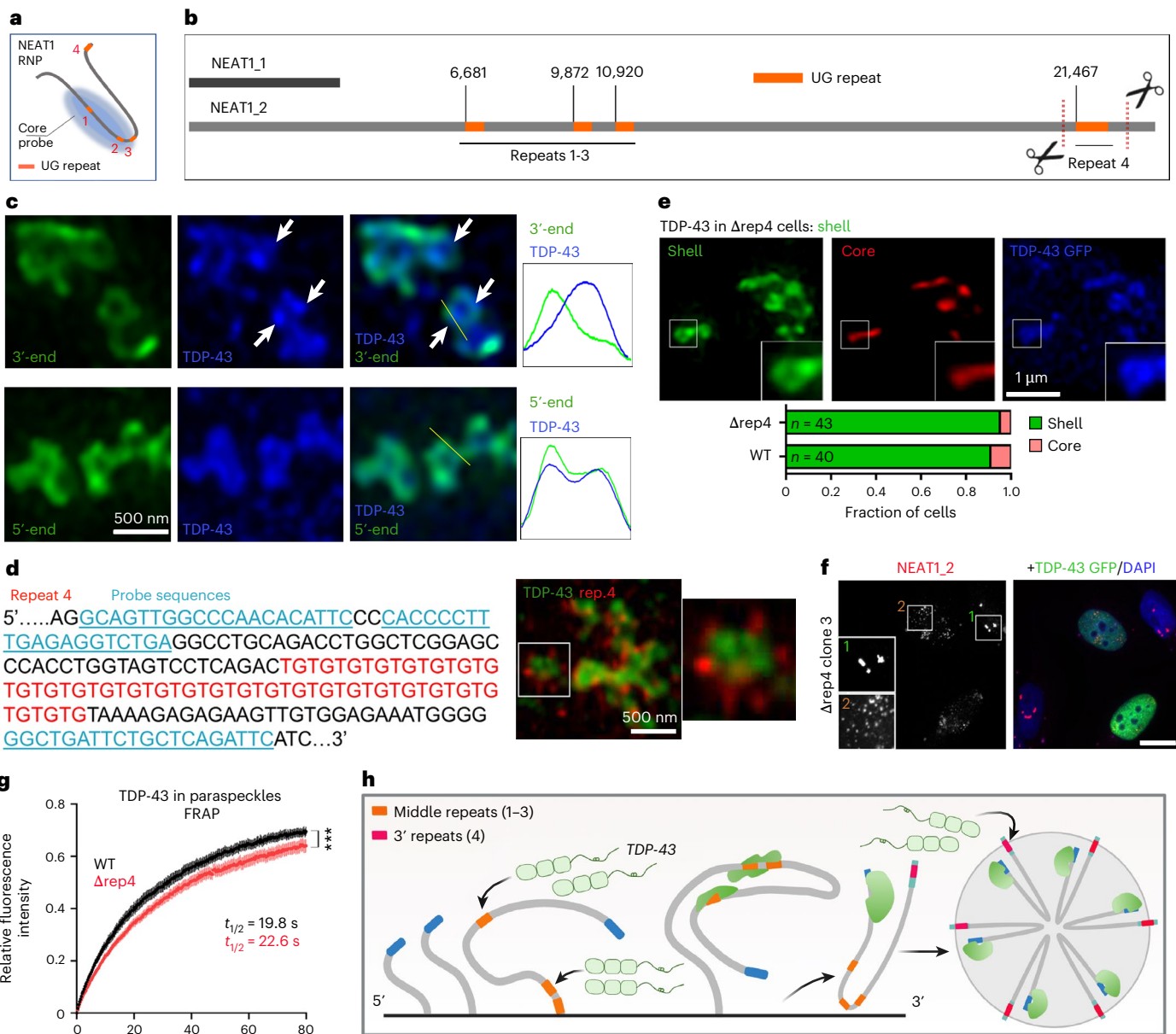

**Fig. 7 | Differential role of the NEAT1_2 middle and 3′-end UG repeats in paraspeckle regulation. a**, The predicted NEAT1_2 UG-repeat positions upon its folding within NEAT1 RNP. **b**, The NEAT1_2 UG-repeat positions in the linear RNA and positions of CRISPR sites used to generate cell lines with the deletion of the 3′-end NEAT1_2 UG repeat (Δrep4 cells). **c**, The TDP-43 clusters overlap with the 5′-end of NEAT1_2 within the paraspeckle shell. Top: arrows indicate TDP-43 clusters not overlapping with the 3′-end NEAT1_2 probe signal. Representative images are shown. **d**, The NEAT1 RNA-FISH with a probe set flanking repeat 4 (left) confirms its localization to the shell, external to TDP-43 clusters (right). A representative image is shown. **e**, The normal shell localization of TDP-43 in

Δrep4 cells. Representative image and quantification are shown. The number of cells analysed (*n*) is indicated within bars. **f**, The TDP-43 GFP OE disperses spheroids in Δrep4 cells. A representative image is shown. Scale bar, 10 µm. **g**, The attenuated TDP-43 GFP recruitment into paraspeckle clusters in Δrep4 cells, as revealed by FRAP. *n* = 18 cells with two clusters in each were analysed per condition, per repeat (data from clones 2 and 3 combined), from a representative experiment. ***P < 0.001 (P = 0.0006), two-tailed unpaired *t*-test. **h**, A model for co- and posttranscriptional TDP-43 recruitment onto NEAT1_2 and into paraspeckle spheroids. Image in **h** created in BioRender; Shelkovnikova, T. https://biorender.com/c4vzzyl (2025).

via NTD–NTD interactions. Using FRAP, we found that in contrast to TDP-43$^{WT}$, TDP-43$^{2M}$ is highly mobile within the spheroids ($t_{1/2}$ = 18.5 s, versus 4.0 s for TDP-43$^{WT}$; Fig. 5h). Thus, TDP-43 oligomerization is essential for its shell localization, where it exists in the form of low-dynamics, non-liquid microcompartments (Fig. 5d).

Strikingly, the TDP-43/FUS microphase arrangement in paraspeckle spheroids was recapitulated in vitro. Residual FUS condensates found in the TDP-43/FUS samples (Fig. 4e, R) also stained positive for TDP-43, and high-resolution imaging demonstrated that TDP-43 clusters are embedded in the FUS condensate surface (Fig. 5i).

In partially disassembled FUS condensates, when the fixative pre-coating step was omitted, multiple TDP-43 clusters were detected inside the condensate (Fig. 5j). Consistently, TDP-43/FUS immiscibility could be also observed using ybbR-Cy3/5-labelled proteins (Extended Data Fig. 8e).

The co-IP analysis of FUS variants ectopically expressed in FUS-KO cells at a near-endogenous level (complementation) showed that full-length FUS more efficiently interacts with TDP-43 compared with FUS LCD or FUS RNA-binding domains in isolation (Extended Data Fig. 8f,g). The coarse-grained molecular dynamics

simulations showed that TDP-43 LCD (a.a. 274–414) and FUS LCD (a.a. 1–167) are fully miscible and form a homogeneous condensate (Supplementary Fig.1)—unlike the two-phase arrangement we observed in vitro and in cells. Thus, the TDP-43-FUS two-phase arrangement within condensates is a product of multidomain interactions. Our in vitro data were supported by AlphaFold 3 modelling. At the 1:1 FUS:TDP-43 ratio (four molecules each) or physiological ratios (for example, 6:3)[37], full-length proteins were found to form distinct homo-oligomers that maintained a surface contact (Fig. 5k). However, increasing the TDP-43 concentration (for example 3:7) led to FUS oligomer disruption (Fig. 5k).

Collectively, data shown in Figs. 4 and 5 suggest that, depending on its concentration, TDP-43 either joins FUS condensates as non-liquid clusters, which can lead to their emulsification, both of which require intact TDP-43 polymerization and multidomain interactions with FUS.

### TDP-43 sequestration into stress-induced de novo condensates alleviates paraspeckle suppression

Paraspeckles are upregulated during stress, which was proposed to be cytoprotective[6,7,10,43]. Cellular stresses also induce de novo TDP-43 nuclear condensate (TC) formation—leading to TDP-43 inactivation under stress[44,45]. We hypothesized that the TDP-43 sequestration into TCs alleviates the paraspeckle suppression. First, paraspeckles and TCs were tracked in the TDP-43 GFP OE cells during the recovery from NaAsO$_2$ (arsenite). Cells were grouped according to the TC assembly stage: 1—fine nucleoplasmic granulation, 2—small/medium granules and 3—large (mature) TCs. TDP-43 GFP sequestration into TCs was accompanied by restoration of bright NEAT1 foci, which were found to consist of intact spheroids (Fig. 6a–c). This restoration could be prevented by transcription inhibition (Fig. 6d), indicating de novo (cotranscriptional) assembly of paraspeckles in TC-forming cells, rather than the coalescence of pre-existing NEAT1 RNPs. TDP-43 and NEAT1 signals progressively separated during stress, both for overexpressed (Fig. 6a) and endogenous (Fig. 6e,f) protein.

TCs were reported to contain NEAT1[44], and we also observed some NEAT1_2 enrichment within TCs in the above experiments. To establish the role of NEAT1 in TC formation, we generated NEAT1-KO HeLa cells (Supplementary Fig. 2a,b) and analysed TC assembly during the recovery from NaAsO$_2$. NEAT1-deficient cells were able to form TCs, both from ectopically expressed and endogenous TDP-43 (Supplementary Fig. 2c,d). However, the longitudinal imaging revealed altered TC dynamics in NEAT1-deficient cells, with faster TC disassembly (Supplementary Fig. 2e). Furthermore, the FRAP showed higher TDP-43 mobility in TCs upon transient or constitutive NEAT1 depletion (Supplementary Fig. 2f and not shown). Therefore, NEAT1 is not required for TC assembly, but it modulates their properties.

We next compared the properties of TDP-43-depleted and TDP-43-containing paraspeckles, using small interfering RNA (siRNA) knockdown in HeLa cells. The FRAP analysis of SFPQ-labelled paraspeckles demonstrated faster fluorescence recovery in TDP-43-depleted paraspeckles (Extended Data Fig. 9a). We next studied the NEAT1_2 semi-extractability upon TDP-43 knockdown. First, we found that, similar to Trizol[22], commercial RNA isolation kits also require a heating step to efficiently isolate NEAT1_2 and that NEAT1_2 yield is sensitive to DNase treatment (consistent with its chromatin association[46]), sample processing (freeze–thaw cycles) and sample density and concentration (sample:buffer ratio) (Supplementary Fig. 3). The DNase treatment increased NEAT1_2 yield in TDP-43-sufficient but not in TDP-43-deficient cells, potentially pointing to the reduced association of TDP-43-depleted paraspeckles with chromatin (Extended Data Fig. 9b). Unexpectedly, TDP-43 knockdown resulted in reduced NEAT1_2 semi-extractability despite NEAT1_2 upregulation (Fig. 6g). TDP-43 depletion does not affect spheroid assembly (Fig. 2); however, we found that it significantly limits spheroid clustering, which may be responsible for the effect on NEAT1_2 semi-extractability (Fig. 6h).

By reducing spheroid clustering, TDP-43 may increase their intranuclear mobility. To address possible differences between paraspeckles distal and proximal to the *NEAT1* locus, we generated a HeLa cell line with a single *NEAT1* allele labelled by MS2 tagging. The 6xMS2 repeats were inserted in a safe harbour region on the 3′-end of NEAT1_2[17] (Extended Data Fig. 9c). NEAT1_2-MS2 cells were transfected to express nuclear-localized MCP-GFP. Between two and four, and most commonly three, NEAT1-positive clusters were detected in hypertriploid HeLa cells, and a single cluster was invariably GFP-positive in transfected cells (Extended Data Fig. 9c). Strikingly, the 'distal' paraspeckles were found to lose GFP signal: 92.9 ± 4.4% of proximal but only 14.6 ± 6.8% of distal paraspeckles were GFP-positive (Extended Data Fig. 9c). This suggested a remodelling event that alters either MCP-GFP incorporation or its folding within the spheroid and hence fluorescence. Therefore, *NEAT1* locus-distal paraspeckles, which are prevalent in TDP-43 depleted cells, may be subject to structural remodelling.

### Position-dependent contribution of NEAT1_2 UG repeats in paraspeckle regulation via TDP-43

NEAT1_2 UG repeats are prerequisite for TDP-43 recruitment into the paraspeckle spheroid, where their genomic deletion abolishes paraspeckle localization of TDP-43, but can be rescued by re-introduction of 60 repeats at NEAT1_2 3′-end[4]. NEAT1_2 has four UG repeats (≥8 units), three of which are located in the middle (repeats 1–3; 6,681–10,920:19 interrupted, 8 pure and 9 pure repeats) and one—in the 3′-end (repeat 4; 29 UG units; GRCh38) (Fig. 7a,b). The rest of NEAT1_2 is relatively depleted of UG stretches of ≥6 units. The sequencing of repeat 4 in HeLa

**Fig. 8 | NEAT1_2 3′-end UG repeat links paraspeckle stability under stress to neuronal survival and ALS. a**, Paraspeckles are more stable in cells lacking NEAT1_2 UG-repeat 4. Cells were analysed after 1 or 2 h of transcriptional block with DRB. Representative images and the quantification are shown for WT HeLa cells and two Δrep4 clones. The arrow indicates a residual paraspeckle cluster. In total, 99–128 cells from n = 10 fields of view (FoV) were analysed per condition, from a representative experiment. Box, 25th to 75th percentiles; centre line, median; whiskers, mean to max. *P < 0.05 (P = 0.0334), **P < 0.01 (P = 0.0038), one-day ANOVA with Holm–Sidak post hoc test. Scale bar, 20 μm. **b**, The enhanced stress-induced paraspeckle assembly in cells lacking NEAT1_2 UG-repeat 4. Cells were analysed after 6 h into the recovery from NaAsO$_2$ stress. In total, 85–170 cells from n = 10 FoV were analysed per condition, from a representative experiment. *P < 0.05 (P = 0.0101), **P < 0.01 (P = 0.0073), ***P < 0.001 (P = 0.0005), Kruskal–Wallis with Dunn's post hoc test. Box, 25th to 75th percentiles; centre line, median; whiskers, mean to max. **c**, The motor neurons with NEAT1_2 UG-repeat 4 deletion used in the study. The homozygous repeat 4 deletion was confirmed by RNA-FISH using stressed neurons (6 h recovery

from NaAsO$_2$). Representative images are shown. Scale bars, 10 μm. See Extended Data Fig. 10 for cell line generation and characterization. **d,e**, The hMNs with NEAT1_2 UG-repeat 4 deletion are protected from stress-induced apoptosis, whereas hMNs with ablated paraspeckles are more sensitive to stress: neurons were subjected to a low-dose, continuous stress (250 nM proteasome inhibitor MG132 added at 0 h), which induces paraspeckle assembly (**d**) and analysed on an Incucyte live imager using a caspase 3/7 dye (**e**). Data from a representative experiment are shown (n = 4 wells). The graph shows the mean ± s.e.m. ****P < 0.0001, one-way ANOVA with Tukey post hoc test (area under the curve comparisons). **f**, The NEAT1_2 UG-repeats size distribution in the human population from the analysis of WGS data of 4,996 patients with ALS and 1,743 controls. The repeat size was determined using ExpansionHunter. **g**, Longer lengths of NEAT1_2 UG-repeat 4 are associated with shorter survival in ALS. **h**, A model: the regulation of paraspeckles by TDP-43 via polymerization and position-dependent binding of NEAT1_2 UG repeats in steady state, under stress and in ALS. Image in **h** created in BioRender; Shelkovnikova, T. https://biorender.com/c4vzzyl (2025).

and SH-SY5Y cells confirmed its large size (20–28 repeats). Surprisingly, RNA-FISH and SRM demonstrated that shell-localized TDP-43 clusters overlap with the 5′- but not the 3′-end NEAT1 probe signal (Fig. 7c). By RNA-FISH with a probe flanking repeat 4 (43 nt/29 nt up/downstream), this repeat was found to localize externally to TDP-43 clusters (Fig. 7d).

We generated HeLa clones with CRISPR–Cas9-mediated repeat 4 deletion ('Δrep4') (Extended Data Fig. 10a,b). Paraspeckles formed normally in these clones, with TDP-43 localized to the shell, and were sensitive to TDP-43 OE (Fig. 7e,f and Extended Data Fig. 10c). Interestingly, NEAT1_2 was upregulated in Δrep4 clones (Extended Data Fig. 10d).

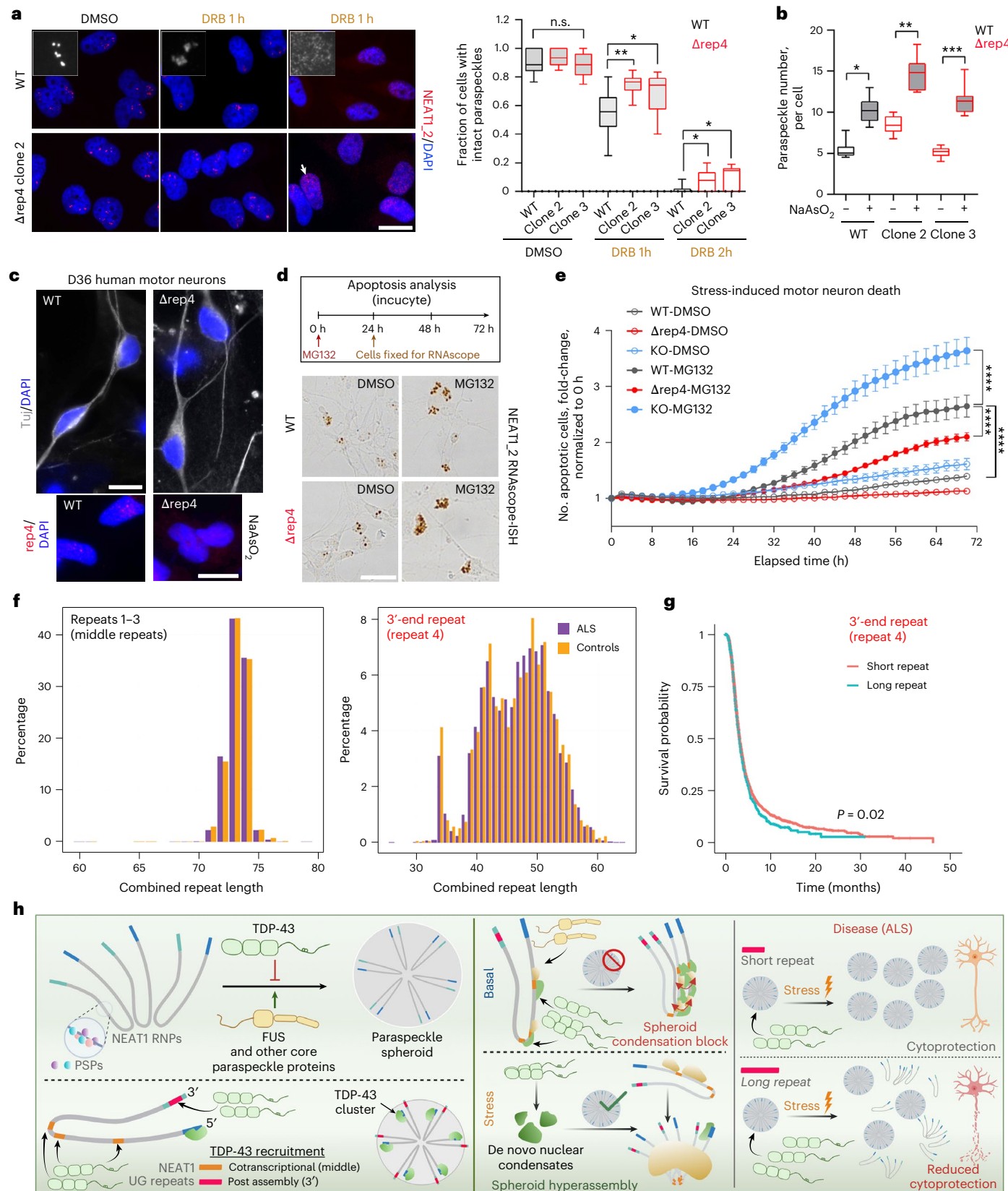

We hypothesized that repeat 4, being in a more 'accessible' position (spheroid surface), plays a role in TDP-43 recruitment into the spheroid after its assembly. Indeed, FRAP demonstrated a slower recovery of TDP-43 GFP fluorescence in Δrep4 cells compared with wild-type (WT) cells (Fig. 7g). Therefore, TDP-43 binding to NEAT1_2 middle/core UG repeats (repeats 1–3) during cotranscriptional RNP assembly mediates both its spheroid localization and condensation regulation, whereas repeat 4, which is synthesized later, is not essential for these processes but may mediate TDP-43 recruitment into the spheroids post assembly (Fig. 7h).

### NEAT1_2 3′-end UG repeat links paraspeckle stability to neuroprotection under stress and neurodegeneration

We hypothesized that decreased TDP-43 recruitment post assembly will influence paraspeckle stability and turnover. To test this, we inhibited the transcription with DRB to abolish the de novo paraspeckle formation. The spheroid integrity rapidly declined after DRB addition; however, the spheroid collapse was significantly slower in Δrep4 cells (Fig. 8a). An analysis of paraspeckle numbers during the stress response (6 h of recovery from NaAsO$_2$, when multiple individual paraspeckles accumulate)[7] revealed more efficient stress-induced upregulation of paraspeckles in Δrep4 cells (Fig. 8b).

Paraspeckles and their protein components are dysregulated in ALS[9]. We therefore studied the effect of NEAT1_2 repeat 4 deletion in hMNs. The widely used iPS cell line KOLF2.1J that possesses 24/25 UG repeats (Extended Data Fig. 10e) was utilized. Cell clones homozygous for repeat 4 depletion were generated (Extended Data Fig. 10f,g). In parallel, clones with ablated paraspeckles (by CRISPR–Cas9-mediated deletion of the NEAT1_2-specific portion of the *NEAT1* gene; NEAT1_2-KO cells) were also generated and characterized (Extended Data Fig. 10h–k). These engineered lines proliferated and differentiated normally, being indistinguishable from WT cells (Fig. 8c). Motor neuron cultures (day 36) obtained from WT, NEAT1_2 Δrep4 and NEAT1_2-KO iPS cells were subjected to chronic stress (continuous low-dose treatment with MG132) (Fig. 8d), and their survival was monitored on an Incucyte live cell imager for 72 h. Paraspeckle induction in WT and NEAT1_2 Δrep4 neurons by this stressor was confirmed using RNAscope–in situ hybridization (ISH) in parallel (Fig. 8d). This survival analysis demonstrated that NEAT1_2 Δrep4 neurons are significantly protected from stress-induced apoptosis, whereas NEAT1_2-KO neurons are more vulnerable compared with WT cells (Fig. 8e).

The above findings led us to hypothesize that repeat 4 size would have a disease-modifying effect in ALS. Using ExpansionHunter[47] and whole genome sequencing (WGS) data from ProjectMine (https://www.projectmine.com/, datafreeze 2), sizes of all four repeats were determined for 6,739 individuals, including 4,996 patients with ALS and 1,743 controls. The sizes of middle repeats (repeats 1–3) were similar to those in the reference sequence, with the modal length of 20 (range 7–25), 8 (6–12) and 9 (7–10) for repeats 1–3, respectively. The combined length of middle repeats (both alleles) was relatively invariable, with no difference in the size distribution between control and patient populations; no association with survival was detected (Fig. 8f and not shown). By contrast, repeat 4 displayed a wide range of sizes (2–37 repeats) in both human cohorts, with multimodal distribution of repeat length and two distinct maxima (Fig. 8f). We found that patients with ALS with the lengths of repeat 4 above the second maxima (*n* = 50) had a significantly shorter survival time than patients with ALS with a shorter repeat length, after adjusting for age, sex and the first ten principal components of genetic variation (difference: 0.27 years, Cox regression, *P* = 0.02) (Fig. 8g).

## Discussion

Here, we elucidate a central role for TDP-43 in controlling paraspeckle condensation—and hence the paraspeckle complex stoichiometry—underpinned by its ability to bind UG repeats and polymerize.

This regulation is encoded in the NEAT1_2's primary sequence, via the positions of UG repeats. We show that TDP-43's regulation of paraspeckle condensation is linked to cytoprotection under stress and to ALS progression (Fig. 8h). Overall, our study links emergent properties of a condensate to a biological function and disease pathogenesis.

Paraspeckle biogenesis is suppressed in the majority of healthy mammalian tissues, being substantial only in the gut epithelium and corpus luteum[3]. By contrast, the shorter NEAT1 isoform is expressed in the majority of tissues[3,48]. Therefore, the *NEAT1* promoter is constitutively active, requiring efficient control of the paraspeckle 'low-assembly' state. The NEAT1_2 production essential for paraspeckle assembly is regulated by the coordinated activity of polyadenylation factors hnRNPK, CPSF6, NUTD21[15] and TDP-43[4]. Here we demonstrate that TDP-43 is a key player in two additional paraspeckle regulation mechanisms downstream of NEAT1_2 synthesis. First, TDP-43 blocks the coalescence of NEAT1 RNPs into the paraspeckle condensate. Second, TDP-43 limits the dynamics of these condensates. Such a three-pronged mechanism (limited NEAT1_2 production[4], restricted paraspeckle condensation and maintenance of their mobility and dynamism—this study) ensures efficient paraspeckle suppression. Stress response, on the contrary, requires paraspeckle upregulation[6,7,43] and thus the alleviation of TDP-43's inhibitory activity. We show that this can be achieved by TDP-43 sequestration into de novo nuclear condensates. In development, when paraspeckle levels fluctuate[4,5,49], TDP-43's availability may be controlled by its different biophysical states and recruitment into other MLOs. We demonstrate that TDP-43's self assembly is crucial for paraspeckle regulation, similar to splicing[24,26]. Moreover, we find that increased TDP-43 dosage can elicit a TDP-43 loss-of-function effect, at least at the *NEAT1* locus, via enhanced TDP-43 oligomerization. Overall, we demonstrate that TDP-43's control of paraspeckle biogenesis is multilayered, extending beyond isoform switch, and is controlled by TDP-43 self-assembly.

Below certain concentrations, TDP-43 can be 'accepted' into FUS condensates; however, within those, it forms a distinct subcompartment. Consistently, recent studies have also found low miscibility between TDP-43 and FUS 'phases' in vitro and in cells[50,51]. Paraspeckle spheroids were proposed to be LLPS assemblies[17]; our data suggest that they are multiphase structures, composed of non-liquid (TDP-43 clusters) and LLPS (core PSP microphase) subcompartments. TDP-43 clusters within the spheroid probably modulate paraspeckle assembly by micellization[52] and paraspeckle autonomy[53]. TDP-43 may act as a modulator of paraspeckle micellization, where the TDP-43-rich phase growing on NEAT1_2 molecules physically interferes with the micelle assembly steps. Indeed, TDP-43 solid-like cluster formation would be expected to affect the coalescence of NEAT1-based block copolymers into a spheroid micellar structure[52]. Furthermore, it can change the physical properties of the micelle, such as its size, curvature and material properties. A recent in vitro and modelling study also supports that immiscible phase formation and competition between core PSPs and TDP-43 for NEAT1_2 binding underlie paraspeckle spheroid formation[51].

We find that TDP-43's regulation of paraspeckles is mediated by NEAT1_2 UG repeats in a position-dependent manner, via two different mechanisms—cotranscriptional and post assembly. The middle repeats recruit TDP-43 cotranscriptionally, promoting or even initiating its polymerization. This can either lead to TDP-43 cluster incorporation in the spheroid or block spheroid assembly, depending on TDP-43 dosage and oligomerization propensity (Fig. 8h). The exceptionally long 3′-end NEAT1_2 UG repeat, which emerges later during transcription and localizes in an accessible position on the paraspeckle surface, probably serves to recruit TDP-43 into the spheroid post assembly and modulate spheroid stability (Fig. 8h).

Paraspeckles were linked to cytoprotection[43], and here, we show that paraspeckle condensates specifically contribute to human neuron survival under stress. Given this and the reported links between TDP-43, NEAT1/paraspeckles and ALS pathology[9,10,54,55], we interrogated a

potential role for NEAT1_2 UG repeats in a large human genetic dataset (6,739 individuals). The invariability of the middle UG-repeat sizes supported their important role in paraspeckle regulation. Contrastingly, the 3′-end repeat size is highly variable (2–37 units), and its longer sizes correlate with faster ALS progression. We propose that longer UG repeats decrease paraspeckle condensate formation and stability under stress via enhanced TDP-43 recruitment, compromising cytoprotection and survival in the context of ALS (Fig. 8h). Several proteins important for paraspeckle assembly are affected by ALS mutations[9]. In particular, FUS mutations cause NEAT1_2 upregulation but impair paraspeckle condensation[22]. Recent studies suggest that TDP-43 loss of function is a primary disease mechanism, due to altered splicing[56]. TDP-43 sequestration onto longer 3′-end UG repeat as well as middle repeats exposed due to inefficient spheroid formation (for example elicited by ALS-linked mutant proteins) may exacerbate its loss of function and contribute to splicing dysregulation. ALS-causative TDP-43 mutations affecting its interactions with other PSPs[57] and oligomerization[26] may also act via paraspeckles. Recently, changes in TDP-43 monomer:oligomer ratios have been reported in ALS[46], and we demonstrate that TDP-43 homo-oligomerization capacity regulates the NEAT1 isoform ratio. Altered TDP-43 self-assembly states may affect paraspeckle formation in ALS via both NEAT1_2 production and condensation mechanisms. Importantly, our data suggest that NEAT1_2 upregulation per se may not be sufficient to elicit a cytoprotective effect—which is only achieved upon paraspeckle condensate formation. Efficient tools to modulate paraspeckle condensation will be required to fully prove this in future studies. Ultimately, the therapeutic targeting of paraspeckle condensation may provide a universal treatment option applicable to multiple ALS subtypes.

The term 'paraspeckle' typically refers to the foci positive for NEAT1_2 and a core PSP. Our study posits that paraspeckles exist as a spectrum of assemblies regulated by the stoichiometry of TDP-43 and core PSPs, which is cell type- and cell state-specific. Paraspeckle condensates sequester additional proteins, for example FUS, and through this probably acquire novel properties and functions. The impaired stoichiometry of paraspeckle complexes will have unwanted cellular effects leading to disease states. Our research creates a framework for the studies of differential and emergent functionalities of paraspeckle assemblies in various physiological and pathological contexts, including development, cancer and neurodegeneration.

## Online content

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

[1]Sheffield Institute for Translational Neuroscience and Neuroscience Institute, University of Sheffield, Sheffield, UK. [2]Cardiff University, Cardiff, UK. [3]Fida Biosystems ApS, Søborg, Denmark. [4]Zernike Institute for Advanced Materials, University of Groningen, Groningen, the Netherlands. [5]Molecular Systems Biology Unit, European Molecular Biology Laboratory, Heidelberg, Germany. [6]School of Life Sciences, University of Essex, Essex, UK. [7]University Medical Center Utrecht, Utrecht, the Netherlands. [8]The Pirbright Institute, Pirbright, Surrey, UK. [9]Centre for Molecular Medicine and Innovative Therapeutics, Murdoch University, Perth, Western Australia, Australia. [40]Present address: The Francis Crick Institute, London, UK. [41]Present address: Big Chemistry Foundation, Nijmegen, the Netherlands. [42]These authors contributed equally: Rachel E. Hodgson, Wan-Ping Huang, Ruaridh Lang. ✉e-mail: t.shelkovnikova@sheffield.ac.uk

## Project MinE ALS Sequencing Consortium

Philip van Damme[10], Philippe Corcia[11,12,13], Philippe Couratier[13,14], Patrick Vourc'h[15,16], Orla Hardiman[17,18], Russell McLaughlin[19], Marc Gotkine[20], Yossef Lerner[20], Shovman Yehuda[20], Vivian Drory[21], Nicola Ticozzi[22,23], Vincenzo Silani[22,23], Jan H. Veldink[24], Leonard H. van den Berg[24], Mamede de Carvalho[25,26], Teresa Salas[27], Jesus S. Mora Pardina[28], Monica Povedano[29], Peter Andersen[30], Markus Weber[31], Nazli A. Başak[32], Ammar Al-Chalabi[33,34], Chris Shaw[33], Pamela J. Shaw[1], Karen E. Morrison[35], John E. Landers[36], Jonathan D. Glass[37,38], Clifton L. Dalgard[39], Joke J. F. A. van Vugt[7] & Johnathan Cooper-Knock[1]

[10]KU Leuven-University of Leuven, Department of Neurosciences, Leuven, Belgium. [11]Centre SLA, CHRU de Tours, Tours, France. [12]UMR, iBrain, Université de Tours, Inserm, Tours, France. [13]Federation des Centres SLA Tours and Limoges, LITORALS, Tours, France. [14]Centre SLA CHU Dupuytren Limoges France, Tours, France. [15]Service de Biochimie et Biologie moléculaire, CHU de Tours, Tours, France. [16]UMR 1253, Université de Tours, Inserm, Tours, France. [17]Academic Unit of Neurology, Trinity College Dublin, Trinity Biomedical Sciences Institute, Dublin, Ireland. [18]Department of Neurology, Beaumont Hospital, Dublin, Ireland. [19]Complex Trait Genomics Laboratory, Smurfit Institute of Genetics, Trinity College Dublin, Dublin, Ireland. [20]Department of Neurology, Hadassah Medical Organization and Faculty of Medicine, Hebrew University of Jerusalem, Jerusalem, Israel. [21]Department of Neurology Tel-Aviv Sourasky Medical Centre, Tel Aviv-Yafo, Israel. [22]Department of Neurology and Laboratory of Neuroscience, IRCCS Istituto Auxologico Italiano, Milan, Italy. [23]Department of Pathophysiology and Tranplantation, 'Dino Ferrari' Center, Università degli Studi di Milano, Milan, Italy. [24]Department of Neurology, UMC Utrecht Brain Center, University Medical Center Utrecht, Utrecht University, Utrecht, the Netherlands. [25]Instituto de Fisiologia, Instituto de Medicina Molecular, Faculdade de Medicina, Universidade de Lisboa, Lisbon, Portugal. [26]Department of Neurosciences, Hospital de Santa Maria-CHLN, Lisbon, Portugal. [27]ALS Unit, Hospital Universitario La Paz-Carlos III, Madrid, Spain. [28]ALS Unit, Hospital San Rafael, Madrid, Spain. [29]la Unitat Funcional de Motoneurona, Cap de Secció de Neurofisiologia, Servei de Neurologia, Hospital Universitario de Bellvitge-IDIBELL, Barcelona, Spain. [30]Department of Clinical Science, Neurosciences, Umeå University, Umeå, Sweden. [31]Neuromuscular Diseases Unit/ALS Clinic, Kantonsspital St. Gallen, St. Gallen, Switzerland. [32]Koç University, School of Medicine, Molecular Biology and Genetics, Suna and Inan Kıraç Foundation, Istanbul, Turkey. [33]Maurice Wohl Clinical Neuroscience Institute, King's College London, Department of Basic and Clinical Neuroscience, London, UK. [34]Department of Neurology, King's College Hospital, London, UK. [35]School of Medicine, Dentistry and Biomedical Sciences, Queen's University, Belfast, UK. [36]Department of Neurology, University of Massachusetts Medical School, Worcester, MA, USA. [37]Department of Neurology, Emory University School of Medicine, Atlanta, GA, USA. [38]Emory ALS Center, Emory University School of Medicine, Atlanta, GA, USA. [39]The American Genome Center, Uniformed Services University—'America's Medical School', Bethesda, MD, USA.

## Methods

### Ethics statement

Research performed in this study complies with all relevant ethical guidelines.

### Plasmids

TDP-43, FUS, NONO and SFPQ expression plasmids from previously published studies were used[10,58]. Plasmids for the expression of all TDP-43 variants, RBM14, hnRNPF and BRG1 were custom-made by VectorBuilder or Genewiz/Azenta. Cry2olig-mCherry plasmid was from Addgene (no. 60032)[59], and TDP-43 ORF was subcloned into this vector using BsrGI/BsiWI and NotI sites. All proteins were tagged on their N-terminus. Expression plasmids generated in this study are available from Addgene[60].

### Cell line generation and NEAT1_2 UG-repeat analysis

HeLa and SH-SY5Y cell lines were obtained from ATCC via Merck (nos. 93021013 and 94030304). Genomically edited cell lines were generated by CRISPR–Cas9 as described earlier[12,61]. Guide RNAs (gRNAs) were designed using ChopChop (https://chopchop.cbu.uib.no/). The HeLa NEAT1-KO line was generated using double targeting with the following gRNAs: upstream, ACAGGGAGGGATGCGCGCCT and CTTGCATAGCTGAGCGAGCCCGG, and downstream, CCTTGTAAAG GCATAGCCAG and CAAAACCTGAGTGCGGCCAT. Single-cell derived clones were initially screened by PCR, and lack of NEAT1 signal in the positive clones was confirmed by RNA-FISH using 5′, 3′ and middle NEAT1 probes (Supplementary Table 1). For the Δrep4 cell line production, two gRNAs were used to generate a 372-bp deletion comprising this repeat: CTCCAAACACACTATGGTGC and GAGAGGACTGTAACCT GCTC. Single-cell derived clones with the 'edited' (~600 bp) but not WT (~1,000 bp) allele were identified by PCR and RT–qPCR. For the CRISPR–Cas9-mediated *TARDBP* inactivation, two gRNAs targeting exon 3 were used: ACATCCGATTTAATAGTGTT and TGGAGAAGTTCTTATGGTGC. The MS2-tagged HeLa cell line was generated by inserting 6xMS2 into pos.21076 of NEAT1_2 (gRNA: TGTGGGATAAGCGAGCTACG). The donor plasmid was constructed using pCR Blunt II-TOPO vector (Thermo Fisher) by amplifying the repeats from phage-cmv-cfp-24xms2 plasmid (Addgene plasmid no. 40651)[62] and adding 25-bp microhomology arms and a Cas9 target site on both ends to facilitate the linearization of the donor template in the nucleus; a point mutation was introduced to destroy the PAM sequence. Clones were screened by PCR after cotransfection of the gRNA-containing and donor plasmids. The pMS2-GFP plasmid was from Addgene (no. 27121)[63]. For NEAT1_2 repeat 4 length analysis, the repeat-containing region was PCR-amplified from genomic DNA using flanking primers, cloned into the pCR Blunt II-TOPO vector, and ≥4 colonies were sequenced. All PCR primers are given in Supplementary Table 1.

### Cell transfection, treatments and optogenetic experiments

For the TDP-43 and NEAT1 knockdown, SilencerSelect siRNAs (Thermo Fisher) were used (nos. s23829 and n272456, respectively), and an endoribonuclease-prepared siRNA (esiRNA) (no. EHU092111, Sigma) was used for FUS, alongside scrambled negative control AllStars siRNA from Qiagen. Transfection with siRNA or plasmids was performed using either Lipofectamine 2000 (Thermo Fisher) or jetPRIME (Jena Bioscience) following manufacturer's instructions, in a 24- or 12-well format. Cells were treated with 0.5 mM NaAsO$_2$ (Sigma) for 1 h and left to recover in fresh media. For optoTDP-43 experiments, HeLa cells were transfected to express optoTDP-43 or Cry2olig-mCherry and stimulated with blue light 24 h post transfection, on a custom array or on Opera Phenix[64]. Cells were protected from light between experiments.

### Stem cell and neuronal models

KOLF2.1J iPS cell line (iNDI/JAX, no. JIPSC001000) was cultured in mTeSR Plus and passaged using ReLeSR (Stemcell Technologies).

Cells were pretreated with Y-27632 (ROCK inhibitor) for 1 h before lifting, which was removed 24 h after plating. For CRISPR–Cas9 editing, preformed RNP complexes composed of sequence-specific CRISPR RNA(s) (crRNA(s)), *trans*-activating crRNA (tracrRNA) (labelled with 5′ ATTO 550) and Cas9 enzyme (all IDT) were formed following the standard IDT protocol for Neon electroporator. The electroporation of iPS cells was performed in suspension, with a single 20-ms pulse at 1,600 V. After 48 h of recovery, cells were lifted using ReLeSR, passed through 40-μm cell strainers to generate single cell suspension and sorted using a BD Biosciences FACSMelody Cell sorter (561-nm laser). ATTO 550-positive cells were plated with less than or equal to one cell per well in 96-well plates. Visible colonies were expanded in 24-well plates. For the Δrep4 iPS cell generation, the same gRNAs and screening primers as for HeLa cells were used. For the paraspeckle ablation in iPS cells, previously validated gRNAs were used[61]. Human motor neurons were differentiated from iPS cells as described for human ES cells[65]. On day 23, motor neurons were lifted with Accutase (Stemcell Technologies), replated and matured for at least 7 days, before being plated in the final format (coverslips or optical plates). Lentiviral particle production and neuronal transduction were performed as described[64]. The viability analysis was performed on Incucyte S3 Live-Cell Analysis System using the Caspase 3/7 Green Dye (Sartorius).

### RNA ISH

RNA-FISH was performed using Stellaris probes (Biosearch Technologies): Quasar570-labelled NEAT1 5′-segment (catalogued probe, SMF-2036-1), Quasar670-labelled NEAT1 middle segment (custom-made) and Quasar570-labelled NEAT1 3′-segment (custom-made) (Supplementary Table 1) as described[10,12]. For the NEAT1_2 repeat 4 detection, three oligonucleotide probes were designed to recognize the repeat-flanking sequences and custom-made by Merck (Cy3-labelled on the 5′-end): 5′-GAATGTGTTGGGCCAACTGC-3′, 5′-TCAGACCTCTCAAAGGGGTG-3′ and 5′-GAATCTGAGCAGAATCAGCC-3′. For super-resolution imaging, ProLong Gold Antifade mountant was used, and the slides were 'cured' for 24 h before imaging. RNAscope-ISH was performed as described[66], using catalogued probes from ACDbio: NEAT1_2-specific (411541) and NEAT1 total (411531).

### Immunostaining and PLA

Immunostaining was performed as described earlier[58], using the following commercial antibodies (1:1,000 dilution): FUS (rabbit polyclonal, no. 11570-1-AP, Proteintech; mouse monoclonal, no. sc-47711, Santa Cruz); TDP-43 (rabbit polyclonal, no. 10782-2-AP, Proteintech; mouse monoclonal, no. MAB7778-SP, R&D Biosystems; rabbit polyclonal C-terminal, Sigma; rabbit monoclonal E2G6G, no. 89718, Cell Signaling); Tuj (Alexa 488-conjugated, no. ab237350, Abcam). The TDP-43 SNAP-tag detection was performed by a 2-h incubation with SNAP-Cell TMR-Star dye (New England Biolabs) in full media, followed by media washes. The PLA was performed using Duolink In Situ Orange Starter Kit Mouse/Rabbit (no. DUO92102, Sigma) according to the manufacturer's instructions, with the mouse monoclonal anti-FUS antibody and either Sigma or Proteintech rabbit anti-TDP-43 antibody (incubated overnight at 4 °C).

### FRAP

Cells were grown and transfected on glass-bottomed 35-mm dishes (Mattek or Ibidi). Cells were treated if required 24 h post transfection, and imaging was done in a CO$_2$/temperature equipped chamber of a ZEISS LSM 800 confocal microscope, with a 63× oil immersion objective. Typically, three condensates were analysed per cell in at least ten cells per condition in a single experiment. For paraspeckle clusters, the entire cluster was bleached. For TDP-43 stress-induced nuclear condensates, a circular region of interest (~0.15 μm) within the condensate was bleached using a 488-nm laser. Images were acquired prebleaching,

immediately after bleaching and then at ~200 ms intervals during recovery. The mean fluorescence intensity within the region of interest was determined for each image using ZEN blue software. The intensity values were corrected for bleaching during imaging and normalized to the prebleach intensity. FRAP curves were fitted using a one-phase association equation using GraphPad Prism.

## Microscopy

Images were taken using the following systems (see text/figure legends for details): (1) conventional fluorescent imaging, Olympus BX57 microscope (100× objective) equipped with ORCA-Flash 4.0 camera (Hamamatsu) and cellSens Dimension software (Olympus); (2) SRM of paraspeckles, DeltaVision OMX SR (SIM) or ZEISS LSM 980 with Airyscan 2 detector and ZEN blue software; (3) time-lapse fluorescence imaging, Opera Phenix (40× objective) with Harmony 4.9 software; and (4) brightfield imaging for RNAscope-ISH, Nikon Eclipse Ni microscope with Nikon DS-Ri1 camera and NIS Elements v2.20.02 software. Image postprocessing was done using ZEN blue or Image J.

## RNA expression analysis

Total RNA was purified using GenElute Mammalian Total RNA Miniprep Kit (Merck), with on-column DNase digest (RNase-Free DNase Set, Qiagen). Samples in lysis buffer were heated at 55 °C for 10 min as a standard, unless for the semi-extractability analysis (see below). The first-strand complementary DNA (cDNA) synthesis was performed using 500 ng of RNA with random primers (Thermo Fisher) and MMLV reverse transcriptase (Promega) as per manufacturer's protocol. The RT–qPCR was performed using qPCRBIO SyGreen Lo-ROX. Primers for NEAT1, GAPDH, FUS and TDP-43 were from a previous study[12]; other primers are given in Supplementary Table 1.

## RNA semi-extractability analysis

For the analysis of RNA semi-extractable properties, one set of samples (in QIAzol, Qiagen or RNA kit-specific buffer) was heated at 55 °C for 10 min, and the second set was left at room temperature (RT) for 10 min. RNA was extracted from both sets as per standard QIAzol/kit protocols, followed by RT–qPCR analysis as above. The semi-extractability was determined by calculating the ratio between heated versus non-heated samples. A panel of commercial kits for total mammalian RNA isolation was tested for RNA semi-extractability analysis: E.Z.N.A. (Omega Bio-tek), PureLink (Thermo Fisher), ReliaPrep (Promega), GenElute (Merck) and RNeasy (Qiagen), alongside the standard method—a Trizol-type reagent[22]. Cells were grown in 12-well plates to ~90% confluency and lysed in either QIAzol or a kit-specific lysis buffer for 10 min at RT. For the DNase sensitivity analysis, samples were treated with intact or heat-inactivated DNase I (Qiagen) on columns for 30 min, with all subsequent steps according to the kit-specific protocol.

## Protein co-IP, oligomer crosslinking and western blotting

Cells expressing FUS deletion mutants (FUS-KO SH-SY5Y) or TDP-43 variants (HeLa) were lysed in 1% TritonX-100–PBS for 30 min on ice, an input sample was taken, followed by centrifuging for 20 min at 17,000g. Supernatant was incubated with GFP-Trap beads (Chromotek) for 3 h on a nutator at 4 °C. Beads were washed with lysis buffer three times, and the protein was eluted by heating the beads for 10 min at 95 °C in 2× Laemmli buffer. The 1% input and 25% eluate were analysed by SDS–polyacrylamide gel electrophoresis (SDS–PAGE) and western blot. For DSG crosslinking, frozen cells on plates were resuspended in 1× PBS with protease inhibitors, and 400 µM DSG (Thermo Fisher) was added for 10 min at RT with gentle rotating. The reaction was quenched with 20 mM of Tris–HCL pH 7.5 for 15 min, and samples were processed for western blot. Samples prepared in parallel without crosslinking were used as controls. SDS–PAGE and detection were carried out as described elsewhere[45]. The following commercial primary antibodies were used (1:1,000 dilution): FUS (rabbit polyclonal, no. 11570-1-AP,

Proteintech); GFP (rabbit polyclonal, no. 50430-2-AP, Proteintech); TDP-43 (rabbit polyclonal, C-terminal, Sigma); mCherry (rabbit polyclonal, no. 26765-1-AP, Proteintech); secondary horesradish peroxidase (HRP)-conjugated antibodies (Amersham). Equal protein loading was determined by GelCode staining, and blots were quantified on Image Studio software (Licor).

## FUS and TDP-43 NTD protein production and purification

FUS protein was expressed from pET24b (+) in Rosetta (DE3) *Escherichia coli*. The cell pellet was lysed in the buffer containing 8 M urea and 0.5% TritonX-100 and sonicated, and the soluble fraction was loaded onto TALON/Cobalt resin (Clontech). Eluate fractions were dialysed, and proteins were separated by gel filtration chromatography using ÄKTA Pure purification system and HiLoad Superdex 200 pg 16/600 column (GE Healthcare). TDP-43 NTD was expressed from pOPINF in BL21 (DE3) *E. coli*, and the protein was purified with nickel–nitroloacetic acid immobilized metal ion affinity chromatography followed by size exclusion chromatography. Protein samples were concentrated and stored at −80 °C.

## In vitro analysis of condensates using immunostaining (ImmuCon)

Coverslips (10-mm) were coated with 0.02% poly-L-lysine (Sigma) for 30 min and washed with water. Coverslips were incubated in 2% glutaraldehyde solution (EMS Diasum) for 40 min, washed with water once and left to dry. Recombinant untagged TDP-43 (R&D Biosystems, AP-190-100) or His-tagged FUS was diluted in the assay buffer (20 mM Tris–HCl pH 7.5, 20 mM KCl, 2 mM MgCl₂, 1 mM dithiothreitol (DTT)) to the specified concentrations. Timings of condensate/aggregate formation and protein:RNA ratios are described in the text and legends. Samples were incubated with 1,6-HD (Sigma, 4% final concentration), TMAO (Sigma, 100 mM) or *bis*-ANS (Cayman Chemicals, at molar ratios to TDP-43 4:1, 2:1, 1:1) for 10 min at RT. RNA oligonucleotides were custom-synthesized and high-performance liquid chromatography (HPLC)-purified by Eurofins. Samples were added onto the coverslip as a 5-µl droplet, mixed by pipetting and left to sediment for 15 min. Glutaraldehyde (4% in water) was added to samples (5 µl, 1:1 ratio) for 30 min. Coverslips were washed with 1× PBS twice. For immunodetection, coverslips were first blocked with 1% bovine serum albumin (BSA) in TritonX-100–PBS at RT for 1 h and then incubated with primary antibodies (1:5,000) diluted in blocking buffer for 1 h at RT. Commercial antibodies against TDP-43 (mouse monoclonal, no. MAB7778, R&D Biosystems) and FUS (rabbit polyclonal, no. 11570-1-AP, Proteintech) were used, after confirming lack of cross-reactivity. Signal was detected using secondary antibodies (Invitrogen, Alexa Fluor 488/546/633 mouse/rabbit), and coverslips were mounted on a glass slide using ImmuMount (Thermo Fisher). Images were taken using Olympus BX57 upright microscope (100× objective) and ORCA-Flash 4.0 camera. A detailed version of this protocol is available at ref. 67.

## TDIPS

The assay was performed in a PC capillary (Fidabio) of 1 m length and an inner diameter of 75 µm, using a Fida 1 equipped with a 280 nm fluorescence detector, with a cutoff filter of 300 nm. The capillary was filled with either storage buffer or assay (LLPS) buffer, and the proteins were loaded into the capillary with a pressure driven flow. The method settings were as follows: 3,500 mbar and 45 s for washes; 50 mbar and 20 s for analyte; 50 mbar and 15 s for indicator; 500 mbar and 180 s for measurement. The assay was performed at 25 °C with the autosampler cooled to 4 °C.

## Phase separation assay with ybbR-cyanine tagged proteins

TDP-43 and FUS were purified from BL21 Gold *E. coli* harbouring the pET-TDP-43/FUS ybbR mutant-TEV-MBP-6×His plasmids generated in-house. Target proteins were labelled as previously described[68] by

incubating with Sfp and CoA-Cy5 or CoA-Cy3, at a 1:1:4 molar ratio. Labelled and unlabelled proteins were combined at a 1:25 ratio and incubated with an equimolar amount of uTEV3 for 2 h at RT at high enough KCl concentrations to prevent phase separation. To initiate phase separation, the final concentrations were adjusted to 150 mM KCl, 2 mM DTT, 4.8 µM unlabelled protein and 0.2 µM labelled protein, the samples were transferred to 96-well non-binding µClear plates (Greiner Bio-One) and incubated for 30 min. Confocal images were captured and analyzed as described previously[69].

### Molecular dynamics simulations of TDP-43 and FUS LCD interactions

The version of the 1-bead-per-amino-acid molecular dynamics model used is described in detail previously[70]. All simulations were carried out at a temperature of 300 K using a timestep of 20 fs. For the equilibration of the droplet, the energy minimization on the initial configuration was used (energy tolerance of $1 \, kJ \, mol^{-1} \, nm^{-1}$), before 50 ns NVT Langevin dynamics simulations (Nosé-Hoover thermostat with $\tau_t = 100 \, ps$), followed by 500 ns NPT Langevin dynamics (Nosé-Hoover thermostat with $\tau_t = 100 \, ps$ and a Berendsen barostat with $\tau_p = 10 \, ps$, 1 bar reference pressure and a compressibility of $4.5 \times 10^{-5} \, bar^{-1}$). The end state of the NPT equilibration step was inserted into a new periodic box with a volume chosen to give a total particle density of 80,000 µM, after recentring on the centre of mass and after the molecules had been unwrapped across the previous periodic boundary conditions. A second energy minimization step was applied in the new simulation box to relax the molecules after the box expansion (energy tolerance of $1 \, kJ \, mol^{-1} \, nm^{-1}$). A final 3 µs NVT production run (Nosé-Hoover thermostat with $\tau_t = 100 \, ps$) was used for data collection. The trajectory was sampled every 5 ns to determine whether convergence was reached. The contact maps were computed and analysed using the protocols detailed in ref. [70].

### AlphaFold predictions

FUS–TDP-43 multimer structures were predicted by AlphaFold 3[71]. Protein sequences were input to the server with the indicated copy number and automatic seed setting. The top-ranked results were output and visualized in Pymol v.2.55.5.

### NEAT1_2 TG repeat analysis in the Project MinE cohort

Project MinE was approved by the Trent Research Ethics Committee 08/H0405/60. Informed consent for genetic research was obtained from all participants. The 6,739 individuals, including 4,996 patients with ALS and 1,743 controls, subject to WGS and included in this study were recruited at specialized neuromuscular centres in the UK, Belgium, Germany, Ireland, Italy, Spain, Turkey, the USA and the Netherlands[72]. Patients were diagnosed with possible, probable or definite ALS according to the 1994 El-Escorial criteria. All controls were free of neuromuscular diseases and matched for age, sex and geographical location. ExpansionHunter (Illumina) software was used to determine repeat lengths of the *NEAT1* TG-repeats located on chromosome 11 between positions: 'repeat 1', 65429481-65429519; 'repeat 2', 65432669 and 65432684; 'repeat 3', 65433717-65433735; 'repeat 4', 65444263 and 6544432 (Genome build 38) in the Project MinE dataset datafreeze 2 (DF2). One outlier probably originating from a sequencing error was excluded.

### Statistics and reproducibility

Datasets were tested for normality using appropriate tests in GraphPad Prism (version 10.6.1). *P* values were calculated in GraphPad Prism using an appropriate parametric or non-parametric test, depending on the nature of the data and the recommended post hoc test for correction for multiple comparisons, where applicable. No statistical methods were used to predetermine sample size, but our sample sizes are similar to those reported in our previous publications[7,10,12,29,45]. No data were excluded from the analyses, except a single outlier in the human genetic dataset. The experiments were not randomized. The investigators were blinded to the experimental condition where possible. Experiments were repeated at least three times, with similar results. *N* indicates the number of biological replicates (obtained in independent experiments performed on different days).

### Reporting summary

Further information on research design is available in the Nature Portfolio Reporting Summary linked to this article.

## Data availability

All data that are necessary to interpret, verify and extend the research in the article are provided in the Article and its Supplementary Information. Expression plasmids generated in the study are available via Addgene at https://www.addgene.org/Tatyana_Shelkovnikova/ (ref. [60]). All other unique study materials are available from the corresponding author; a signed material transfer agreement (MTA) may be required for transfer. All other data supporting the findings of this study are available from the corresponding author on reasonable request. Source data are provided with this paper.

## Code availability

No original code was generated in this study.

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

## Acknowledgements

The work was supported by the UKRI Future Leaders Fellowship (grant no. MR/W004615/1), MRC (grant no. MR/W028522/1) and BBSRC (grant nos. BB/V014110/1 / BB/V014528/1 and BB/X018393/1) grants and the MND Association fellowship/grant (grant no. 968-799) to T.A.S. R.L. and B.S.C.E. are supported by PhD studentships from the MND Association (to T.A.S. and S.P.A.) and MND Scotland (to T.A.S.), respectively. V.K. and E.D. are supported by PhD studentships from the University of Sheffield Neuroscience Institute and Faculty of Health, respectively (to T.A.S.). The work was also funded by the MND Association fellowship (grant no. 2323-799) to T.M., Dutch national e-infrastructure—SURF Cooperative grant EINF-5917 to P.R.O., as well as the MND Association fellowship (grant no. 969-799), Academy of Medical Sciences (grant no. SBF008\1028) and Royal Society (grant no. RGS\R1\231005) grants to G.S.A.W. and BBS/E/PI/23NB0003 grant to N.L. The funders had no role in study design, data collection and analysis, decision to publish or preparation of the manuscript.

## Author contributions

T.A.S. conceived the study, designed and conducted experiments, analysed data, supervised the study and wrote the manuscript with input from all authors. R.E.H., W.-P.H., R.L., V.K., Z.P.C., M.D.D., H.A., E.G.P.S., E.D. and K.B. designed and conducted experiments and analysed data. A.S.A., B.C.S.E., J.A.R., A.S. and T.M. conducted experiments. J.J.F.A.V. and J.C.-K. analysed the datasets assembled by Project MinE Sequencing Consortium. G.S.A.W., P.R.O., O.D., N.L., S.P.A., I.P. and S.F. provided tools, contributed to the experimental design and interpretation of findings or funding acquisition. All authors contributed to the manuscript writing and approved its final version.

## Competing interests

E.G.P.S. is an employee of Fidabio. The other authors declare no competing interests.

## Additional information

**Extended data** is available for this paper at https://doi.org/10.1038/s41556-026-01895-y.

**Correspondence and requests for materials** should be addressed to Tatyana A. Shelkovnikova.

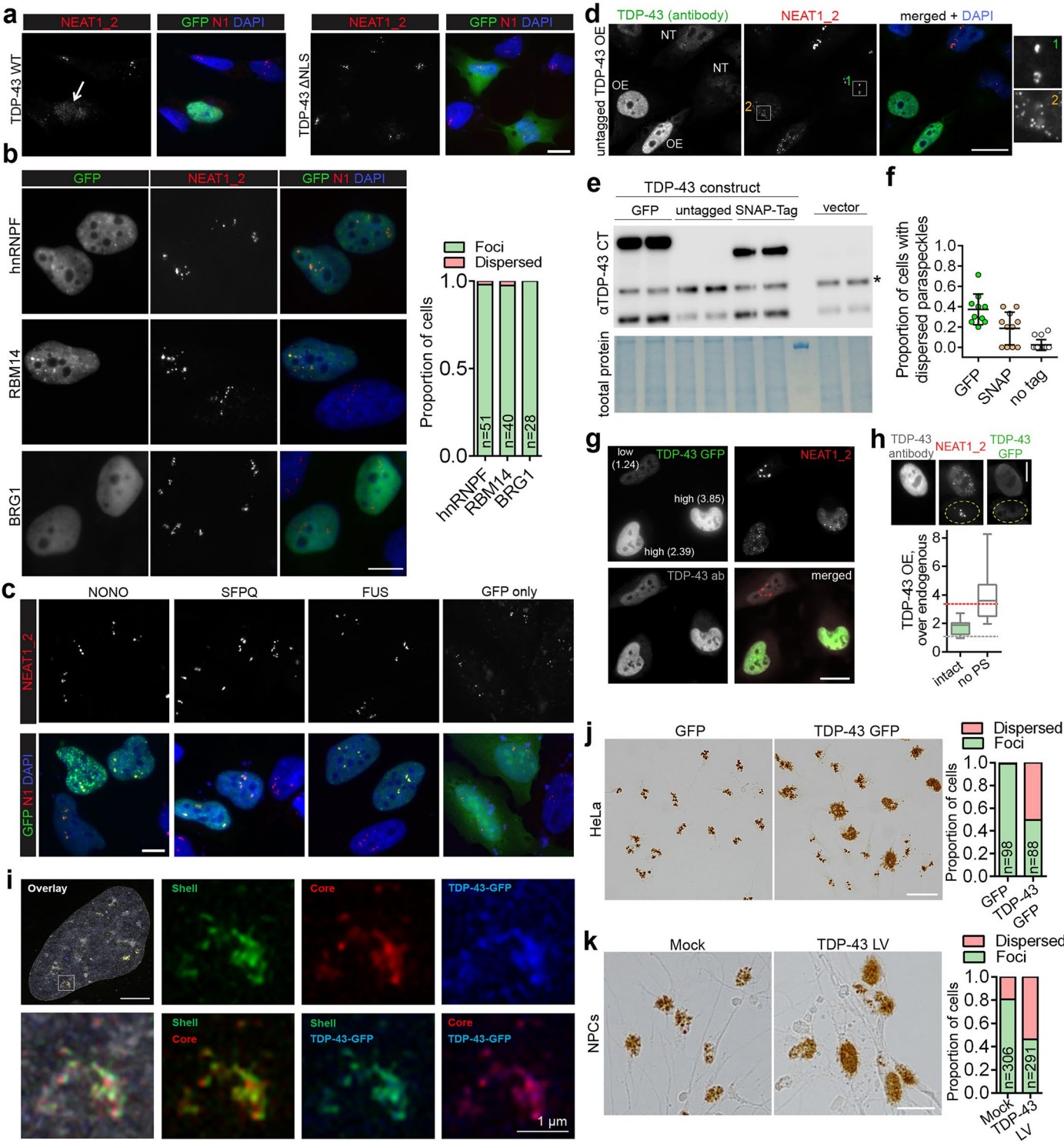

**Extended Data Fig. 1 | See next page for caption.**

**Extended Data Fig. 1 | Ectopically overexpressed nuclear TDP-43, but not other PSPs, disperses paraspeckles, in a concentration-dependent manner.** **a**) Nuclear but not cytoplasmic TDP-43 OE disrupts paraspeckles. WT and ΔNLS TDP-43 were expressed in SH-SY5Y cells, and paraspeckles were analysed by NEAT1_2 RNA-FISH. Arrow indicates a transfected cell with dispersed paraspeckles. Scale bar, 5 μm. **b**) The OE of shell-localized PSPs does not affect paraspeckle clusters. Quantification from a representative experiment is shown. Scale bar, 10 μm. **c**) The OE of GFP alone or core-localized PSPs does not affect paraspeckle clusters. Scale bar, 5 μm. **d**) Paraspeckle dispersal by untagged TDP-43 OE. NT, non-transfected. Scale bar, 20 μm. **e**) TDP-43 OE validation. Representative blot is shown; empty pEGFP-C1 vector was used as a control. Endogenous TDP-43 band is indicated with an asterisk. **f**) Paraspeckle dispersal in cells with ectopic expression of TDP-43 GFP, TDP-43 SNAP-Tag and untagged TDP-43. 66, 54 and 98 cells (n = 10, 11 and 19 FoV) were quantified for GFP, SNAP-tag and untagged TDP-43, respectively, from a representative experiment. Graph shows mean ± s.d. **g,h**) Concentration threshold for paraspeckle disruption by TDP-43. Representative image (**g**) and quantification (**h**) of triple-labelled cells with TDP-43 OE are shown. In **g**, fold OE is indicated for low- and high-expressing

cells. In **h**, NT cell is circled. Grey and red lines indicate the endogenous level and mean disruptive concentration, respectively. Fluorescence intensity of TDP-43 antibody signal (OE+endogenous protein) was analysed in n = 54 transfected and 19 NT cells, from a representative experiment. Graph: box, 25th to 75th percentiles; center line, median; whiskers, mean to max. Scale bars, 15 μm in **g** and 10 μm in **h**. **i**) Disordered NEAT1 RNP aggregates in cells with high TDP-43 OE are enriched in TDP-43. Aggregates were analysed by dual NEAT1 RNA-FISH and SRM. Scale bar in overlay, 3 μm. **j**) Paraspeckle dispersal analysis using RNAscope-ISH for NEAT1_2. Representative images and quantification, from a representative experiment, are shown. Scale bar, 30 μm. **k**) Lentiviral expression of untagged TDP-43 leads to paraspeckle dispersal in human NPCs. Day 16 NPCs were used. Cells were analysed 72 h post-transduction using a NEAT1_2 probe. Representative images and quantification, from a representative experiment, are shown. Scale bar, 20 μm. HeLa cells were used, except in **a** and **k**. Cells were analysed 24 h post-transfection. Representative images are shown in **a-d**, **i**. Number of cells (n) analysed is indicated within bars in **b,j** and **k**. Source numerical data and unprocessed blots are available in the Source data.

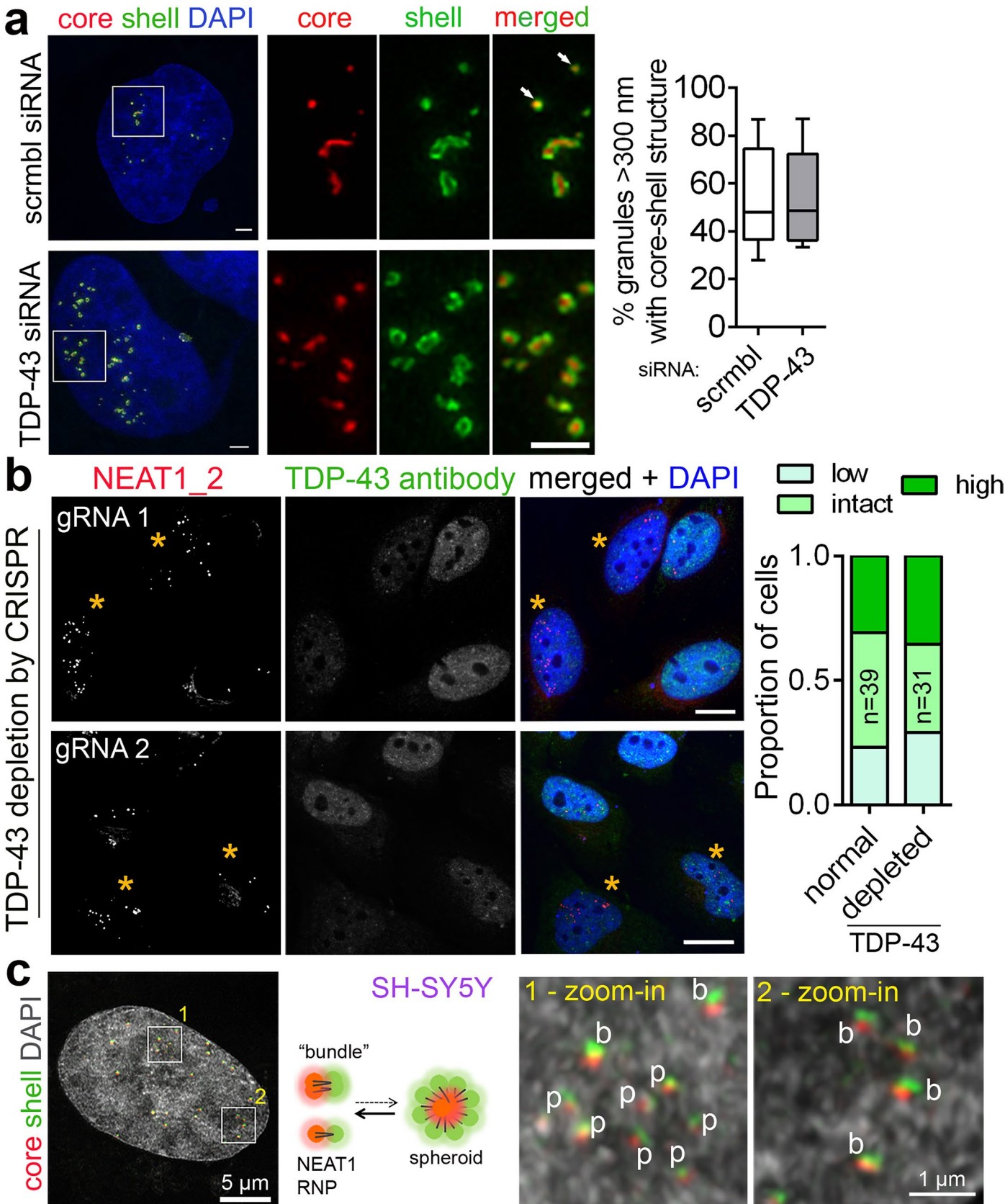

**Extended Data Fig. 2 | See next page for caption.**

**Extended Data Fig. 2 | Paraspeckle spheroid formation is not affected by TDP-43 depletion. a**) Normal paraspeckle core–shell structure upon siRNA-mediated TDP-43 depletion. Dual NEAT1 RNA-FISH and SRM was used in HeLa cells. Arrows indicate incomplete spheroids. 263 and 231 large ( > 300 nm) granules were analysed for scrambled and TDP-43 siRNA, respectively (from n = 10 FoV), from a representative experiment. Box, 25th to 75th percentiles; center line, median; whiskers, mean to max. Scale bar, 1 µm. **b**) Normal paraspeckle core–shell structure upon CRISPR/Cas9-mediated TDP-43 depletion. Two gRNAs were used in HeLa cells, and quantification was performed for gRNA1, 72 h post-transfection. Cells with visible TDP-43 loss (asterisks) were categorised as 'depleted'. Cells with normal TDP-43 levels were quantified from the same field of view. 'Low', 'intact' or 'high' – arbitrary scoring of NEAT1-positive cluster abundance. Number of cells analysed (n) (from a representative experiment) is indicated within bars. Scale bar, 10 µm. **c**) Smaller NEAT1-positive particles in SH-SY5Y cells represent a mix of single NEAT1 RNPs (paraspeckle precursors, *p*) and their higher-order assemblies (bundles, *b*). Representative images are shown. Note that in SH-SY5Y cells, NEAT1 RNPs in bundles are joined both at their middle and termini, in contrast to disordered NEAT1 RNP multiplets in TDP-43 OE cells (Fig. 1g). Source numerical data are available in the Source data.

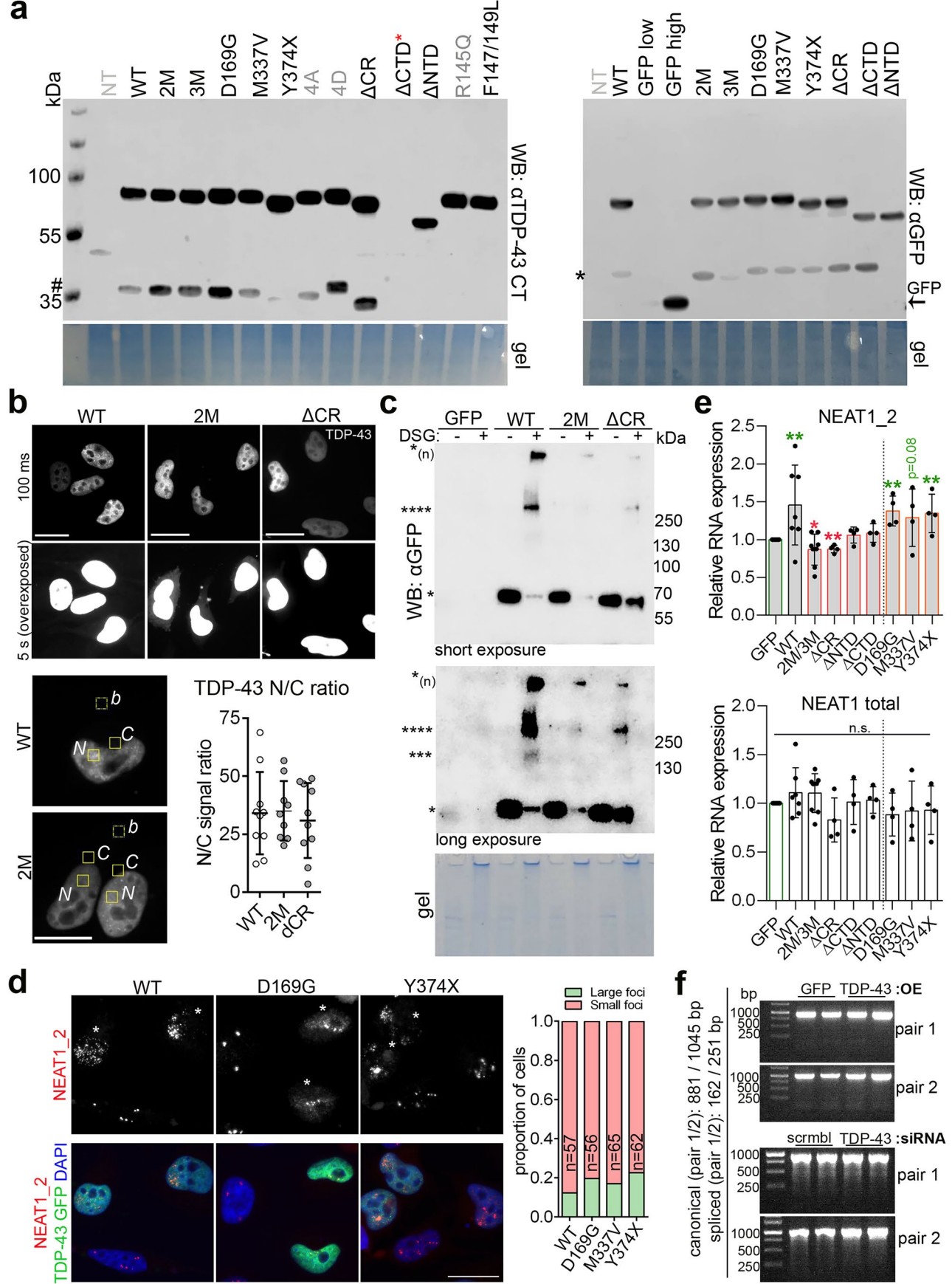

**Extended Data Fig. 3 | See next page for caption.**

**Extended Data Fig. 3 | TDP-43 oligomerization is NEAT1 and paraspeckle regulation. a**) Western blot analysis of TDP-43 variant expression. Variants not investigated in this study are greyed. # indicates a C-terminal TDP-43 cleavage product and asterisk indicates a non-specific band. Note that ΔCTD (red asterisk) is not recognised by the anti-TDP-43 C-terminal antibody used. **b**) TDP-43 oligomerization-deficient variants are efficiently retained in the nucleus in HeLa cells. GFP-tagged variants were used. Imaging with long exposure reveals some cytoplasmic accumulation of TDP-43$^{2M}$ (top), however nuclear/cytoplasmic ratio (N/C) is not significantly different between the variants (bottom). Fluorescence intensity was measured in ROIs in the nucleus (N) and cytoplasm (C), with background (*b*) fluorescence subtracted. n = 10 cells for each condition, from a representative experiment. Scale bars, 20 μm. **c**) Confirmation of reduced oligomerization ability for TDP-43$^{2M}$ and TDP-43$^{ΔCR}$ variants using DSG crosslinking. *, ***, ****, and *$_{(n)}$ indicate TDP-43 GFP monomer, trimer, tetramer and higher-order multimer, respectively (based on molecular weight; monomer, ~70 kDa). **d**) ALS-linked TDP-43 mutations do not significantly affect paraspeckle

dispersal by TDP-43 OE. Asterisks indicate TDP-43 GFP expressing cells. Number of cells analysed (n) (from a representative experiment) is indicated within bars. Scale bar, 20 μm. **e**) Monomeric TDP-43 mutants decrease NEAT1_2 production. NEAT1 was analysed by qRT-PCR in cells expressing natural or engineered TDP-43 variants, 24 h post-transfection. Graph shows mean ± s.e.m. N = 4-8. *p ≤ 0.05, **p < 0.01, one-tailed Mann-Whitney *U* test (pairwise comparisons): WT, ΔCR, 2 M/3 M, D169G and Y374X *vs*. GFP: p = 0.0084, p = 0.003, p = 0.05, p = 0.003 and p = 0.003, respectively; n.s., non-significant. Green and red asterisks – increased and decreased expression compared to GFP control, respectively. Note that oligomerisation-deficient TDP-43$^{2M}$ and TDP-43$^{ΔCR}$ decrease NEAT1_2 production, whereas TDP-43$^{WT}$ and two out of three familial mutants increase it. **f**) TDP-43 OE or knockdown does not cause NEAT1_2 splicing. The NEAT1_2 splicing isoform reported previously (Ref. 22) was analysed by PCR with two primer pairs flanking the splice sites. The spliced isoform was undetectable in all cases. Source numerical data and unprocessed blots/gels are available in the Source data.

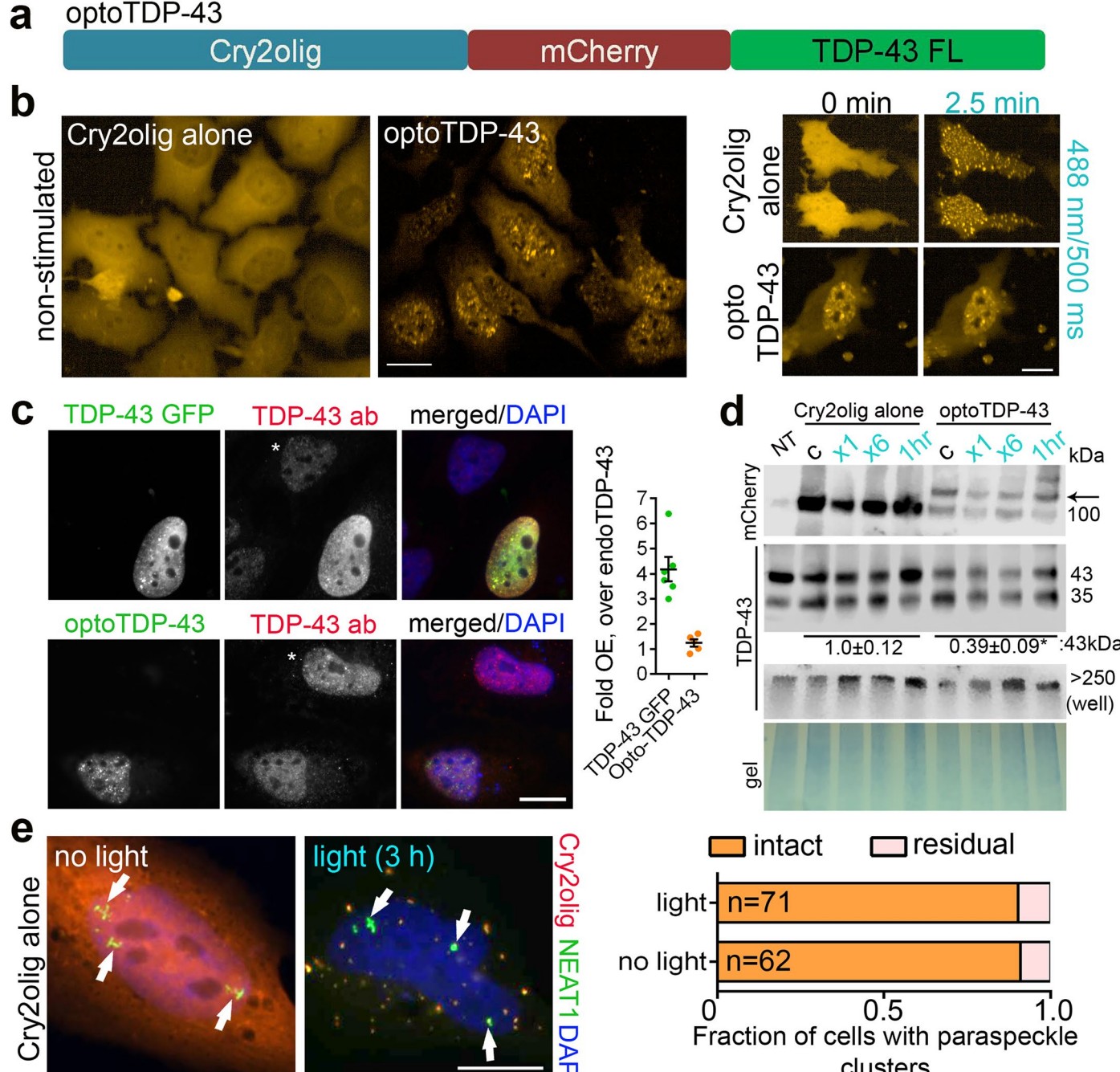

**Extended Data Fig. 4 | OptoTDP-43 for the analysis of TDP-43 self-association in paraspeckle assembly. a**) Schematic representation of the optoTDP-43 construct. FL, full-length. **b**) OptoTDP-43 and Cry2olig-only distribution with and without blue light stimulation. Representative images are shown. Scale bars, 10 μm. **c**) OptoTDP-43 is expressed at near-endogenous levels. Representative images of TDP-43 GFP and optoTDP-43 expressing cells stained for TDP-43 and single-cell fluorescence intensity quantification relative to the endogenous level (measured in NT cells in the same FoV) are shown. C-terminal TDP-43 antibody was used. Asterisks indicate NT cells. 13 transfected and 25 non-transfected cells (n = 5 or 6 FoV) were analysed, from a representative experiment. Graph shows mean ± s.e.m. Scale bar, 10 μm. **d**) Endogenous TDP-43 downregulation in optoTDP-43 expressing cells. Cells expressing optoTDP-43 or Cry2olig were subjected to repetitive (x1/x6 pulses) or continuous (1-h long) blue light stimulation and analysed by western blot. Arrow indicates optoTDP-43. Note no

endogenous TDP-43 or optoTDP-43 aggregation or fragmentation after blue light stimulation (*that is* no difference between Cry2olig vector and optoTDP-43 in the 35-kDa fragment abundance or protein amount retained in the well – complexes >250 kDa). Endogenous TDP-43 (43-kDa band) was quantified by densitometry (mean ± s.e.m.). N = 4, *p < 0.05 (p = 0.0143), one-tailed Mann-Whitney *U* test. **e**) Cry2olig-only oligomerisation does not affect paraspeckle clusters. Note that Cry2olig readily forms cytoplasmic aggregates upon blue light stimulation. Paraspeckle clusters (arrows) detected with a NEAT1_2 middle (core) probe in cells with or without continuous blue-light stimulation. Representative images and quantification (from a representative experiment) are shown. Number of cells analysed (n) is shown within bars. Scale bar, 10 μm. HeLa cells were used in these experiments. Analyses were performed 24 h post-transfection. Source numerical data and unprocessed blots are available in the Source data.

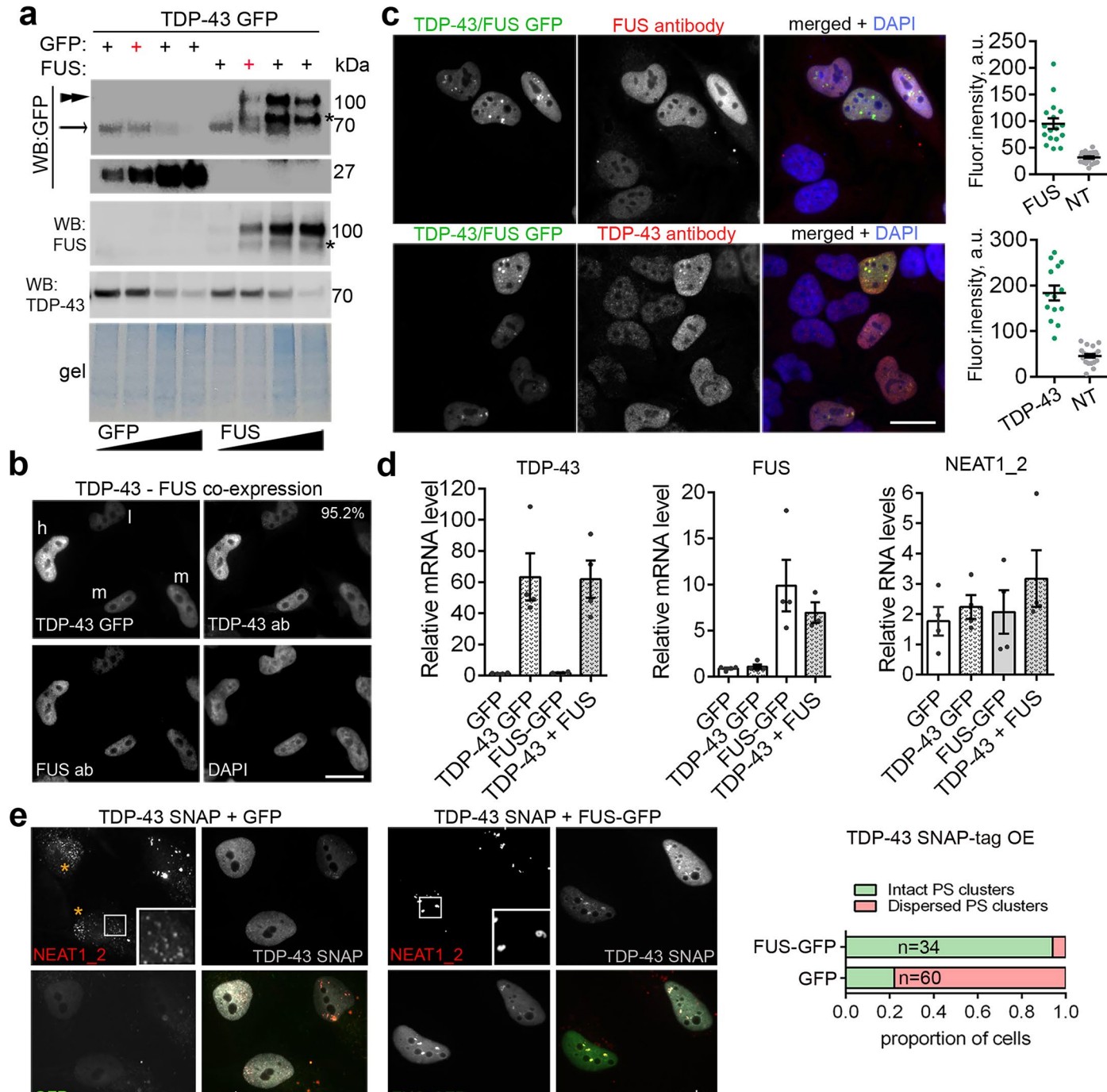

**Extended Data Fig. 5 | FUS supplementation rescues paraspeckles in cells with increased TDP-43 dosage. a**) Plasmid titration as analysed by western blot. Red (+) indicates optimal co-transfection ratios used in further studies. Arrow, double arrowhead and asterisk indicate TDP-43 GFP, FUS-GFP and a non-specific band, respectively. **b**) Triple-colour fluorescence (GFP, green; TDP-43, red; FUS, far-red) confirms efficient co-expression of GFP-tagged TDP-43 and FUS (in 95.2% of cells). l,m,h – low, medium and high GFP signal, respectively. Representative images are shown. **c**) Single-cell fluorescence quantification indicates ~3-4 fold OE of GFP-tagged FUS and TDP-43 upon their co-expression (over the endogenous). Representative images and quantification are shown.

NT cells in the same FoV were used to determine the endogenous protein levels. n = 37 cells for FUS and n = 35 cells for TDP-43, analysed from three independent experiments (N = 3). Graphs show mean±s.e.m. **d**) TDP-43 mRNA, FUS mRNA and NEAT1_2 RNA levels upon co-expression of GFP-tagged TDP-43 and FUS as analysed by qRT-PCR. N = 4. Graphs show mean±s.e.m. **e**) FUS-GFP restores paraspeckle clusters in cells expressing TDP-43 SNAP-Tag. Representative images and quantification (from a representative experiment) are shown. Number of cells quantified (n) is shown within bars. Scale bar, 20 μm. HeLa cells were used for these studies. Source numerical data and unprocessed blots are available in the Source data.

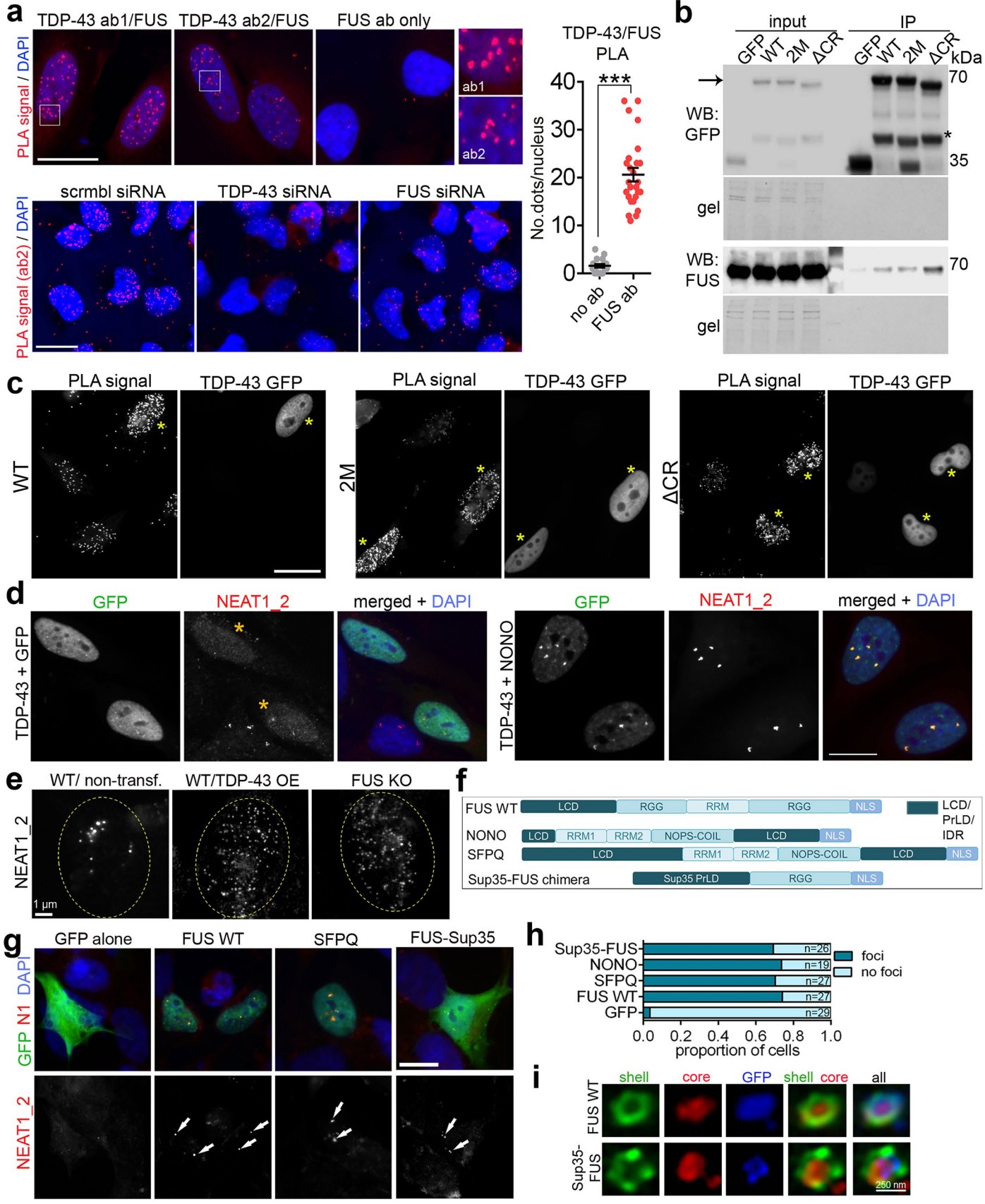

**Extended Data Fig. 6 | See next page for caption.**

**Extended Data Fig. 6 | The interplay between TDP-43 and core PSPs in paraspeckle assembly. a)** TDP-43 and FUS rarely interact in the nucleoplasm, as revealed by proximity ligation assay (PLA). Representative image and quantification are shown. ab1,2 – two different TDP-43 antibodies were used. Specificity of interaction was also confirmed using TDP-43 and FUS knockdown (bottom). Representative images and quantification are shown. n = 16 and 26 cells were analysed for 'no ab' and 'FUS ab', respectively, from a representative experiment. Graph shows mean±s.e.m. ***p = 0.001, one-tailed Student's *t* test. Scale bars, 20 μm. **b,c)** Oligomerisation-deficient TDP-43 mutants retain the ability to interact with FUS. Interactions were analysed by co-IP (**b**) or PLA (**c**) in cells ectopically expressing TDP-43 GFP. Representative western blots are shown. Asterisks indicate TDP-43 OE cells. Scale bar, 20 μm. **d)** Increased NONO dosage restores paraspeckle clusters in TDP-43 OE cells. Representative images are shown. Asterisks indicate cells with dispersed paraspeckles. Scale bar, 10 μm. **e)** FUS knockout (KO) cells and TDP-43 OE cells have a converging phenotype with disrupted paraspeckle condensation, despite NEAT1 RNP accumulation. Paraspeckles in WT SH-SY5Y cells with and without TDP-43 GFP OE and in FUS KO SH-SY5Y cells were analysed. Representative images are shown. **f)** Proteins used in paraspeckle spheroid rescue experiments (as N-terminally GFP-tagged fusions). **g-i)** Paraspeckle cluster (**g,h**) and spheroid (**i**) rescue by ectopic expression of proteins from **e** in FUS KO SH-SY5Y cells. Representative images (conventional, **g** and SRM, **i**) and quantification for clusters (**h**) (from a representative experiment) are shown. Number of cells quantified (n) is shown within bars. Arrows indicate restored paraspeckle clusters. In **f**, proteins are not strictly to scale. Scale bar, 10 μm. HeLa cells were used in a-d. Source numerical data and unprocessed blots are available in the Source data.

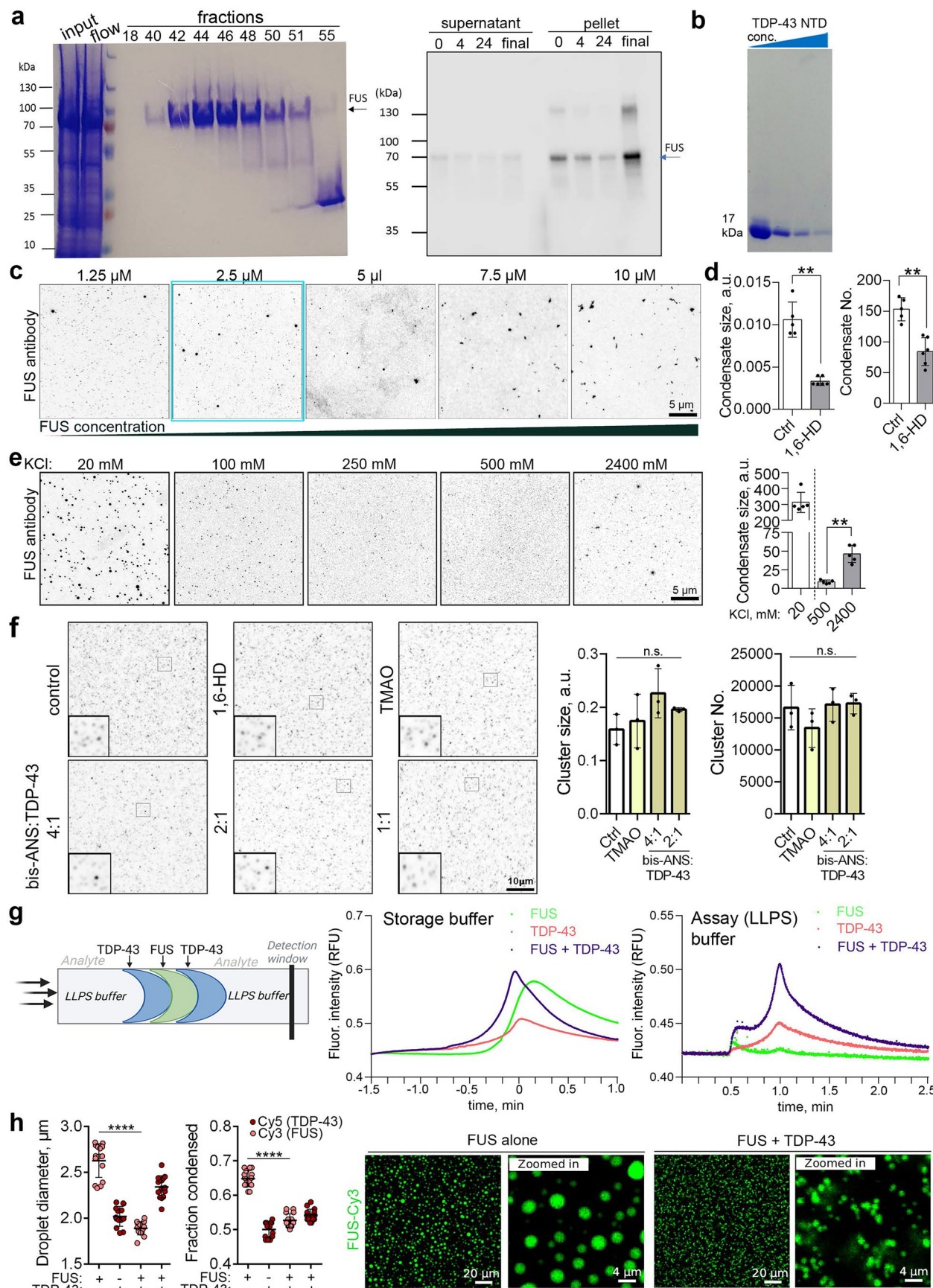

**Extended Data Fig. 7 | See next page for caption.**

**Extended Data Fig. 7 | ImmuCon assay development and the effect of TDP-43 on FUS condensation in orthogonal assays. a**) FUS (His-tag) protein fractions during purification (40-51 were collected for experiments) and FUS protein solubility in the ImmuCon assay buffer. FUS protein was incubated in the assay buffer, and assemblies were pelleted by 17,000xg centrifugation at the indicated time-points. 'Final' corresponds to the sample collected by adding Laemmli buffer directly to the tube at the 24-h time-point. Note that FUS condensates are progressively lost from the sample through adsorption on the tube walls (ending in the 'final' pellet fraction). Assay window of 10-30 min was established based on these experiments, to avoid FUS (condensate) loss from the assay. **b**) Recombinant TDP-43 NTD used in the study. **c**) FUS condensation is concentration-dependent. Note that FUS shows signs of aggregation (assembly into irregularly shaped structures) at a concentration ≥7.5 μM. Condensates were analysed after 10 min of incubation. Representative images are shown. Blue square indicates an optimal concentration used in most experiments. **d**) FUS condensates are sensitive to 1,6-hexanediol (1,6-HD). Preformed condensates were incubated with 4.0% 1,6-HD for 10 min. Condensate number and area were quantified from n = 5-6 FoV per condition (~ 600 condensates in the control sample), from a representative experiment. Graphs show mean±s.d. **p < 0.01 (p = 0.0022), one-tailed Mann-Whitney *U* test. **e**) Re-entrant phase separation of FUS. FUS condensates are dissolved by increasing salt concentration (≥ 100 mM) but reform in a very high salt (≥ 2400 mM). Condensates were analysed after 30 min of incubation. Condensate area was quantified from n = 5 FoV per condition, from a representative experiment. Graph shows mean±s.d. **p < 0.01 (p = 0.004), one-tailed Mann-Whitney *U* test. **f**) Recombinant TDP-43 clusters do not respond to known modulators of TDP-43 phase separation, bis-AMS and TMAO. Post-centrifuge TDP-43 samples were incubated with the compounds for 10 min. Cluster number and area were quantified from n = 3 FoV per condition, from a representative experiment. Graphs show mean±s.d. n.s., non-significant, one-tailed Mann-Whitney *U* test. **g**) Validation of TDP-43 effect on FUS condensation using Taylor dispersion-induced phase separation (TDIPS). FUS and TDP-43 were injected at 17.5 μM and 9 μM, respectively. Note that the curves are additive in the storage buffer (indicative of no condensation), that FUS is almost entirely in the condensate state in the assay buffer, and a dramatic increase in the monomeric protein upon co-injection with TDP-43. Data is from a representative experiment. **h**) TDP-43 effect on FUS condensation studied using full-length recombinant proteins site-specifically labelled with small fluorophores (Cy3/Cy5) via the ybbR peptide tag. Labelled proteins were spiked into unlabelled protein samples at 1:25 ratio, before MBP tag cleavage and phase separation analysis. Analysis was done on n = 15 FoV per condition, from a representative experiment. Graph shows mean±s.d. ****p < 0.0001, one-way ANOVA with Holm-Sidak post-hoc test (comparison of FUS-only and FUS-TDP-43 samples – FUS-Cy3 channel). Image in panel **g** created using BioRender. Source numerical data and unprocessed blots are available in the Source data.

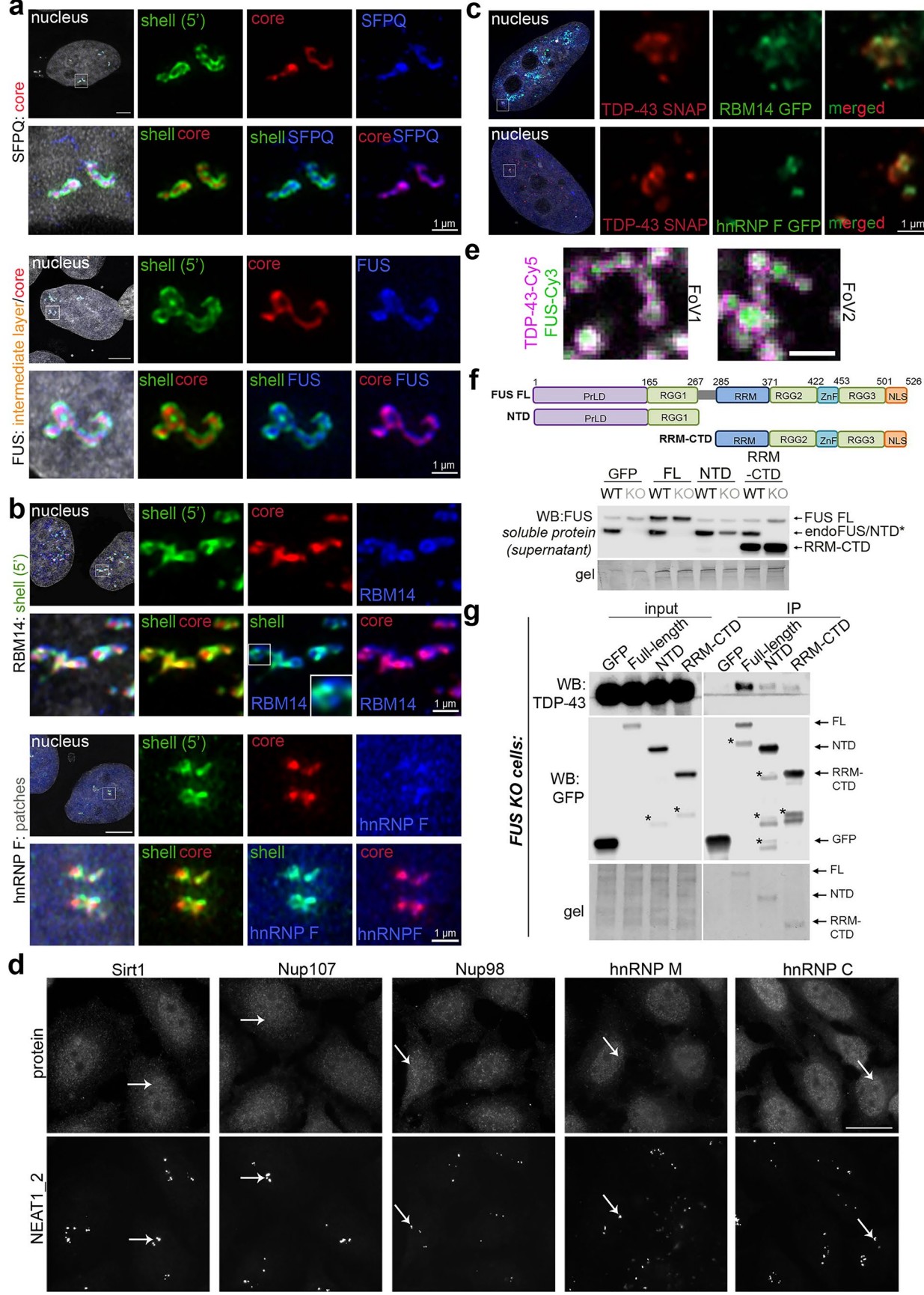

**Extended Data Fig. 8 | See next page for caption.**

**Extended Data Fig. 8 | PSP interaction in the microphase arrangement of the paraspeckle condensate. a**) GFP-tagged SFPQ and FUS demonstrate localisation to the core or intermediate layer within the paraspeckle spheroid, respectively. **b**) Analysis of PSP microphases in the paraspeckle spheroid shell. RBM14 forms clusters in the paraspeckle shell, orderly positioned and not overlapping with NEAT1 5' end probe signal (inset). In contrast, hnRNP F forms smaller clusters, or patches, within the paraspeckle, without an obvious localisation pattern. **c**) RBM14 and hnRNP F clusters within the spheroid do not overlap with TDP-43 clusters. GFP-tagged versions of these shell proteins and TDP-43 SNAP-Tag were used. **d**) Paraspeckles are not enriched in RBPs previously reported in other TDP-43 nuclear condensates. **e**) FUS and TDP-43 are immiscible within condensates reconstituted in the phase separation assay using ybbR-Cy3/5-labelled proteins. **f**) Optimised conditions for FUS deletion mutant complementation in FUS KO cells. GFP-tagged FUS variants were used. Note that the bands for endogenous FUS and ectopically expressed FUS NTD overlap on the blot (same MW, *). Soluble protein extracts were used for these experiments due to FUS NTD being prone to aggregation. Representative western blots are shown. **g**) More efficient interaction of full-length FUS with TDP-43, as compared to FUS NTD and FUS CTD in isolation. Representative western blots are shown. Asterisks indicate non-specific bands. Representative images are shown in **a-e**. Source numerical data and unprocessed blots are available in the Source data.

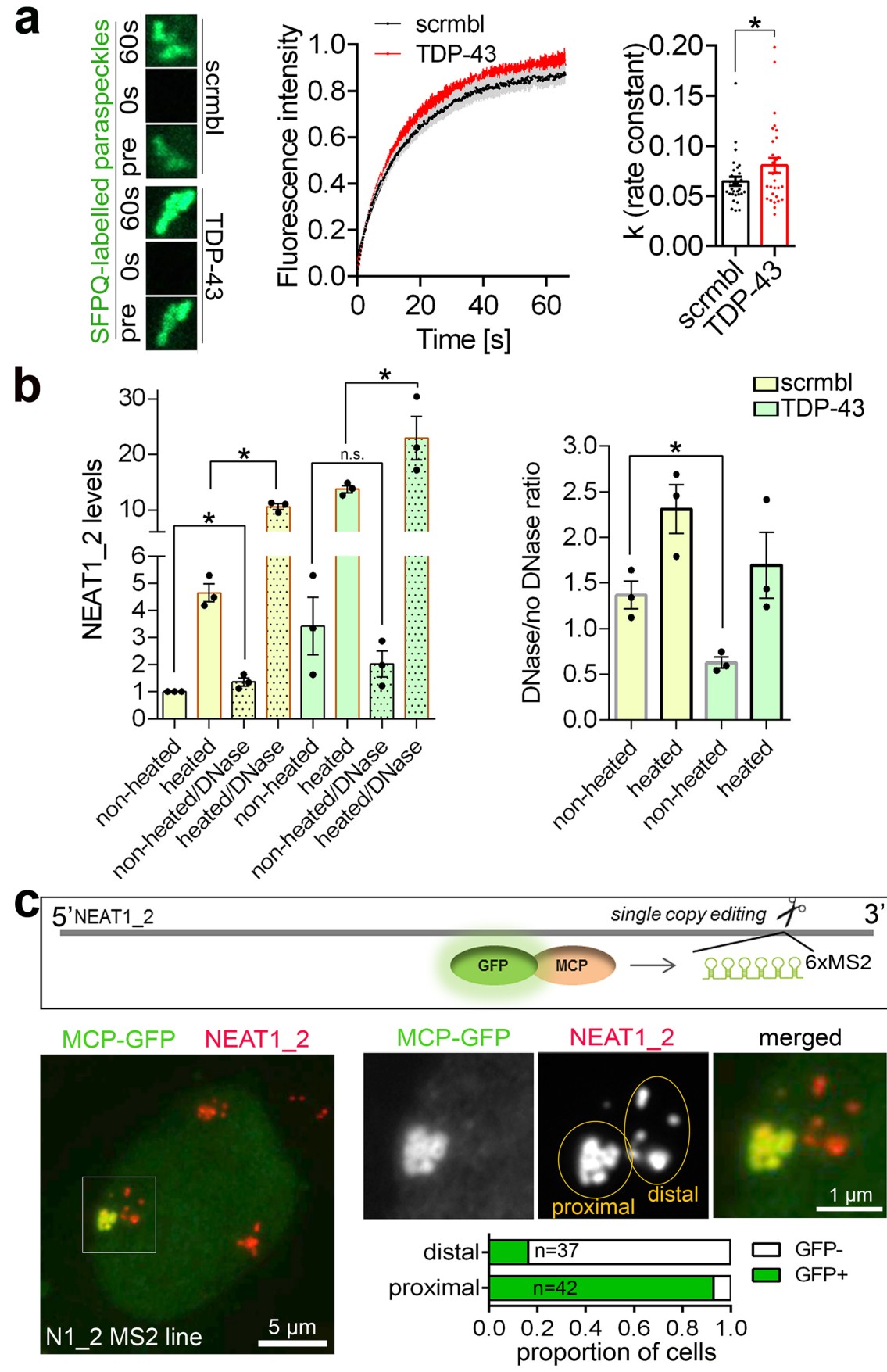

**Extended Data Fig. 9 | See next page for caption.**

**Extended Data Fig. 9 | Paraspeckle properties upon TDP-43 depletion.**
**a**) TDP-43-depleted paraspeckles are more dynamic. Paraspeckle clusters (SFPQ-GFP labelled) display faster fluorescence recovery in TDP-43-depleted cells in FRAP analysis. Cells were analysed 48 h post-transfection with scrambled or TDP-43 siRNA (co-transfected with SFPQ-GFP). Knockdown efficiency was confirmed by immunostaining. Analysis is for n = 15 cells per condition, from a representative experiment. Graph shows mean±s.e.m.; *p = 0.0362, one-tailed Mann-Whitney *U* test. **b**) TDP-43 knockdown decreases NEAT1_2 sensitivity to DNase digest. TDP-43 was downregulated using siRNA, total RNA was purified from cells 48 h post-transfection, and NEAT1_2 levels were analysed by qRT-PCR.

High sample/buffer ratio was used in this experiment (see Supplementary Fig.3). Note that DNase significantly increases NEAT1_2 extraction in TDP-43-sufficient but not TDP-43-deficient cells, where this effect is eliminated by sample heating. Graphs show mean±s.e.m. N = 3, *p = 0.05, one-tailed Mann-Whitney *U* test. **c**) Remodelling of the paraspeckles distal to the *NEAT1* locus. Schematic of MS2 labeling of the *NEAT1* locus, representative image of a cell with groups of distal and proximal paraspeckles labelled and quantification of GFP-positivity of proximal and distal paraspeckles (from a representative experiment) are shown. Number of cells quantified (n) is indicated within bars. Source numerical data are available in the Source data.

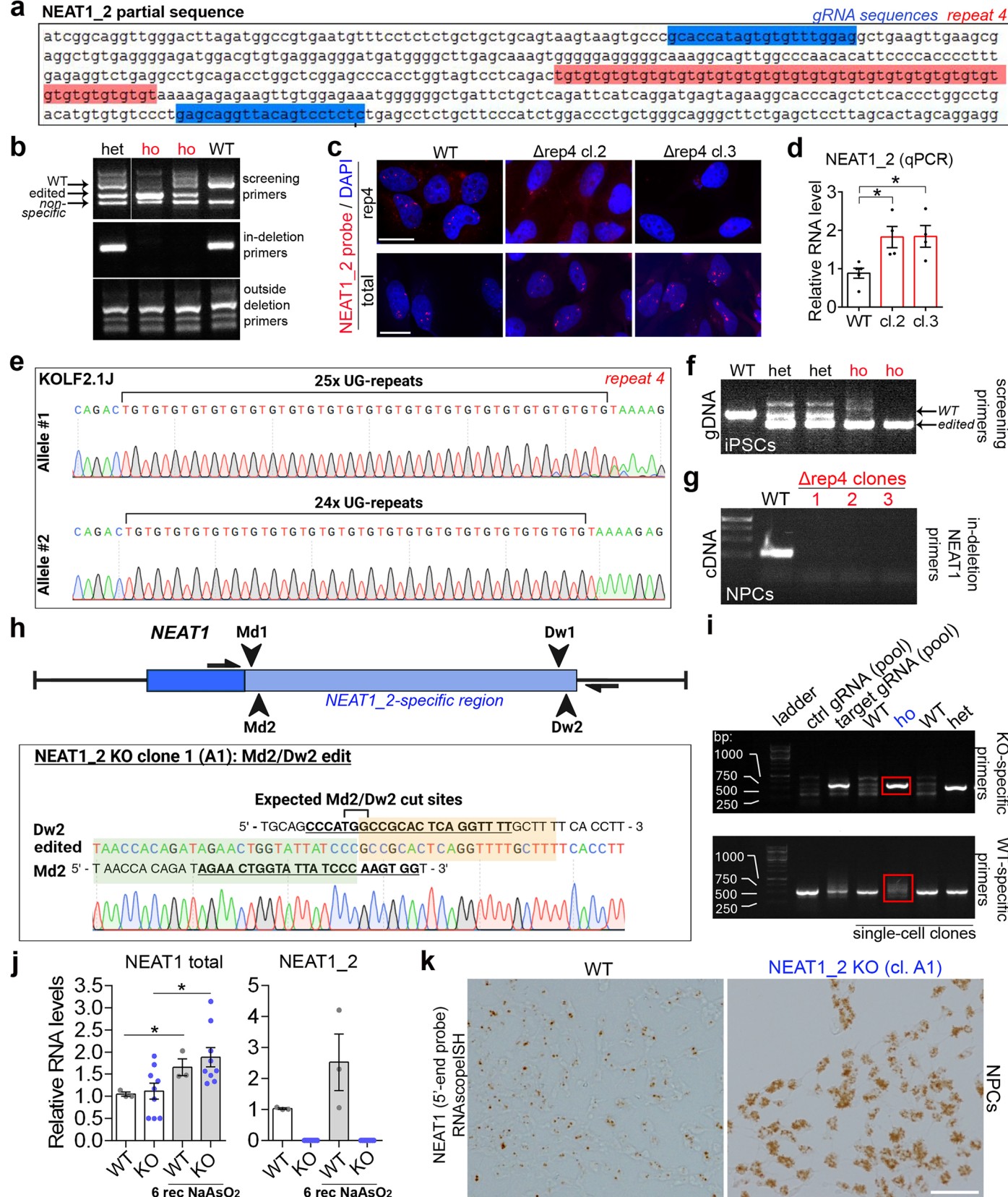

**Extended Data Fig. 10 | See next page for caption.**

**Extended Data Fig. 10 | Generation of cell lines with NEAT1_2 UG-repeat 4 deletion and NEAT1_2 knockout. a**) gRNAs used for repeat 4 deletion by CRISPR/ Cas9. **b**) PCR confirmation of repeat 4 deletion in HeLa cells. Screening primer pair was designed to flank the deleted region (top panel); 'in-deletion' primer pair has both primers located within the deleted region; 'outside deletion' primer pair maps to an unaffected region of NEAT1_2. Homozygous (ho, clones 2 and 3) and heterozygous (het, clone 1) Δrep4 clones were obtained. **c**) Lack of signal from the repeat 4-specific probe in Δrep4 HeLa cells. For probe positions, see Fig. 7d. **d**) NEAT1_2 is upregulated in cells with repeat 4 deleted, as analysed by qRT-PCR. Graph shows mean±s.e.m. N = 4-5, *p < 0.05 (p = 0.0254), Kruskal-Wallis with Dunn's post-hoc test. **e**) NEAT1_2 UG-repeat 4 sizes in KOLF2.1J iPSC line. **f**) PCR screening of single-cell derived iPSC clones for repeat 4 deletion after CRISPR/ Cas9 editing. **g**) Confirmation of repeat 4 deletion in iPSCs at the RNA level after NPC differentiation. cDNA was prepared on the RNA purified from Day 16 NPCs. **h**) NEAT1 locus editing for NEAT1_2 KO. Top: double targeting (two gRNAs per cut site) was used – in the middle (Md) and downstream (Dw) of NEAT1_2 genomic

sequence. Dark-blue – region corresponding to NEAT1_1. Bottom: desired editing was confirmed by sequencing of the edited junction in all clones. Positions of screening PCR primers are also shown. **i**) PCR screening of single-cell derived NEAT1_2 KO iPSC clones. 'Pool', nucleofected cell pool used for single-cell plating (analysed 72 h post-nucleofection).**j**) Confirmation of abolished NEAT1_2 expression and retained NEAT1_1 expression in NEAT1_2 KO iPSC clones. Day 16 NPCs were analysed under basal conditions and after 6 h of recovery from arsenite stress. Data for 3 clones were combined. Graph shows mean±s.e.m. N = 3 or 9 (N = 3 for each KO clone, combined); *p ≤ 0.05: p = 0.05 for unstressed and p = 0.0118 for stressed WT cells, one-tailed Mann-Whitney U test. **k**) Loss of paraspeckles in NEAT1_2 KO NPCs confirmed by RNAscope-ISH. Day 16 NPCs were analysed using the total NEAT1 probe. Note distinct foci corresponding to paraspeckles in WT cells and dispersed NEAT1_1 signal in NEAT1_2 KO cells. Representative images are shown. Scale bar, 50 μm. Source numerical data and unprocessed gels are available in the Source data.

# Reporting Summary

## Statistics

For all statistical analyses, confirm that the following items are present in the figure legend, table legend, main text, or Methods section.

| n/a | Confirmed | |
|---|---|---|
| ☐ | ☒ | The exact sample size (*n*) for each experimental group/condition, given as a discrete number and unit of measurement |
| ☐ | ☒ | A statement on whether measurements were taken from distinct samples or whether the same sample was measured repeatedly |
| ☐ | ☒ | The statistical test(s) used AND whether they are one- or two-sided<br>*Only common tests should be described solely by name; describe more complex techniques in the Methods section.* |
| ☒ | ☐ | A description of all covariates tested |
| ☐ | ☒ | A description of any assumptions or corrections, such as tests of normality and adjustment for multiple comparisons |
| ☐ | ☒ | A full description of the statistical parameters including central tendency (e.g. means) or other basic estimates (e.g. regression coefficient) AND variation (e.g. standard deviation) or associated estimates of uncertainty (e.g. confidence intervals) |
| ☐ | ☒ | For null hypothesis testing, the test statistic (e.g. *F*, *t*, *r*) with confidence intervals, effect sizes, degrees of freedom and *P* value noted<br>*Give P values as exact values whenever suitable.* |
| ☒ | ☐ | For Bayesian analysis, information on the choice of priors and Markov chain Monte Carlo settings |
| ☒ | ☐ | For hierarchical and complex designs, identification of the appropriate level for tests and full reporting of outcomes |
| ☒ | ☐ | Estimates of effect sizes (e.g. Cohen's *d*, Pearson's *r*), indicating how they were calculated |

*Our web collection on statistics for biologists contains articles on many of the points above.*

## Software and code

Policy information about availability of computer code

| Data collection | The following software was used for data collection: ZEN blue software (Zeiss); CellSens Dimension software (Olympus); Harmony 4.9 High-Content Imaging and Analysis Software (Revvity); Image Studio (LICORbio); NIS Elements v2.20.02 (Nikon); FIDA Software Suite (Fidabio); Pymol v.2.55.5; Image J 1.54m. |
|---|---|
| Data analysis | No original code was generated in this study. |

For manuscripts utilizing custom algorithms or software that are central to the research but not yet described in published literature, software must be made available to editors and reviewers. We strongly encourage code deposition in a community repository (e.g. GitHub). See the Nature Portfolio guidelines for submitting code & software for further information.

## Data

Policy information about availability of data

All manuscripts must include a data availability statement. This statement should provide the following information, where applicable:
- Accession codes, unique identifiers, or web links for publicly available datasets
- A description of any restrictions on data availability
- For clinical datasets or third party data, please ensure that the statement adheres to our policy

All data that are necessary to interpret, verify and extend the research in the article are provided in the main and extended figures, supplementary data or source files. Expression plasmids generated in the study are available from the Addgene repository: https://www.addgene.org/Tatyana_Shelkovnikova/ All other unique

## Research involving human participants, their data, or biological material

Policy information about studies with [human participants or human data](). See also policy information about [sex, gender (identity/presentation), and sexual orientation]() and [race, ethnicity and racism]().

| | |
|---|---|
| Reporting on sex and gender | Not applicable |
| Reporting on race, ethnicity, or other socially relevant groupings | Not applicable |
| Population characteristics | The 6,739 individuals including 4,996 ALS patients and 1,743 controls subject to WGS and included in this study were recruited at specialised neuromuscular centres in the UK, Belgium, Germany, Ireland, Italy, Spain, Turkey, the United States and the Netherlands. |
| Recruitment | Patients were diagnosed with possible, probable or definite ALS according to the 1994 El-Escorial criteria. All controls were free of neuromuscular diseases and matched for age, sex and geographical location. |
| Ethics oversight | Project MinE was approved by the Trent Research Ethics Committee 08/H0405/60. Informed consent for genetic research was obtained from all participants. |

Note that full information on the approval of the study protocol must also be provided in the manuscript.

# Field-specific reporting

Please select the one below that is the best fit for your research. If you are not sure, read the appropriate sections before making your selection.

☒ Life sciences ☐ Behavioural & social sciences ☐ Ecological, evolutionary & environmental sciences

For a reference copy of the document with all sections, see [nature.com/documents/nr-reporting-summary-flat.pdf]()

# Life sciences study design

All studies must disclose on these points even when the disclosure is negative.

| | |
|---|---|
| Sample size | RNA expression analysis (qRT-PCR): a minimum 3 biological replicates were analysed in the experiments with statistical significance analysis. Cell phenotype quantification studies: experiments were repeated at least 3 times, and quantification was done typically in at least 40 cells (for transient transfection, and typically more) per condition, from 10 fields of view in a representative experiment. Exceptions: in highly homogeneous populations, where nearly all cells showed the same phenotype, cell numbers were >20. Number of intracellular condensates analysed (e.g. paraspeckles) was typically >100. In vitro imaging experiments: experiments were repeated at least 3 times, and at least 4 fields of view were analysed from a representative experiment. Biochemical analysis (e.g. western blot) experiments were repeated at least 3 times (N=3). FRAP experiments: experiments were repeated at least 3 times, with 10 or more cells analysed in each experiment (2 or more particles in each cell). The numbers and biological and technical repeats were determined for each type of experiment based on prior extensive experience in this type of studies. |
| Data exclusions | No data exclusions have been made, except a single outlier in the human genetic dataset analysis. |
| Replication | Only data where replication was achieved in at least 3 experiments were used for conclusions and included in the manuscript. |
| Randomization | Not relevant - the experiments did not involve randomisation. |
| Blinding | Investigators were blinded to the condition in experiments requiring manual quantification, where possible. |

# Reporting for specific materials, systems and methods

We require information from authors about some types of materials, experimental systems and methods used in many studies. Here, indicate whether each material, system or method listed is relevant to your study. If you are not sure if a list item applies to your research, read the appropriate section before selecting a response.

## Materials & experimental systems

| n/a | Involved in the study |
|---|---|
| ☐ | ☒ Antibodies |
| ☐ | ☒ Eukaryotic cell lines |
| ☒ | ☐ Palaeontology and archaeology |
| ☒ | ☐ Animals and other organisms |
| ☒ | ☐ Clinical data |
| ☒ | ☐ Dual use research of concern |
| ☒ | ☐ Plants |

## Methods

| n/a | Involved in the study |
|---|---|
| ☒ | ☐ ChIP-seq |
| ☒ | ☐ Flow cytometry |
| ☒ | ☐ MRI-based neuroimaging |

# Antibodies

| | |
|---|---|
| Antibodies used | Antibody, Company, Catalogue number, and Lot number:<br>FUS (rabbit polyclonal) Proteintech Cat#11570-1-AP, 00107882<br>FUS (mouse monoclonal) Santa Cruz Cat#sc-47711, K1915<br>TDP-43 (rabbit polyclonal) C-terminal Sigma Cat# T1580, 0000193756<br>TDP-43 (mouse monoclonal) R&D Biosystems Cat# MAB7778, CHGW0123021<br>TDP-43 E2G6G (rabbit monoclonal) Cell Signaling Cat# #89718, Lot 1<br>mCherry (rabbit polyclonal) Proteintech Cat# 26765-1-AP, 00062779<br>GFP (rabbit polyclonal) Proteintech Cat# 50430-2-AP, 00150738<br>Tuj (Alexa®488-conjugated), Abcam, Cat# ab237350<br>Secondary fluorescently labelled antibodies: Alexa488/546/633 Fluor anti-mouse/rabbit IgG, ThermoFisher: Rabbit 488 #A-11008, 2897813; Mouse 488 #A-11001, 2551357; Mouse 546 #A-11030, 2465085; Rabbit 546, #A11010, 2902337; Mouse 633 #A21050, 2920646.<br>Mouse IgG HRP Linked Whole Ab Amersham Cat# NA931, 18095933<br>Rabbit IgG HRP Linked Whole Ab Amersham Cat# NA934, 18225602<br><br>Primary antibodies were used at 1:1000 dilution for immunocytochemistry and western blot and at 1:5000 in ImmuCon. Secondary antibodies were used at 1:1000 dilution. |
| Validation | Primary antibodies were validated in the required application prior to use: by western blot (single dominant band of predicted molecular weight) and ICC (expected/known subcellular distribution); select antibodies - by siRNA knockdown (and subsequent WB and ICC). |

# Eukaryotic cell lines

Policy information about cell lines and Sex and Gender in Research

| | |
|---|---|
| Cell line source(s) | Human: HeLa cells (ATCC), female, Merck, Cat# 93021013<br>Human: SH-SY5Y cells (ATCC), female, Merck, Cat# 94030304<br>Human: MCF7 cells (ECACC), female, Merck, Cat# 86012803<br>Human: FUS knockout SH-SY5Y cell line - the above parental line was used<br>Human: NEAT1 knockout HeLa cell line - the above parental line was used<br>Human: NEAT1_2 Δrep4 HeLa cell line - the above parental line was used<br>Human: 1xMS2-NEAT1_2 HeLa cell lines - the above parental line was used<br>Human: KOLF2.1J iPSC line, male, JAX, Cat#JIPSC001000<br>Human: NEAT1_2 Δrep4 iPCS line - the above parental line was used<br>Human: NEAT1_2 KO iPCS line - the above parental line was used |
| Authentication | The cell lines were procured directly from the approved repository vendor, with all the authentication documentation (e.g. STR profiling) available. |
| Mycoplasma contamination | Mycoplasma testing of all cell lines was performed routinely (every 2 weeks) using a sensitive luminescence kit; all cell lines stayed mycoplasma-free throughout the study. |
| Commonly misidentified lines<br>(See ICLAC register) | Such lines were not used in the study. |

## Plants

| | |
|---|---|
| Seed stocks | N/A |
| Novel plant genotypes | N/A |
| Authentication | N/A |

