## [Peer Review File · Nature Cell Biology]

Paraspeckle condensation is controlled via TDP-43 polymerisation and linked to neuroprotection

Corresponding Author: Dr Tatyana Shelkovich

Version 0:

Decision Letter:

*Please delete the link to your author homepage if you wish to forward this email to co-authors.

Dear Tatyana,

I hope you are well. We have now received Reviewer #3's report, which I have appended below. I'm sorry for the delay, and we do hope that you find the reviewer reports useful. Please do address this reviewers' concerns as well.

Best wishes,
Daryl

Dear Dr Shelkovich,

I apologize for the delay. Despite our best efforts, we have been unable to receive the comments of reviewer #3, who is an expert in ALS/FTD molecular mechanisms. If/when we do receive the missing comments, we will pass them on to you.

Please note that if/when we do receive the comments of the missing referee, we will pass them on to you; should we be in touch with the missing comments, your revised manuscript should address all of the referees' concerns including those sent subsequently.

Your manuscript, "TDP-43 is a master regulator of paraspeckle condensation", has now been seen by 2 referees, who are experts in biomolecular condensation including within the nucleus (referee 1); TDP-43 (referee 2). As you will see from their comments (attached below) they find this work of potential interest, but have raised substantial concerns, which in our view would need to be addressed with considerable revisions before we can consider publication in Nature Cell Biology.

Nature Cell Biology editors discuss the referee reports in detail within the editorial team, including the chief editor, to identify key referee points that should be addressed with priority, and requests that are overruled as being beyond the scope of the current study. To guide the scope of the revisions, I have listed these points below. We are committed to providing a fair and constructive peer-review process, so please feel free to contact me if you would like to discuss any of the referee comments further.

I should stress that the referees' concerns point to unclear mechanistic links which would need to be addressed with experiments and data, and reconsideration of the study for this journal and re-engagement of referees would depend on strength of these revisions.

In particular, it would be essential to:

A) Assess concerns about molecular mechanisms between RNA repeat length and ALS progression (Reviewer #1) and RNA – TDP-43 interaction (both Reviewers)

B) Address concerns about TDP-43 interactions and paraspeckle composition with additional experimental data (both Reviewers)

C) All other referee concerns pertaining to strengthening existing data, providing controls, methodological details, clarifications and textual changes, should also be addressed.

D) Finally please pay close attention to our guidelines on statistical and methodological reporting (listed below) as failure to do so may delay the reconsideration of the revised manuscript. In particular please provide:

We would be happy to consider a revised manuscript that would satisfactorily address these points, unless a similar paper is published elsewhere, or is accepted for publication in Nature Cell Biology in the meantime.

- ensure that it conforms to our format instructions and publication policies (see below and <https://www.nature.com/nature/for-authors>).

- provide a point-by-point rebuttal to the full referee reports verbatim, as provided at the end of this letter.

- provide the completed Reporting Summary (found here <https://www.nature.com/documents/nr-reporting-summary.pdf>). This is essential for reconsideration of the manuscript will be available to editors and referees in the event of peer review. For more information see <http://www.nature.com/authors/policies/availability.html> or contact me.

Nature Cell Biology is committed to improving transparency in authorship. As part of our efforts in this direction, we are now requesting that all authors identified as 'corresponding author' on published papers create and link their Open Researcher and Contributor Identifier (ORCID) with their account on the Manuscript Tracking System (MTS), prior to acceptance. ORCID helps the scientific community achieve unambiguous attribution of all scholarly contributions. You can create and link your ORCID from the home page of the MTS by clicking on 'Modify my Springer Nature account'. For more information please visit www.springernature.com/orcid.

This journal strongly supports public availability of data. Please place the data used in your paper into a public data repository, or alternatively, present the data as Supplementary Information. If data can only be shared on request, please explain why in your Data Availability Statement, and also in the correspondence with your editor. Please note that for some data types, deposition in a public repository is mandatory - more information on our data deposition policies and available repositories appears below.

Link Redacted

We would like to receive a revised submission within six months.

We hope that you will find our referees' comments, and editorial guidance helpful. Please do not hesitate to contact me if there is anything you would like to discuss.

Best wishes,

Daryl

Daryl Jason Verzosa David, PhD

Senior Editor, Nature Cell Biology
Advisory Editor, npj Biological Physics and Mechanics
Nature Portfolio

Heidelberger Platz 3, 14197 Berlin, Germany
Email: daryl.david@nature.com
ORCID: <https://orcid.org/0000-0002-9253-4805>

Reviewers' Comments:

Reviewer #1 (Remarks to the Author):

In the presented manuscript, Hodgson and colleagues analyze the inhibitory role of TDP-43, a multifunctional protein that regulates gene expression and RNA metabolism, in forming a nuclear structure called paraspeckles. Paraspeckles are scaffolded by the long non-coding RNA NEAT1_2, which interacts with a cohort of paraspeckle proteins. They showed that TDP-43 blocks the condensation of NEAT1-RNP particles into the paraspeckle via self-multimerization on the scaffolding NEAT1_2 RNA 3'-end UG-repeats. This act promotes the accumulation of smaller NEAT1_2 positive micro-condensates, which fail to fuse into mature functional paraspeckles. Depletion of TDP-43 increases the formation of larger paraspeckles, suggesting that TDP-43 limits paraspeckle maturation. This inhibitory effect of TDP-43 is counterbalanced by the core paraspeckle protein FUS, which restores paraspeckle formation. Furthermore, the 3'-end UG-repeat likely contributes to paraspeckle inhibitory regulation post-assembly as a recruitment scaffold for TDP-43. The authors speculate that NEAT1_2 UG polymorphism is associated with a neurodegenerative disease, amyotrophic lateral sclerosis (ALS). The length of the 3'-end UG repeat 4 is highly variable, and longer repeats might correlate with quicker ALS progression. This is an interesting manuscript that provides novel results indicating how the paraspeckle assembly is post-transcriptionally regulated. It suggests that paraspeckles can form only in specific cells and tissues, and their assembly and disassembly can be correlated with disease pathology.

My points of criticism are highlighted below:

The authors suggest that TDP-43 forms higher-order homomultimers by NTD-CTD self-oligomerization, organized into micro-condensates. They also indicated that TDP-43 tends to form aggregates even in the nucleus under pathological conditions. How different are these nuclear aggregates from TDP-43 microaggregates? The literature suggests they might contain Sirtuin 1, nucleoporins, and some transcription factors. The authors should check for other protein factors in these micro-condensates to exclude the possibility that they are heterogeneous multimers.

In many cases, cells require a mechanism to control the size of phase-separated structures that would otherwise grow without bounds. Several mechanisms for size limitation have previously been reported in the literature. Examples include the elastic energy of the cytoskeleton, kinetic limitations on coarsening, stoichiometric constraints, multiple nucleation sites, and micellization. It has been proposed by the Hirose lab that paraspeckles have a structure that diverges from that of phase separation-driven biomolecular condensates but resembles that of micelles. Micellization as a self-assembly process provides separate mechanisms for size control, but that control is limited to molecular dimensions. Thus, the excluded volume in the amphipathic shell of paraspeckles may provide a repulsive force that favors a curved surface and explain the paraspeckle assembly and physical appearance in normal and pathological conditions. The authors ignore the micellization phenomenon in their discussion. TDP-43 is not a prototypical amphipathic protein like a surfactant. It's a hydrophobic LCD domain, and the hydrophilic RRM domain is not clearly segregated as in micelle-forming molecules. Furthermore, the authors propose that TDP-43 micro-condensates are non-liquid, gel-like, differentiating them from dynamic micelles with fluid-like properties. If TDP-43 micro-condensates as aggregate multimers are part of the larger complex biomolecular condensate, this likely limits the previously proposed analogy to the formation of a micelle. This needs to be discussed and explained.

The authors claim that TDP-43 micro-condensates limit paraspeckle formation under normal conditions. This inhibitory regulatory step indicates that paraspeckles form only when they are needed. They should discuss the functional relevance of their formation in NEAT1_2 low-expressing cells, which limits their functional significance as a gene-regulatory structure. Are stress-induced paraspeckles, when TDP-43 is sequestered, the same as paraspeckles formed in NEAT1_2 low-expressing cells?

I am not convinced that TDP-43 micro-condensates are an exclusive paraspeckle subcomponent of the shell with an XY SIM resolution of 110-120 nm. For example, it has been shown that some fractions of BRG1 localize in the shell and HNRNPF. Please provide super-resolution colocalization images of TDP-43 with BRG1 and HNRNPF in the core-shell configuration in HeLa cells using STED with 50 nm resolution.

The authors claim that droplets were sensitive to 1,6-hexanediol, confirming their LLPS nature. 1,6-hexanediol is often employed to disrupt weak hydrophobic interactions. However, it has been established that sensitivity to hexanediol is insufficient to demonstrate unequivocally that the effect studied here is via LLPS. Thus, hexanediol is not a quantifiable parameter to map onto phase separation behavior, and the authors cannot claim that observed changes caused by

hexanediol are due to phase separation.

It has been shown that the level and localization of TDP-43 are finely regulated via an autoregulatory negative feedback mechanism (doi.org/10.1038/emboj.2010.310). TDP-43 binds to 3' UTR sequences in its own mRNA, downregulating TDP-43 transcript levels via an exosome-mediated pathway of mRNA degradation. However, in pathological conditions, this autoregulatory mechanism might be impaired. Therefore, the exogenous overexpression of TDP-43 in normal HeLa cells should be critically evaluated, and the elevation of TDP-43 in neurodegenerative conditions should be assessed at the TDP-43 mRNA levels.

The authors extensively analyzed the opposite effect of TDP-43 to FUS in paraspeckle condensation in cells and using an in vitro droplet assay. Furthermore, they localized NONO in the paraspeckle core and the rescued paraspeckle assembly under TDP-43 overexpression. They showed that NONO can fully rescue paraspeckle assembly in cells with FUS KO expression. Since FUS and NONO are both present in the core of paraspeckles and play essential roles in standard paraspeckle assembly, it would be valuable to check whether TDP-43's prevention of forming paraspeckles is identical to that of other paraspeckle factors.

The authors primarily used aneuploid HeLa cells with very high NEAT_2 expression levels (likely due to triploidy of the 11q13.1 NEAT1 locus). The use of SH-SY5Y as a neuronal model relevant to ALS is quite limited. The authors should use at least one other neuronal-specific cell line for ALS to confirm the results.

Based on analysis of whole genome sequencing data, the authors claim that the longer 3'-end UG repeats (repeat 4) of NEAT1_2 are associated with shorter survival in a neurodegenerative disease, amyotrophic lateral sclerosis (ALS). Their model hypothesizes that longer repeats 4 can increase TDP-43 recruitment and sequestration, compromising paraspeckle homeostasis and, consequently, their cytoprotection under stress, leading to more severe ALS disease. I believe the link between the length of repeat 4 and ALS survival is not experimentally validated. The manuscript does not provide functional evidence on how longer repeats 4 impair paraspeckle functions to impact ALS progression directly. Given the complexity of ALS pathology with many contributing factors, this paraspeckle model linked to ALS is preliminary and can be overstated.

Reviewer #2 (Remarks to the Author):

This study addresses important questions concerning the regulation of TDP-43 function and the biology of RNP granules under stress—areas that remain largely undefined. There is value in investigating how RNA-protein interactions involve condensate properties that may be mediated by homotypic and heterotypic interactions. These are noted strengths of the manuscript. However, there are major concerns regarding the experimental design that limit the interpretability of the findings. Foremost among these is the reliance on cellular overexpression of TDP-43 and FUS constructs, which can significantly alter their self-assembly behavior, condensate properties, and aggregation propensity. This in addition to the levels of overexpression of WT and mutant TDP-43 may not be physiological or pathologically-relevant. Transient transfection of these constructs typically results in widely heterogeneous expression levels complicating the interpretation of condensate formation, subcellular localization, and protein interactions. In addition, there are important questions regarding the design and interpretation of the ImmunoCon assay developed here to study phase separation of the purified proteins, which require clarification.

Concerns:

Page 3, line 87. Relative Neat1_2 transcript expression upon TDP-43 OE should be quantified to determine whether changes in Neat 1_2 body size are caused by changes in levels of the transcript.

Page 4, line 144. Experiments using TDP-43 fused to different tags and the sentence describing “saturating the autoregulatory mechanism” is unclear. There is no quantification of autoregulation and it is unclear whether the authors refer to autoregulation of endogenous TDP-43 or the GFP and SNAP constructs, in which case the TARDBP 3'UTR should have been included in the expression construct.

Page 4, line 155. Perez-Berlanga et al. (2023) and Dos Passos et al. (2023) showed that multivalent TDP-43 interactions promote the formation of macromolecular TDP-43 complexes and these should be included as reference. Importantly, these studies and others, including work from the Hayes group, demonstrate that disruption of RNA binding, NTD-mediated oligomerization and CTD-driven self-assembly increases TDP-43 exit to the cytoplasm. In evaluating the effect of these mutations in paraspeckle formation, the studies and their conclusions should take into consideration their diminished TDP-43 nuclear localization.

Fig. 4. The assays used to study TDP-43 and FUS phase separation, Immuncon, lack important controls to determine whether they may be used for studying phase separation properties and behavior. The experiments should include conditions, such as site directed mutations—and preferably not large deletion constructs—previously established to disrupt TDP-43 and FUS separation. There are several questions raised by the method design and observations. The assay does not distinguish between aggregate accumulation and condensate formation. In addition, the complexes identified by antibody detection are significantly smaller than those reported in multiple studies for FUS and TDP-43 condensates. Furthermore, antibody binding to these proteins may disrupt phase separation depending on the epitopes these interact with. Similar experiments should be conducted in antibody-free conditions to demonstrate that the antibody does not interfere with

protein assembly.

TDP-43 phase separation in the presence of GU-repeat RNA (as shown in Fig. 4I) was previously reported by Grese et al., 2021, who observed a dramatic increase in the size of TDP-43 condensates. The authors should include a discussion addressing the potential reasons for the differences between their findings and those of Grese et al.

Fig 5B, PLA experiments to study TDP-43-FUS interactions should include controls of FUS and TDP-43 KD to eliminate concerns of background non-specific signal.

Co-IP experiments to determine domain-specific interactions between TDP-43 and FUS should be performed in the absence of protein overexpression.

The experiments relying on transient expression of GFP-TDP-43 (WT and mutant variants) throughout the manuscript should be validated using assays in which these proteins are expressed at levels comparable to endogenous TDP-43. Transient transfection introduces substantial expression heterogeneity, which can confound interpretation of results related to condensate formation, localization, and protein interactions. To address this, the authors should employ stable cell lines with inducible or constitutive expression of GFP-TDP-43 at near-endogenous levels. Such systems have been successfully used to investigate TDP-43 phase separation and function, as shown by Hallegger et al., 2021.

Reviewer #3 (Remarks to the Author):

The study by Hodgson and colleagues presents important and timely findings supporting a novel and unexpected role for TDP-43 in regulating the formation of paraspeckle spheroids. The latter are enigmatic nuclear structures with unclear functions in gene regulation. The study provides valuable insights into nuclear phase separation and the functional dynamics of TDP-43. Importantly, the work provides genetic evidence that the uncovered mechanism of TDP-43-dependent paraspeckle formation correlates with ALS disease progression, an exciting discovery, opening new avenues for investigation and potentially interfering with TDP-43 toxicity in disease, beyond its well-characterized function in suppressing cryptic exons, which is lost in neurodegeneration. Given the significant contribution of the study to the field of RNA biology and neurodegeneration, I enthusiastically recommend it for publication in *Nature Cell Biology*, after the authors address the following points of criticism.

1. The manuscript does not clearly state whether the GFP tag on TDP-43 is placed at the N- or C-terminus. This information is crucial for interpreting the localization and behavior of the fusion protein and should be specified not only in the Methods section but also in the Results.
2. The image analysis presented throughout the study is elegant and enhances the quantitative interpretation of the results. However, in Figure 1G, the same analytical approach should be applied to the knockdown (KD) condition as is done for the overexpression (OE) condition. This would facilitate a more balanced and informative comparison between the two conditions.
3. The findings of the current study should be compared to those from Perez-Berlanga et al. *EMBO Journal*, 2023, which demonstrates that RNA-deficient TDP-43 forms smaller and aberrant LLPS particles in the nucleus (using GFP-TDP-43 in HEK cells, similarly as in the current study). Furthermore, Perez-Berlanga et al. showed that monomeric TDP-43, carrying the same (6M) mutations fails to form any droplets—except for the occasional association with Cajal bodies. Can the authors explain the discrepancy in their findings? These comparisons, similarities, and differences between the studies should be acknowledged and critically discussed in the Discussion section.
4. The claim that the 2M variant behaves like a monomer lacks sufficient supporting evidence. The authors should provide experimental validation or reference prior work that supports this assertion.
5. The manuscript notes that ALS-linked mutations (Q331K and M337V) do not prevent paraspeckle dispersal, which is intriguing. While this is stated, the data are not shown. Including these results, even in supplementary figures, would significantly strengthen the manuscript.
6. The study would benefit from a deeper exploration of the functional consequences of paraspeckle dispersal. Specifically, is there any evidence of cellular toxicity or other phenotypic effects that result from the altered phase behavior of TDP-43? This is particularly important given the potential role of these findings for neurodegenerative diseases.
7. The authors should clarify how they distinguish between phase-separated droplets and aggregates in the ImmuCon assay. Are there specific markers or dynamic properties that were used to make this distinction?
8. The formation of the multiphase structures within paraspeckle spheroids is elegant and intriguing. What is the role of these structures in cells?
9. Given the proposed role of the described mechanism for ALS, the authors should perform some of their key experiments in neurons. While I understand the potential hesitation of the authors to do this, due to consideration of technical limitations in culturing neurons, I can assure them that the widely used NGN2 human neurons are easy to obtain and can be differentiated

rapidly, allowing results within a short amount of time (within a few weeks).

10. The annotation of the mutations in the schematic of Figure 2A is not intuitive. It would be helpful to adopt the same labeling style used in Figure 2E, where mutations are annotated directly above or below the domain lines to indicate their positions.

Methods should be written concisely, but should contain all elements necessary to allow interpretation and replication of the results. As a guideline, Methods sections typically do not exceed 3,000 words. The Methods should be divided into subsections listing reagents and techniques. When citing previous methods, accurate references should be provided and any alterations should be noted. Information must be provided about: antibody dilutions, company names, catalogue numbers and clone numbers for monoclonal antibodies; sequences of RNAi and cDNA probes/primers or company names and catalogue numbers if reagents are commercial; cell line names, sources and information on cell line identity and authentication. Animal studies and experiments involving human subjects must be reported in detail, identifying the committees approving the protocols. For studies involving human subjects/samples, a statement must be included confirming that informed consent was obtained. Statistical analyses and information on the reproducibility of experimental results should be provided in a section titled "Statistics and Reproducibility".

All Nature Cell Biology manuscripts submitted on or after March 21 2016 must include a Data availability statement as a separate section after Methods but before references, under the heading "Data Availability". For Springer Nature policies on data availability see <http://www.nature.com/authors/policies/availability.html>; for more information on this particular policy see <http://www.nature.com/authors/policies/data/data-availability-statements-data-citations.pdf>. The Data availability statement should include:

- Accession codes for primary datasets (generated during the study under consideration and designated as "primary accessions") and secondary datasets (published datasets reanalysed during the study under consideration, designated as "referenced accessions"). For primary accessions data should be made public to coincide with publication of the manuscript. A list of data types for which submission to community-endorsed public repositories is mandated (including sequence, structure, microarray, deep sequencing data) can be found here <http://www.nature.com/authors/policies/availability.html#data>.
- Unique identifiers (accession codes, DOIs or other unique persistent identifier) and hyperlinks for datasets deposited in an approved repository, but for which data deposition is not mandated (see here for details <http://www.nature.com/sdata/data-policies/repositories>).
- At a minimum, please include a statement confirming that all relevant data are available from the authors, and/or are included with the manuscript (e.g. as source data or supplementary information), listing which data are included (e.g. by figure panels and data types) and mentioning any restrictions on availability.
- If a dataset has a Digital Object Identifier (DOI) as its unique identifier, we strongly encourage including this in the Reference list and citing the dataset in the Methods.

We recommend that you upload the step-by-step protocols used in this manuscript to [protocols.io](https://www.protocols.io). More details can be found at <https://www.protocols.io/help/publish-articles>.

All imaging data should be accompanied by scale bars, which should be defined in the legend.

Cropped images of gels/blots are acceptable, but need to be accompanied by size markers, and to retain visible background signal within the linear range (i.e. should not be saturated). The boundaries of panels with low background have to be demarked with black lines. Splicing of panels should only be considered if unavoidable, and must be clearly marked on the figure, and noted in the legend with a statement on whether the samples were obtained and processed simultaneously. Quantitative comparisons between samples on different gels/blots are discouraged; if this is unavoidable, it should only be performed for samples derived from the same experiment with gels/blots were processed in parallel, which needs to be stated in the legend.

EXTENDED DATA FIGURES - When re-submitting your manuscript, please ensure that any supplementary figures and tables that are crucial to the manuscript's conclusions are converted into Extended Data figures and tables to increase visibility of these data. Extended Data figures and tables are online-only (present in the online PDF and full-text HTML versions of the paper), peer-reviewed display items that provide essential background to the article but are not included in the main article due to space constraints. A maximum of ten Extended Data display items (figures and tables) is permitted.

The total number of Supplementary Figures (not including the "unprocessed scans" Supplementary Figure) should not exceed the number of main display items (figures and/or tables) (see our Guide to Authors and March 2012 editorial <http://www.nature.com/ncb/authors/submit/index.html#suppinfo>; <http://www.nature.com/nbc/journal/v14/n3/index.html#ed>). No restrictions apply to Supplementary Tables or Videos, but we advise authors to be selective in including supplemental data.

GUIDELINES FOR EXPERIMENTAL AND STATISTICAL REPORTING

REPORTING REQUIREMENTS – We are trying to improve the quality of methods and statistics reporting in our papers. To

that end, we are now asking authors to complete a reporting summary that collects information on experimental design and reagents. The Reporting Summary can be found here <https://www.nature.com/documents/nr-reporting-summary.pdf>) If you would like to reference the guidance text as you complete the template, please access these flattened versions at <http://www.nature.com/authors/policies/availability.html>.

Version 1:

Decision Letter:

Our ref: NCB-A57397A

3rd December 2025

Dear Dr. Shelkovnikova,

Thank you for submitting your revised manuscript "Paraspeckle condensation is controlled via TDP-43 polymerisation and linked to neuroprotection" (NCB-A57397A). It has now been seen by the original referees and their comments are below. The reviewers find that the paper has improved in revision, and therefore we'll be happy in principle to publish it in Nature Cell Biology, pending minor revisions to satisfy the referees' final requests and to comply with our editorial and formatting guidelines.

In your revisions, please add textual edits to note further caveats and discussion to address all remaining concerns of Reviewers #1 and #2.

Thank you again for your interest in Nature Cell Biology Please do not hesitate to contact me if you have any questions.

Sincerely,
Daryl

Daryl Jason Verzosa David, PhD

Senior Editor, Nature Cell Biology
Advisory Editor, npj Biological Physics and Mechanics
Nature Portfolio

Heidelberger Platz 3, 14197 Berlin, Germany
Email: daryl.david@nature.com
ORCID: <https://orcid.org/0000-0002-9253-4805>

Reviewer #1 (Remarks to the Author):

I have re-evaluated the manuscript in light of the authors' rebuttal and newly presented data. The revisions have substantially strengthened the study, adding conceptual clarity and experimental depth. However, a few areas remain underdeveloped—particularly the mechanistic integration of TDP-43 clusters within the micellization framework and the subtle differences between basal and stress-induced paraspeckles.

1. The revised Discussion now appropriately presents TDP-43 as a modulator of paraspeckle micellization rather than a direct surfactant, providing a more precise and accurate interpretation of its role. This section would, however, benefit from a clearer mechanistic link between TDP-43's gel-like (non-liquid) clusters and the physical parameters of micellization, specifically, how such rigid subphases might influence paraspeckle fluidity, curvature, or size criticality. Please expand the Discussion with one to two sentences addressing this connection, supported by relevant recent literature, to add conceptual depth and avoid the impression of a superficial alignment with the micelle model.
2. The authors' clarification of terminology ("nuclear clusters" versus "stress-induced de novo nuclear condensates of TDP-43 -TCs") and the negative enrichment data for Sirt1 and nucleoporins (Figure R1) effectively differentiate these structures. The results support the interpretation that TDP-43 clusters are relatively homogeneous; however, given that full-length TDP-43 forms gel-like condensates with restricted dynamics, it remains plausible that stress-responsive chaperones (e.g., HSP70) could be incorporated into paraspeckle-embedded clusters. To address this point more completely, please include Figure R1 in the Extended Data and briefly discuss this possibility in the main text.
3. The authors' observation that stress-induced and basal paraspeckles exhibit no significant differences in size or core suborganization is reasonable, and a detailed comparative study would indeed be beyond the current scope of the manuscript. To enhance broader relevance, however, it would be helpful to briefly discuss potential implications for cytoprotective function, particularly in light of recent findings (PMID: 40661327) that NEAT1 depletion coincides with TDP-43 loss in amyotrophic lateral sclerosis motor neurons, potentially affecting stress responses. Please add one or two sentences to the Discussion highlighting this link as a possible future direction.

Reviewer #2 (Remarks to the Author):

The authors carried out a comprehensive investigation of the interactions and regulation of paraspeckle formation by TDP-43 that is mediated by RNA binding and changes in phase separation properties. The revised manuscript is greatly improved with new experiments and discussion in response to the reviewers' comments. Still, one important limitation is the use of large tags and overexpression systems for many of the assays presented. These two modifications can significantly impact the interpretation of TDP-43 studies. This should be mentioned in the report.

A possible explanation for the differences observed in mutant TDP-43 nuclear-cytoplasmic ratio between these studies (Extended Data Fig. 3) and previous reports should be included in the manuscript. The authors should also consider differences in cellular distribution due to the expression of fusion proteins that are considerably larger than untagged protein or short peptide tags.

Furthermore, the authors should clarify in the manuscript whether monomeric GFP, or non-modified GFP was used as protein tag. This is important because GFP molecules are naturally dimers, unless the monomeric variant is used. This difference may alter tagged protein nuclear-cytoplasmic distribution, phase separation and cellular dynamics.

Reviewer #3 (Remarks to the Author):

The authors have successfully addressed all my concerns and, as far as I can tell, also those from the other referees. I think the addition of new experiments, especially the demonstration of the stress-induced death in human motor neurons, strengthens the study and its conclusions. Overall, I think the work elucidates a previously unrecognized role of TDP-43 in paraspeckle formation and dynamics that because functionally important upon cellular stress, and has clear implications for understanding disease mechanisms linked to ALS and related neurodegenerative diseases. It also points to a potential mechanism of toxicity linked to TDP-43 overexpression that has long been used to model TDP-43 proteinopathies in cellular and animal models. Therefore, I think the work is highly relevant and appropriate for publication in Nature Cell Biology.

Version 2:

Decision Letter:

Dear Dr Shelkovich,

I am pleased to inform you that your manuscript, "Paraspeckle condensation is controlled via TDP-43 polymerisation and linked to neuroprotection", has now been accepted for publication in *Nature Cell Biology*. Congratulations!

Over the next few weeks, your paper will be copyedited to ensure that it conforms to *Nature Cell Biology* style. Once your paper is typeset, you will receive an email with a link to choose the appropriate publishing options for your paper and our Author Services team will be in touch regarding any additional information that may be required.

Publication is conditional on the manuscript not being published elsewhere and on there being no announcement of this work to any media outlet until the online publication date in *Nature Cell Biology*.

Please note that *Nature Cell Biology* is a Transformative Journal (TJ). Authors may publish their research with us through the traditional subscription access route or make their paper immediately open access through payment of an article-processing charge (APC). Authors will not be required to make a final decision about access to their article until it has been accepted. <https://www.springernature.com/gp/open-research/transformative-journals> Find out more about Transformative Journals

Authors may need to take specific actions to achieve compliance with funder and institutional open access mandates. If your research is supported by a funder that requires immediate open access (e.g. according to <https://www.springernature.com/gp/open-science/plan-s-compliance> Plan S principles or the <https://www.springernature.com/gp/open-science/us-federal-agency-compliance> NIH public access policy) then you should select the gold OA route, and we will direct you to the compliant route where possible. Because authors warrant under our subscription licensing terms that they haven't committed to licensing any version of their article under a licence inconsistent with the terms of our agreement – including the applicable embargo period – publication under the subscription model isn't suitable for authors whose funders require no embargo.

If you have not already done so, we strongly recommend that you upload the step-by-step protocols used in this manuscript

to protocols.io (<https://protocols.io>), an open online resource that allows researchers to share their detailed experimental know-how. All uploaded protocols are made freely available and are assigned DOIs for ease of citation. Protocols and Nature Portfolio journal papers in which they are used can be linked to one another, and this link is clearly and prominently visible in the online versions of both. Authors who performed the specific experiments can act as primary authors for the Protocol as they will be best placed to share the methodology details, but the Corresponding Author of the present research paper should be included as one of the authors. By uploading your Protocols onto protocols.io, you are enabling researchers to more readily reproduce or adapt the methodology you use, as well as increasing the visibility of your protocols and papers. You can also establish a dedicated workspace to collect your lab Protocols. Further information can be found at <https://www.protocols.io/help/publish-articles>.

Nature Cell Biology encourages authors presenting evidence for cell, biological, molecular, and genetic interactions to consider communicating these findings using Biofactoid (<https://biofactoid.org/>). This tool helps users share a searchable representation of interactions (e.g. binding, gene expression, post-translational modification) between genes, gene products, or chemicals. Information added to Biofactoid, with author attribution, is shared on social media and public databases, such as Pathway Commons, where it can be discovered and analyzed in the context of a large and growing corpus of knowledge.

With kind regards,

Sabrya Carim, PhD
(she/her/hers)
Senior Editor, Nature Cell Biology
Nature Portfolio

Springer Nature
The Campus, 4 Crinan Street, London N1 9XW, UK
sabrya.carim@springernature.com
<https://orcid.org/0000-0001-9485-1938>

** Visit the Springer Nature Editorial and Publishing website at http://editorial-jobs.springernature.com?utm_source=ejp_NCB_email&utm_medium=ejp_NCB_email&utm_campaign=ejp_NCB for more information about our career opportunities. If you have any questions please click [here](mailto:editorial.publishing.jobs@springernature.com).

We are grateful to the anonymous Reviewers for taking time to carefully read and comment on the manuscript and their constructive feedback that allowed us to improve the manuscript. We have now addressed all of the points raised, by providing additional, extensive experimental data, as outlined in the point-by-point response below.

In brief, the study has been expanded and enhanced in the following ways:

1. It has been demonstrated that TDP-43's negative effect on paraspeckles is *unique among paraspeckle proteins* including the shell components.
2. TDP-43 role in paraspeckle regulation has been characterised in further detail, including the role of oligomerisation and interactions with FUS.
2. TDP-43 regulation of paraspeckles has been validated in *neuronal models*, including human iPSC-derived neurons.
3. NEAT1_2 UG-repeat 4 role in *neuronal survival under stress* has been confirmed in newly generated iPSC-derived models.

Reviewer #1.

The authors suggest that TDP-43 forms higher-order homomultimers by NTD-CTD self-oligomerization, organized into micro-condensates. They also indicated that TDP-43 tends to form aggregates even in the nucleus under pathological conditions. How different are these nuclear aggregates from TDP-43 microaggregates? The literature suggests they might contain Sirtuin 1, nucleoporins, and some transcription factors. The authors should check for other protein factors in these micro-condensates to exclude the possibility that they are heterogeneous multimers.

We appreciate this comment about possible overlap between TDP-43 clusters within the paraspeckle shell and stress-induced *de novo* nuclear condensates of TDP-43 (TCs). Of note, we termed these structures *TDP-43 clusters* throughout the manuscript, to distinguish them from TCs and dynamic condensates such as those formed by FUS and unify the cellular and *in vitro* terminologies. From our previous proteomic analysis of TCs (PMID 38941189), these structures have relatively low complexity with only a small subset of other proteins enriched, for example, NONO, hnRNPM and hnRNPC. NONO is not a component of TDP-43 clusters in the paraspeckle shell since it exclusively localises in the paraspeckle core (Fig. 5a). We did examine localisation of Sirt1 and several nucleoporins (e.g. Nup107 and Nup98), however these proteins did not show enrichment in paraspeckles (Figure R1 – for Reviewer only). Our SRM studies for two other shell-localised proteins demonstrated that, unlike TDP-43, they do not overlap with the 5'-end NEAT1 probe signal (Extended Data Fig. 9b). However, we would like to emphasise that TDP-43 clusters within paraspeckles may include other protein components and might not be homogenous, TDP-43-only assemblies.

Figure R1. Proteins reported to be recruited into *de novo* TDP-43 stress-induced condensates (TCs) are not enriched in paraspeckles. Arrows indicate a paraspeckle cluster. Scale bar, 20 μm .

In many cases, cells require a mechanism to control the size of phase-separated structures that would otherwise grow without bounds. Several mechanisms for size limitation have previously been reported in the literature. Examples include the elastic energy of the cytoskeleton, kinetic limitations on coarsening, stoichiometric constraints, multiple nucleation sites, and micellization. It has been proposed by the Hirose lab that paraspeckles have a structure that diverges from that of phase separation-driven biomolecular condensates but resembles that of micelles. Micellization as a self-assembly process provides separate mechanisms for size control, but that control is limited to molecular dimensions. Thus, the excluded volume in the amphipathic shell of paraspeckles may provide a repulsive force that favors a curved surface and explain the paraspeckle assembly and physical appearance in normal and pathological conditions. The authors ignore the micellization phenomenon in their discussion. TDP-43 is not a prototypical amphipathic protein like a surfactant. It's a hydrophobic LCD domain, and the hydrophilic RRM domain is not clearly segregated as in micelle-forming molecules. Furthermore, the authors propose that TDP-43 micro-condensates are non-liquid, gel-like, differentiating them from dynamic micelles with fluid-like properties. If TDP-43 micro-condensates as aggregate multimers are part of the larger complex biomolecular condensate, this likely limits the previously proposed analogy to the formation of a micelle. This needs to be discussed and explained.

Our original version of the Discussion did highlight micellization as a proposed mechanism of paraspeckle assembly: “It is possible that TDP-43 has a role in overcoming the repulsive forces on the spheroid surface, as per the micellization model⁶⁸”. Since currently we do not have experimental data on TDP-43 effect on micellization, in the revised version, we have modified this statement: “TDP-43 micro-condensates within the spheroid likely modulate micellization-dependent paraspeckle assembly⁵¹ and paraspeckle autonomy⁵².” TDP-43 clusters “embedded” in the dynamic micelle structures may modify the micelle properties, and we believe that their formation is compatible with the micelle model. In this regard, TDP-43 molecules *per se* may not act as a surfactant – instead, TDP-43 clusters modulate micelle properties through mechanisms different from those of individual TDP-43 molecules.

The authors claim that TDP-43 micro-condensates limit paraspeckle formation under normal conditions. This inhibitory regulatory step indicates that paraspeckles form only when they are needed. They should discuss the functional relevance of their formation in NEAT1_2 low-expressing cells, which limits their functional significance as a gene-regulatory structure.

We would like to clarify that paraspeckle assembly (spheroid condensation) is inhibited by TDP-43 recruitment onto NEAT1_2 co-transcriptionally and not by TDP-43 clusters *per se* (once they formed, they are embedded in the paraspeckle spheroid and may regulate spheroid properties). When NEAT1_2 expression is low (the majority of cells *in vivo*), low availability of NEAT1_2 itself is a limiting factor for paraspeckle assembly. However, when NEAT1_2 becomes upregulated (e.g. in stress and differentiation), TDP-43 acts to regulate the paraspeckle spheroid – precursor equilibrium. Since the differential properties of these two classes of NEAT1-containing RNP particles are unknown, it is not possible to predict the exact effects on gene expression regulation with our current knowledge.

Are stress-induced paraspeckles, when TDP-43 is sequestered, the same as paraspeckles formed in NEAT1_2 low-expressing cells?

The Reviewer may be referring to NEAT1_2 high-expressing cells in the second part of the question (*i.e.* paraspeckle assembly is high under stress; are paraspeckle properties same in cells with high constitutive paraspeckle formation and in cells with low paraspeckles but experiencing stress?). We have studied possible structural differences between “basal” and stress-induced paraspeckles. However, we did not observe detectable differences in the paraspeckle spheroid size or core size, for example. An in-depth comparative analysis of more subtle differences would represent a separate study, beyond the scope of this manuscript.

I am not convinced that TDP-43 micro-condensates are an exclusive paraspeckle subcomponent of the shell with an XY SIM resolution of 110-120 nm. For example, it has been shown that some fractions of BRG1 localize in the shell and HNRNPF. Please provide super-resolution colocalization images of TDP-43 with BRG1 and HNRNPF in the core-shell configuration in HeLa cells using STED with 50 nm resolution.

Contrary to previous reports, we do not observe BRG1 partitioning into paraspeckles (either ectopically expressed or endogenous) (e.g. Extended Data Fig. 1b). We therefore analysed two other shell proteins, RBM14 and hnRNPF. We found that Airyscan SRM at a ~100 nm resolution is in fact sufficient to perform co-localisation analysis with individual TDP-43 clusters. Firstly, we found that, unlike TDP-43, RBM14 co-localises with 5'-end of NEAT1 and that hnRNPF forms smaller patches throughout the paraspeckle (Extended Data Fig. 9b). Both proteins were not enriched in TDP-43 clusters in the shell, forming independent clusters (Extended Data Fig. 9c). This supports that TDP-43 sub-domains in the paraspeckles may be homogeneous, however, as stated above, given their size and inability to purify these structures, it cannot be fully addressed at present. Therefore, in the manuscript, we avoided making claims about the homogeneity of TDP-43 clusters in the shell.

The authors claim that droplets were sensitive to 1,6-hexanediol, confirming their LLPS nature. 1,6-hexanediol is often employed to disrupt weak hydrophobic interactions. However, it has been established that sensitivity to hexanediol is insufficient to demonstrate unequivocally that the effect studied here is via LLPS. Thus, hexanediol is not a quantifiable parameter to map onto phase separation behavior, and the authors cannot claim that observed changes caused by hexanediol are due to phase separation.

We agree that additional verification of condensate properties in ImmuCon would be beneficial. We have now confirmed the sensitivity of FUS condensates to high salt and their re-entrant phase behaviour (Extended Data Fig. 7e). Furthermore, we characterised recombinant TDP-43 aggregates using molecular chaperones for TDP-43, bis-ANS and TMAO, both of which were reported to promote TDP-43 phase separation (PMID 30520303, 33149109). Both compounds did not affect TDP-43 aggregates in ImmuCon, corroborating that they are not LLPS assemblies (Extended Data Fig. 7f).

It has been shown that the level and localization of TDP-43 are finely regulated via an autoregulatory negative feedback mechanism (doi.org/10.1038/emboj.2010.310). TDP-43 binds to 3' UTR sequences in its own mRNA, downregulating TDP-43 transcript levels via an exosome-mediated pathway of mRNA degradation. However, in pathological conditions, this autoregulatory mechanism might be impaired. Therefore, the exogenous overexpression of TDP-43 in normal HeLa cells should be critically evaluated, and the

elevation of TDP-43 in neurodegenerative conditions should be assessed at the TDP-43 mRNA levels.

We agree that TDP-43 autoregulation and its possible disruption in disease is an important aspect of its regulation. Unfortunately, data of TDP-43 autoregulation in human tissue is very limited – we are aware of only one study that reported it (e.g. PMID 17569064). In our expression vectors, TDP-43 ORF lacking regulatory sequences was used, and its OE was confirmed at the mRNA level at least for TDP-43 GFP (Extended Data Fig. 5d), in addition to western blot for this and other constructs (Extended Data Fig. 1e and 3a).

The authors extensively analyzed the opposite effect of TDP-43 to FUS in paraspeckle condensation in cells and using an in vitro droplet assay. Furthermore, they localized NONO in the paraspeckle core and the rescued paraspeckle assembly under TDP-43 overexpression. They showed that NONO can fully rescue paraspeckle assembly in cells with FUS KO expression. Since FUS and NONO are both present in the core of paraspeckles and play essential roles in standard paraspeckle assembly, it would be valuable to check whether TDP-43's prevention of forming paraspeckles is identical to that of other paraspeckle factors.

We thank the Reviewer for this suggestion. We have now studied the effect of three other shell proteins, RBM14, hnRNP and BRG1 on paraspeckle integrity. These proteins did not affect paraspeckle condensate integrity (Extended Data Fig. 1b). Therefore, TDP-43 appears to be unique in its ability to negatively regulate paraspeckle condensation.

The authors primarily used aneuploid HeLa cells with very high NEAT_2 expression levels (likely due to triploidy of the 11q13.1 NEAT1 locus). The use of SH-SY5Y as a neuronal model relevant to ALS is quite limited. The authors should use at least one other neuronal-specific cell line for ALS to confirm the results.

We fully appreciate the request to test TDP-43's effect in physiological models. We have now included data using minimal and uniform TDP-43 overexpression (via lentiviral transduction) in human iPSC-derived neuronal models (Fig. 1j,k; Extended Data Fig. 1j,k), which fully corroborate our data in non-neuronal models. In order to overcome technical difficulties with paraspeckle detection in neurons that have very low basal paraspeckle assembly, we optimised RNAscope-ISH with a NEAT1_2 probe. Since this detection approach includes a signal amplification step, it allows ultra-sensitive detection of NEAT1-positive particles. Using this approach, we have been able to demonstrate that only ~2-fold TDP-43 upregulation in NPCs and motor neurons leads to a defect in paraspeckle condensation and precursor accumulation.

Based on analysis of whole genome sequencing data, the authors claim that the longer 3'-end UG repeats (repeat 4) of NEAT1_2 are associated with shorter survival in a neurodegenerative disease, amyotrophic lateral sclerosis (ALS). Their model hypothesizes that longer repeats 4 can increase TDP-43 recruitment and sequestration, compromising paraspeckle homeostasis and, consequently, their cytoprotection under stress, leading to more severe ALS disease. I believe the link between the length of repeat 4 and ALS survival is not experimentally validated. The manuscript does not provide functional evidence on how longer repeats 4 impair paraspeckle functions to impact ALS progression directly. Given the complexity of ALS pathology with many contributing factors, this paraspeckle model linked to ALS is preliminary and can be overstated.

We agree that in the original version, we did not provide direct experimental evidence to support this link. We have now generated human iPSC lines with NEAT1_2 repeat 4 selectively deleted in a homozygous state (repeat length in the parental line – 24/25

repeats). In addition, we have generated NEAT1_2 KO iPSCs that do not form paraspeckles (Extended Data Fig. 14). These lines have been differentiated into human motor neurons, and their survival under stress has been analysed using longitudinal analysis on Incucyte liver imager. These experiments revealed that repeat 4 deletion significantly protects motor neurons from stress-induced toxicity. Conversely, neurons with ablated paraspeckles have lower survival under stress (Fig. 8c-e). This dataset provides a direct experimental support to the contribution of longer NEAT1_2 UG-repeat length to the disease severity in ALS.

Reviewer #2

This study addresses important questions concerning the regulation of TDP-43 function and the biology of RNP granules under stress—areas that remain largely undefined. There is value in investigating how RNA-protein interactions involve condensate properties that may be mediated by homotypic and heterotypic interactions. These are noted strengths of the manuscript.

We are very grateful for this positive assessment of the conceptual importance of our study.

However, there are major concerns regarding the experimental design that limit the interpretability of the findings. Foremost among these is the reliance on cellular overexpression of TDP-43 and FUS constructs, which can significantly alter their self-assembly behavior, condensate properties, and aggregation propensity. This in addition to the levels of overexpression of WT and mutant TDP-43 may not be physiological or pathologically-relevant. Transient transfection of these constructs typically results in widely heterogeneous expression levels complicating the interpretation of condensate formation, subcellular localization, and protein interactions. In addition, there are important questions regarding the design and interpretation of the ImmunoCon assay developed here to study phase separation of the purified proteins, which require clarification.

We appreciate these concerns and have fully addressed them, by inclusion of additional data, as described below.

Page 4, line 144. Experiments using TDP-43 fused to different tags and the sentence describing “saturating the autoregulatory mechanism” is unclear. There is no quantification of autoregulation and it is unclear whether the authors refer to autoregulation of endogenous TDP-43 or the GFP and SNAP constructs, in which case the TARDBP 3'UTR should have been included in the expression construct.

We apologise that the reference to autoregulation appeared misleading. This ambiguous statement has now been removed from the text. TDP-43 OE constructs (GFP, SNAP-Tag and untagged) used in this study contain only its ORF, without regulatory sequences. We do not observe significant downregulation of endogenous TDP-43 protein upon their expression (Extended Data Fig. 1e). However, we observe endogenous TDP-43 downregulation in optoTDP-43 expressing cells (Extended Data Fig. 4d).

Page 3, line 87. Relative Neat1_2 transcript expression upon TDP-43 OE should be quantified to determine whether changes in Neat 1_2 body size are caused by changes in levels of the transcript.

This data is now provided in Extended Data Fig. 3e. As indicated above, WT TDP-43 leads to a moderate NEAT1_2 upregulation (~50%). Therefore, the disruptive effect on paraspeckles that we observe is not due to NEAT1_2 downregulation.

Page 4, line 155. Perez-Berlanga et al. (2023) and Dos Passos et al. (2023) showed that multivalent TDP-43 interactions promote the formation of macromolecular TDP-43 complexes and these should be included as reference. Importantly, these studies and others, including work from the Hayes group, demonstrate that disruption of RNA binding, NTD-mediated oligomerization and CTD-driven self-assembly increases TDP-43 exit to the cytoplasm. In evaluating the effect of these mutations in paraspeckle formation, the studies and their conclusions should take into consideration their diminished TDP-43 nuclear localization.

We thank the Reviewer for pointing out our oversight in referencing these studies (which have been added now) and raising this important point. In our system, we did not observe obvious mislocalisation of 2M, Δ CR and Δ CTD mutants. However, we have now quantified nuclear-cytoplasmic ratio for these mutants, which has confirmed similar mean nuclear levels for these variants in HeLa cells (Extended Data Fig. 3b). We did observe some cytoplasmic redistribution for the 2M mutant – which was visible in overexposed images, however mislocalised fraction was very low and did not affect the N/C ratio (Extended Data Fig. 3b). We note that both cited studies used Hek293 cells, and we used HeLa, therefore the difference can be cell line-specific. Δ NTD mutant, which has a deletion in NLS, does become mislocalised to the cytoplasm of HeLa cells, as was shown in our previous study (PMID 38941189). Therefore, when analysing this mutant, we included only cells with TDP-43 nuclear level (fluorescence intensity values) similar to that of WT TDP-43. We have now included these considerations/discussions in the Results section.

Fig. 4. The assays used to study TDP-43 and FUS phase separation, Immuncon, lack important controls to determine whether they may be used for studying phase separation properties and behavior. The experiments should include conditions, such as site directed mutations—and preferably not large deletion constructs—previously established to disrupt TDP-43 and FUS separation. There are several questions raised by the method design and observations. The assay does not distinguish between aggregate accumulation and condensate formation. In addition, the complexes identified by antibody detection are significantly smaller than those reported in multiple studies for FUS and TDP-43 condensates. Furthermore, antibody binding to these proteins may disrupt phase separation depending on the epitopes these interact with. Similar experiments should be conducted in antibody-free conditions to demonstrate that the antibody does not interfere with protein assembly.

This is a novel assay, and we appreciate the concerns, which have been also raised by another Reviewer. We have now performed additional experiments to further validate this assay and benchmark it against the existing phase separation assays.

- As indicated in the response to Reviewer#1, we have confirmed the sensitivity of FUS condensates to high salt and their re-entrant behaviour (Extended Data Fig. 7e). In addition, we further characterised recombinant TDP-43 aggregates detected in ImmuCon using molecular chaperones bis-ANS and TMAO, previously reported to promote TDP-43 phase separation (PMID: 30520303, 33149109). Both compounds did not affect TDP-43 aggregates in ImmuCon, confirming that they are not LLPS assemblies (Extended Data Fig. 7f).
- We have utilised an assay with full-length recombinant FUS and TDP-43 proteins site-specifically labelled with small organic dyes (Cy3/Cy5) via a small peptide tag ybbR (after cleavage of the solubilising tag, MBP), as described in PMID 32302572 and 39633055. This analysis has confirmed the disruption of FUS condensation by TDP-43 (Extended Data Fig. 8b).

- On the point of antibody effect – we would like to clarify that antibody staining is performed *after* the condensate/complexes fixation step, therefore antibody binding cannot interfere with phase separation and condensate dynamics.
- On the point of smaller size of FUS condensates. Indeed, in droplet assays reported previously and in our own experiments with fluorescently labelled proteins after MBP tag cleavage, FUS condensates reach the sizes of ~5 µm. However, FUS condensate size in ImmunoCon, typically ranging from 100 nm to 1 µm, is closer to the size of paraspeckles (~350 nm) and is therefore more representative as a model of these cellular condensates.

TDP-43 phase separation in the presence of GU-repeat RNA (as shown in Fig. 4I) was previously reported by Grese et al., 2021, who observed a dramatic increase in the size of TDP-43 condensates. The authors should include a discussion addressing the potential reasons for the differences between their findings and those of Grese et al.

We thank the Reviewer for highlighting this study, which has now been referenced. We believe that our data are in agreement, since the promotion of phase separation by UG-rich RNA would create favourable conditions for TDP-43 polymerisation and clustering on NEAT1_2, by increasing the local protein concentration.

Fig 5B, PLA experiments to study TDP-43-FUS interactions should include controls of FUS and TDP-43 KD to eliminate concerns of background non-specific signal.

We have now added data both for depletion (siRNA KD) and overexpression controls for PLA experiments (Extended Data Fig. 6a,c).

Co-IP experiments to determine domain-specific interactions between TDP-43 and FUS should be performed in the absence of protein overexpression.

We have now performed these pulldown experiments in a FUS KO cell line using ectopic expression of mutants for complementation (Extended Data Fig. 9e,f). These experiments demonstrated that RNA-binding and protein-binding (low complexity) domains of FUS both contribute to the interactions with TDP-43.

The experiments relying on transient expression of GFP-TDP-43 (WT and mutant variants) throughout the manuscript should be validated using assays in which these proteins are expressed at levels comparable to endogenous TDP-43. Transient transfection introduces substantial expression heterogeneity, which can confound interpretation of results related to condensate formation, localization, and protein interactions. To address this, the authors should employ stable cell lines with inducible or constitutive expression of GFP-TDP-43 at near-endogenous levels. Such systems have been successfully used to investigate TDP-43 phase separation and function, as shown by Hallegger et al., 2021.

We appreciate the concern regarding the ectopic protein expression and variability associated with this approach. Firstly, we would like to note that the effect of TDP-43 oligomerisation was also assessed at a near-endogenous level using an opto-system. As suggested, we have now addressed this concern by using models with low and homogenous expression of untagged TDP-43, using lentiviral delivery in human neural precursors and postmitotic neurons. These cells represent models with moderate or very low paraspeckle assembly, respectively, and lentiviral delivery allowed a uniform and mild (~2-fold) TDP-43 OE. TDP-43's negative effect on paraspeckle condensation has been confirmed in these models, alleviating the concerns of OE-associated artefacts. We would like to point out that an increase in TDP-43 dosage (or its enhanced oligomerisation) is

required to modulate paraspeckle condensation, therefore some level of overexpression had to be used in our studies, to unravel this effect.

Reviewer #3:

The study by Hodgson and colleagues presents important and timely findings supporting a novel and unexpected role for TDP-43 in regulating the formation of paraspeckle spheroids. The latter are enigmatic nuclear structures with unclear functions in gene regulation. The study provides valuable insights into nuclear phase separation and the functional dynamics of TDP-43. Importantly, the work provides genetic evidence that the uncovered mechanism of TDP-43-dependent paraspeckle formation correlates with ALS disease progression, an exciting discovery, opening new avenues for investigation and potentially interfering with TDP-43 toxicity in disease, beyond its well-characterized function in suppressing cryptic exons, which is lost in neurodegeneration. Given the significant contribution of the study to the field of RNA biology and neurodegeneration, I enthusiastically recommend it for publication in Nature Cell Biology, after the authors address the following points of criticism.

We are delighted by this extremely positive and supportive assessment by this Reviewer.

1. The manuscript does not clearly state whether the GFP tag on TDP-43 is placed at the N- or C-terminus. This information is crucial for interpreting the localization and behavior of the fusion protein and should be specified not only in the Methods section but also in the Results.

We apologise for this oversight and agree that the location of a tag for TDP-43 is extremely important. In all cases, TDP-43 was tagged on its N-terminus – commonly used for this protein and confirmed to not affect its subcellular distribution and properties (e.g. PMID 37431963). This has now been stated in the Results, Methods and figure legends.

2. The image analysis presented throughout the study is elegant and enhances the quantitative interpretation of the results. However, in Figure 1G, the same analytical approach should be applied to the knockdown (KD) condition as is done for the overexpression (OE) condition. This would facilitate a more balanced and informative comparison between the two conditions.

We appreciate a difference in data presentation for the KD panels. This difference was dictated by a different approach to quantification we had to use for these opposite phenotypes. We have now included data in neurons into this figure that “separated” the gain- and loss-of-function experiments. Some knockdown data was also moved to supplementary, streamlining the presentation and message from these panels. We hope that the Reviewer will find the amended layout acceptable.

3. The findings of the current study should be compared to those from Perez-Berlanga et al. EMBO Journal, 2023, which demonstrates that RNA-deficient TDP-43 forms smaller and aberrant LLPS particles in the nucleus (using GFP-TDP-43 in HEK cells, similarly as in the current study). Furthermore, Perez-Berlanga et al. showed that monomeric TDP-43, carrying the same (6M) mutations fails to form any droplets—except for the occasional association with Cajal bodies. Can the authors explain the discrepancy in their findings? These comparisons, similarities, and differences between the studies should be acknowledged and critically discussed in the Discussion section.

We indeed observe prominent nuclear granulation for RNA-binding deficient mutants – also observed in previous studies, e.g. PMID 25556531, 38582774. This discrepancy can be due to the expression level – Perez-Berlanga et al. reported reduced stability for their

RRMm mutant resulting in 5x reduction in its levels compared to WT TDP-43. Higher levels achieved in our model likely promote nuclear aggregation of this variant and formation of stable, non-LLPS structures (e.g. PMID 34239072).

Secondly, Hek293 cells used in Perez-Berlanga et al. have low paraspeckle numbers – unlike HeLa that have prominent paraspeckles (*Figure R2 – for Reviewer only*). NTD mutants can be recruited to paraspeckles (Fig. 2b, Fig. 5e), which explains higher granulation in the nucleus of HeLa as compared to Hek293 cells. However, on average, we do observe less granulation in the nucleus for all oligomerisation-deficient mutants, as seen in Extended Data Fig. 3b, which is in agreement with Perez-Berlanga et al. These considerations have been acknowledged in the Results section.

Figure R2. Paraspeckles in Hek293 vs. HeLa cells. Scale bar, 20 μ m.

4. *The claim that the 2M variant behaves like a monomer lacks sufficient supporting evidence. The authors should provide experimental validation or reference prior work that supports this assertion.*

We have now performed DSG cross-linking analysis for NTD and CTD mutants (2M and Δ CR), which both demonstrated severely reduced oligomer formation compared to WT protein (Extended Data Fig. 3c).

5. *The manuscript notes that ALS-linked mutations (Q331K and M337V) do not prevent paraspeckle dispersal, which is intriguing. While this is stated, the data are not shown. Including these results, even in supplementary figures, would significantly strengthen the manuscript.*

The data for three fALS mutants, M337V, D169G and Y374X (images and quantification) have now been included (Extended Data Fig. 3d).

6. *The study would benefit from a deeper exploration of the functional consequences of paraspeckle dispersal. Specifically, is there any evidence of cellular toxicity or other phenotypic effects that result from the altered phase behavior of TDP-43? This is particularly important given the potential role of these findings for neurodegenerative diseases.*

We absolutely agree that physiological and pathological insights into the function of paraspeckle complexes are of outstanding interest. We have now more deeply explored the consequences of (altered) TDP-43 oligomerisation on NEAT1_2 regulation and paraspeckle assembly. We were initially puzzled by the finding that TDP-43 OE leads to NEAT1_2 downregulation (Extended Data Fig. 3e) - when the opposite is expected given TDP-43's ability to promote NEAT1 polyadenylation and isoform switch in favour of NEAT1_1 (previously established using TDP-43 depletion, PMID 31047794). We found that in contrast, OE of TDP-43 2M or Δ CR mutants that are predominantly monomeric (Extended Data Fig. 3d) downregulates NEAT1_2 (Extended Data Fig. 3e). This suggests

that NEAT1 isoform ratio is predominantly regulated by monomeric TDP-43, whereas an increase in TDP-43 oligomerisation by its OE (including with its endogenous counterpart) may lead to a loss-of-function phenotype, at least in paraspeckle regulation. Therefore, altered TDP-43 oligomerisation would impact not only paraspeckle condensation but also regulation of the NEAT1 isoform ratio. These results have now been included and discussed in the manuscript.

7. The authors should clarify how they distinguish between phase-separated droplets and aggregates in the ImmuCon assay. Are there specific markers or dynamic properties that were used to make this distinction?

This point was also raised by other Reviewers and has now been addressed by including additional data on FUS and TDP-43 assembly properties in ImmuCon. Firstly, we confirmed the sensitivity of FUS condensates to high salt and FUS re-entrant phase behaviour in this assay (Extended Data Fig. 7e). Further characterisation of the recombinant TDP-43 aggregates, using two molecular chaperones, bis-ANS and TMAO (PMID 30520303, 33149109), also confirmed their non-LLPS nature (Extended Data Fig. 7f).

8. *The formation of the multiphase structures within paraspeckle spheroids is elegant and intriguing. What is the role of these structures in cells?*

We thank the Reviewer for acknowledging an advance in understanding the paraspeckle structure provided in our study. Our data suggest that this internal condensate complexity regulates their properties (for example, paraspeckles lacking TDP-43 are more dynamic, Fig. 6g-i). A significant effort will be required to unravel the emergent properties of paraspeckle condensates, including other roles of their micro-domains, in the future studies.

9. *Given the proposed role of the described mechanism for ALS, the authors should perform some of their key experiments in neurons. While I understand the potential hesitation of the authors to do this, due to consideration of technical limitations in culturing neurons, I can assure them that the widely used NGN2 human neurons are easy to obtain and can be differentiated rapidly, allowing results within a short amount of time (within a few weeks).*

This was requested by other Reviewers and has now been fully addressed using human iPSC-derived neuronal models. Firstly, we have validated the paraspeckle dispersal phenotype upon an increase in TDP-43 dosage in NPCs and motor neurons, for which RNAscope-ISH paraspeckle detection was established and optimised (Fig. 1j,k; Extended Data Fig. 1j,k). Secondly, we have generated human iPSC lines with NEAT1_2 UG-repeat 4 selectively deleted in a homozygous state (repeat length parental line – 24/25 repeats). In addition, we have generated NEAT1_2 KO iPSCs lacking paraspeckles (Extended Data Fig. 14). These lines have been differentiated into human motor neurons, and their survival under stress has been characterised using longitudinal analysis on Incucyte. These experiments revealed that repeat 4 deletion significantly protects motor neurons from stress-induced toxicity. Conversely, neurons with ablated paraspeckles have lower survival under stress (Fig. 8c-e). This dataset provides a direct experimental support to the contribution of longer repeat length to the disease severity in ALS.

10. The annotation of the mutations in the schematic of Figure 2A is not intuitive. It would

be helpful to adopt the same labeling style used in Figure 2E, where mutations are annotated directly above or below the domain lines to indicate their positions.

We have now unified the labelling in this figure, as suggested.

We thank the Reviewers for additional feedback. We have now implemented the suggested changes, by making amendments and additions in the text and adding Figure R1 to an extended data figure.

Reviewer #1:

I have re-evaluated the manuscript in light of the authors' rebuttal and newly presented data. The revisions have substantially strengthened the study, adding conceptual clarity and experimental depth. However, a few areas remain underdeveloped—particularly the mechanistic integration of TDP-43 clusters within the micellization framework and the subtle differences between basal and stress-induced paraspeckles.

1. The revised Discussion now appropriately presents TDP-43 as a modulator of paraspeckle micellization rather than a direct surfactant, providing a more precise and accurate interpretation of its role. This section would, however, benefit from a clearer mechanistic link between TDP-43's gel-like (non-liquid) clusters and the physical parameters of micellization, specifically, how such rigid subphases might influence paraspeckle fluidity, curvature, or size criticality. Please expand the Discussion with one to two sentences addressing this connection, supported by relevant recent literature, to add conceptual depth and avoid the impression of a superficial alignment with the micelle model.

We thank Reviewer for this suggestion. Relevant text has now been added to the Discussion, as requested.

2. The authors' clarification of terminology ("nuclear clusters" versus "stress-induced de novo nuclear condensates of TDP-43 -TCs") and the negative enrichment data for Sirt1 and nucleoporins (Figure R1) effectively differentiate these structures. The results support the interpretation that TDP-43 clusters are relatively homogeneous; however, given that full-length TDP-43 forms gel-like condensates with restricted dynamics, it remains plausible that stress-responsive chaperones (e.g., HSP70) could be incorporated into paraspeckle-embedded clusters. To address this point more completely, please include Figure R1 in the Extended Data and briefly discuss this possibility in the main text.

We have now included this figure in Extended data fig. 8.

3. The authors' observation that stress-induced and basal paraspeckles exhibit no significant differences in size or core suborganization is reasonable, and a detailed comparative study would indeed be beyond the current scope of the manuscript. To enhance broader relevance, however, it would be helpful to briefly discuss potential implications for cytoprotective function, particularly in light of recent findings (PMID: 40661327) that NEAT1 depletion coincides with TDP-43 loss in amyotrophic lateral sclerosis motor neurons, potentially affecting stress responses. Please add one or two sentences to the Discussion highlighting this link as a possible future direction.

We have now cited this reference and extended the Discussion on the implications of the neuroprotective potential of paraspeckles in ALS. We would like to point out that although our results confirm NEAT1_2 upregulation upon TDP-43 loss (unlike the above report), both studies support loss of paraspeckle function as a disease-contributing factor.

Reviewer #2:

The authors carried out a comprehensive investigation of the interactions and regulation of paraspeckle formation by TDP-43 that is mediated by RNA binding and changes in phase separation properties. The revised manuscript is greatly improved with new experiments and discussion in response to the reviewers' comments. Still, one important limitation is the use of large tags and overexpression systems for many of the assays presented. These two modifications can significantly impact the interpretation of TDP-43 studies. This should be mentioned in the report.

We thank this Reviewer for this positive assessment. We would like to emphasise that experiments were repeated with untagged TDP-43, both for non-neuronal cells and neurons, with a similar result. Likewise, for *in vitro* studies, we used an independent assay with Cy3/5-labelled proteins, in the absence of large tags. Nevertheless, we have now included a statement of the large tag effect when describing the above validation studies in the Results section.

A possible explanation for the differences observed in mutant TDP-43 nuclear-cytoplasmic ratio between these studies (Extended Data Fig. 3) and previous reports should be included in the manuscript. The authors should also consider differences in cellular distribution due to the expression of fusion proteins that are considerably larger than untagged protein or short peptide tags.

We have now mentioned cell line specificity as a possible explanation of these discrepancies in the Results section. We also note that at least for the data from Perez-Berlanga et al., the size of the tag was not the main differential factor, because that study also used GFP-tagged TDP-43.

Furthermore, the authors should clarify in the manuscript whether monomeric GFP, or non-modified GFP was used as protein tag. This is important because GFP molecules are naturally dimers, unless the monomeric variant is used. This difference may alter tagged protein nuclear-cytoplasmic distribution, phase separation and cellular dynamics.

We used non-modified GFP in these studies. We agree that GFP dimerization may affect condensation of tagged proteins – this was a reason for performing validation studies with SNAP-tag and untagged protein. We have now stated this in the text: *“Since large tags and GFP dimerization can affect the properties of tagged proteins, we sought to validate these findings using different systems. Untagged TDP-43 and TDP-43 with SNAP-tag also disrupted paraspeckles (Extended Data Fig. 1d-f), confirming that it is a TDP-43 specific effect.”*

Reviewer #3:

The authors have successfully addressed all my concerns and, as far as I can tell, also those from the other referees. I think the addition of new experiments, especially the demonstration of the stress-induced death in human motor neurons, strengthens the study and its conclusions. Overall, I think the work elucidates a previously unrecognized role of TDP-43 in paraspeckle formation and dynamics that because functionally important upon cellular stress, and has clear implications for understanding disease mechanisms linked to ALS and related neurodegenerative diseases. It also points to a potential mechanism of

toxicity linked to TDP-43 overexpression that has long been used to model TDP-43 proteinopathies in cellular and animal models. Therefore, I think the work is highly relevant and appropriate for publication in Nature Cell Biology.

We are delighted to receive such a positive evaluation from this Reviewer.